# ROBUST DEEP REINFORCEMENT LEARNING AGAINST ADVERSARIAL BEHAVIOR MANIPULATION

**Shojiro Yamabe**[1]**, Kazuto Fukuchi**[2,3]**, Jun Sakuma**[1,3]
[1]Institute of Science Tokyo, [2]University of Tsukuba, [3]RIKEN AIP
`yamabe.s.2fb0@m.isct.ac.jp`

## ABSTRACT

This study investigates behavior-targeted attacks on reinforcement learning and their countermeasures. Behavior-targeted attacks aim to manipulate the victim's behavior as desired by the adversary through adversarial interventions in state observations. Existing behavior-targeted attacks have some limitations, such as requiring white-box access to the victim's policy. To address this, we propose a novel attack method using imitation learning from adversarial demonstrations, which works under limited access to the victim's policy and is environment-agnostic. In addition, our theoretical analysis proves that the policy's sensitivity to state changes impacts defense performance, particularly in the early stages of the trajectory. Based on this insight, we propose time-discounted regularization, which enhances robustness against attacks while maintaining task performance. To the best of our knowledge, this is the first defense strategy specifically designed for behavior-targeted attacks.

## 1 INTRODUCTION

Applications of Deep Reinforcement Learning (DRL) have grown significantly in recent years (Berner et al., 2019; Ouyang et al., 2022; Guo et al., 2025). However, when DRL is deployed in mission-critical tasks (Sallab et al., 2017; Degrave et al., 2022; Yu et al., 2021), it is crucial to understand its susceptibilities to adversarial attacks and deal with them. One such vulnerability is the manipulation of a victim's behavior by an adversary that can deceive the state observed by a victim agent (Huang et al., 2017; Zhang et al., 2020b; Sun et al., 2022).

The primary objective of this study is to introduce *behavior-targeted attacks* and corresponding countermeasures. In this attack, the adversary's goal is to steer the victim's trained policy toward a target policy specified by the adversary. Behavior-targeted attacks introduce novel threats that cannot be realized by conventional *reward-minimization attacks*, where the adversary's goal is to minimize the victim's reward (Pattanaik et al., 2018; Zhang et al., 2021; McMahan et al., 2024). Moreover, as we detail later, existing defenses tailored to counter reward-minimization attacks are ineffective against these novel threats, which underscores the necessity of dedicated countermeasures. Below, we illustrate two scenarios where the behavior-targeted attacks are particularly relevant.

Scenario 1. Consider an autonomous vehicle controlled by a DRL agent that receives higher rewards for reaching its destination quickly and safely, but incurs a large penalty if it causes an accident (Kiran et al., 2021). Reward-minimization attacks would aim to induce an accident at some point along the vehicle's route. In contrast, behavior-targeted attacks can take various forms, regardless of the victim's reward. For instance, the adversary could manipulate the vehicle to slow down at a specific crossing point, creating congestion in a strategically chosen region. Alternatively, the adversary might steer the autonomous vehicle toward a particular store to generate economic benefit for the adversary.

Scenario 2. Consider a recommendation system powered by DRL (Chen et al., 2023; Fu et al., 2022) that maximizes user satisfaction (reward) by observing the user's purchase history and recommending relevant items. In this context, reward-minimization attacks attempt to degrade user satisfaction by generating suboptimal recommendations. In contrast, behavior-targeted attacks manipulate the system to serve specific objectives, such as prompting certain products that benefit the adversary or suppressing the recommendation of items specified by the adversary.

**Attack.** Although several behavior-targeted attacks have been proposed (Hussenot et al., 2020; Boloor et al., 2020; Bai et al., 2024; 2025), they all share a critical limitation: requiring white-box access to the victim's policy, including its architecture and parameters. In many real-world scenarios, the adversary is unlikely to have full knowledge of the victim's policy, which severely limits practical applicability.

To overcome this limitation, we show that an optimal adversarial policy can be learned without requiring white-box access to the victim's policy. Specifically, we present a novel theoretical reformulation of the behavior-targeted attack objective as the problem of finding an adversarial policy that maximizes cumulative reward in an MDP specially constructed for the attack purpose. Here, the adversarial policy replaces the victim's observed state with a falsified state at each time step (see Figure 1). Crucially, because the victim's policy is incorporated into the transition dynamics of the constructed MDP, training the adversarial policy does not require white-box access.

Building on this formulation, we propose the Behavior Imitation Attack (BIA), a novel attack framework that is applicable even when access to the victim's policy is severely limited. Since the reformulated objective can be optimized via demonstration-based imitation learning, we utilize established algorithms (Ho & Ermon, 2016; Kostrikov et al., 2018; Chang et al., 2024). The advantage of leveraging imitation learning is that BIA enables training adversarial policies directly from demonstrations of the desired behavior, eliminating the need for reward modeling. The adversary can readily prepare demonstrations necessary for BIA simply by performing the desired behavior several times. We empirically show that even under the most restrictive no-box setting, where the adversary cannot observe any output of the victim's policy, BIA achieves attack performance competitive with baselines requiring white-box access.

**Defense.** Most existing defense strategies, such as adversarial training (Zhang et al., 2021; Oikarinen et al., 2021; Sun et al., 2022; Liang et al., 2022), certified defenses (Wu et al., 2022; Kumar et al., 2022; Mu et al., 2024; Sun et al., 2024; Wang et al., 2025), and regret-based robust learning (Jin et al., 2018; Rigter et al., 2021; Belaire et al., 2024), are not readily adaptable to counter behavior-targeted attacks. This is because the adversary's target policy is unknown to the defender and determined independently of the victim's reward. Although policy smoothing (Shen et al., 2020; Zhang et al., 2020b) may offer some degree of robustness by stabilizing the policy's outputs against adversarial perturbations, their theoretical robustness guarantees are limited to reward-minimization attacks and not directly extended to behavior-targeted attacks. Moreover, policy smoothing often degrades performance on the victim's original tasks, as it imposes excessive constraints on the policy's representational capacity.

To address these challenges, we derive a tractable upper bound on the adversary's gain that holds for arbitrary target policies and integrate it into a robust training objective. Our theoretical analysis of the upper bound reveals two key insights: (i) reducing the sensitivity of the policy's action outputs to state changes improves robustness, and (ii) this effect is particularly pronounced when sensitivity is suppressed in the early stages of trajectories.

Motivated by these insights, we introduce Time-Discounted Robust Training (TDRT), which incorporates a time-discounted regularization term to suppress the sensitivity of the policy's actions. This regularization more strongly reduces the sensitivity at critical earlier stages of trajectories and progressively weakens at later stages. By concentrating regularization in early trajectories, TDRT maintains the policy's representational capacity and thus preserves original-task performance, while simultaneously enhancing robustness against behavior-targeted attacks. In our experiments, compared to uniform policy-smoothing defenses without time-discounting, TDRT improves original-task performance by 28.2% while maintaining comparable robustness. To the best of our knowledge, TDRT is the first defense specifically designed for behavior-targeted attacks.

**Contributions.** Our primary contributions are summarized as follows:

- We theoretically reformulate the objective of behavior-targeted attack and introduce the Behavior Imitation Attack (BIA), a novel method that leverages well-established imitation learning algorithms and operates under limited access to the victim's policy.

- We present Time-Discounted Robust Training (TDRT), the first defense tailored to behavior-targeted attacks. Time-discounted regularization in TDRT is grounded in our theoretical analysis and mitigates original-task performance degradation while preserving robustness.

- We demonstrate the effectiveness of our proposed attack and defense methods using the Meta-World (Yu et al., 2020), MuJoCo (Todorov et al., 2012), and MiniGrid (Chevalier-Boisvert et al., 2018) environments.

## 2 RELATED WORKS

### 2.1 ATTACK METHODS

Existing attack methods on DRL agents can be broadly classified into three categories: reward-minimization attacks, enchanting attacks, and behavior-targeted attacks. Reward-minimization attacks aim to reduce the victim's cumulative reward and have been extensively studied (Zhang et al., 2021; Sun et al., 2022; McMahan et al., 2024). As another direction, enchanting attacks aim to lure the victim into a predetermined terminal state without specifying the full trajectory (Ying et al., 2023). Since neither reward-minimization nor enchanting attacks are intended to manipulate the victim's entire behavior, we treat them as out of scope.

Behavior-targeted attacks instead aim to steer the victim toward an adversary-specified target policy. However, existing approaches typically assume white-box access to the victim's policy, as discussed above. In addition, some attacks (Hussenot et al., 2020; Boloor et al., 2020) are tailored to specific domains such as autonomous driving, which limits their generalizability. More recently, Bai et al. (2024; 2025) proposed a non-heuristic attack based on preference-based reinforcement learning, but it still relies on white-box access and requires three separate stochastic optimization procedures, making it highly resource-intensive.

While several studies have proposed poisoning attacks that intervene during the victim's training phase (Sun et al., 2021; Rangi et al., 2022; Xu et al., 2023; Xu & Singh, 2023; Rathbun et al., 2024), they fall outside our threat model (see Appendix A.1 for details).

### 2.2 DEFENSE METHODS

Adversarial training (Zhang et al., 2021; Oikarinen et al., 2021; Sun et al., 2022; Liang et al., 2022; Li et al., 2024a) optimizes the agent to be robust against adversarial perturbations generated by a hypothetical adversary. However, the effectiveness of adversarial training against behavior-targeted attacks is limited, as the defender cannot anticipate which behavior the adversary aims to impose. Existing policy smoothing methods (Shen et al., 2020; Zhang et al., 2020b) are designed for reward-minimization attacks and significantly degrade performance on the original tasks as previously noted.

Other approaches include regret-based robust learning (Jin et al., 2018; Rigter et al., 2021; Belaire et al., 2024) and certified defenses (Wu et al., 2022; Kumar et al., 2022; Mu et al., 2024; Sun et al., 2024; Wang et al., 2025). Regret is defined as the difference between the victim's attacked and unattacked rewards, and regret-based robust learning focuses on minimizing this difference. Certified defenses guarantee a lower bound on the victim's reward under attack. As they both use reward-based metrics and are designed for reward-minimization attacks, they are less effective against behavior-targeted attacks that are independent of the victim's reward.

Liu et al. (2024) proposed an adaptive defense method robust against multiple types of attacks, not limited to reward-minimization attacks. In their method, the defender prepares multiple robust policies and then selects the policy that maximizes the victim's reward under attack based on rewards obtained in previous episodes. However, our approach focuses on a single static victim in a stationary environment, so their adaptive setting differs from ours.

Several works (Sun & Zheng, 2024; YANG & Xu, 2024) propose diffusion-based defenses, which perform purification at inference time to remove adversarial perturbations from corrupted states. In contrast, our approach is a training-time regularization method and requires no additional preprocessing during inference. As a result, these defenses operate on different components of the pipeline, making them orthogonal and complementary. For a more detailed and comprehensive review of related work, please refer to Appendix A.

## 3 PRELIMINARIES

**Notation.** We denote the Markov Decision Process (MDP) as $(\mathcal{S}, \mathcal{A}, R, p, \gamma)$, where $\mathcal{S}$ is the state space, $\mathcal{A}$ is the action space, $R : \mathcal{S} \times \mathcal{A} \to \mathbb{R}$ is the reward function, and $\gamma \in (0, 1)$ is the discount factor. We use $\mathcal{P}(\mathcal{X})$ as the set of all possible probability measures on $\mathcal{X}$. We denote $p : \mathcal{S} \times \mathcal{A} \to \mathcal{P}(\mathcal{S})$ as the transition probability, $\pi : \mathcal{S} \to \mathcal{P}(\mathcal{A})$ as a stationary policy, and $p_0$ as an initial state distribution. When the agent follows policy $\pi$ in MDP $M$, the objective of reinforcement learning is to train a policy that maximizes $J_{\text{RL}}(\pi) \triangleq \mathbb{E}_\pi^M [R(s, a, s')] = \mathbb{E} [\sum_{t=0}^\infty \gamma^t R(s_t, a_t, s_{t+1})]$, the expected sum of discounted rewards from the environment.

**State-Adversarial Markov Decision Process.** To model the situation where an adversary intervenes in the victim's state observations, we use SA-MDP (Zhang et al., 2020b). Let the victim follow a fixed policy $\pi$ in the MDP $M$. In an SA-MDP, the adversary introduces an adversarial policy $\nu : \mathcal{S} \to \mathcal{P}(\mathcal{S})$ that interferes with the victim's state observations by making the victim observe a false state $\hat{s} \sim \nu(\cdot|s)$ at each time step without altering the true state of the environment (see Figure 1).

The SA-MDP is defined as $M = (\mathcal{S}, \mathcal{A}, R, \mathcal{B}, p, \gamma)$, where $\mathcal{B}$ is a mapping from the true state $s \in \mathcal{S}$ to the false state space $\mathcal{B}(s) \subseteq \mathcal{S}$ that the adversarial policy can choose from. The size of $\mathcal{B}(s)$ indicates the adversary's intervention capability and is typically set in the neighborhood of $s$. The smaller the size of $\mathcal{B}(s)$, the more challenging it becomes to achieve the attack objective.

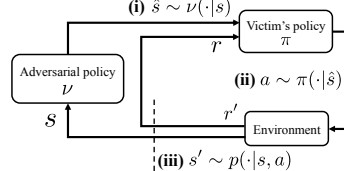

Figure 1: Overview of SA-MDP

## 4 THREAT MODEL

This section outlines the threat model for the behavior-targeted attack. We assume that the victim follows a fixed policy $\pi$, and it observes falsified states generated through the adversary's intervention. We define this process as an SA-MDP $M = (\mathcal{S}, \mathcal{A}, R, \mathcal{B}, p, \gamma)$. The adversary's objective is to manipulate the victim into performing a specific behavior desired by the adversary, which is denoted by the target policy $\pi_{\text{tgt}}$. To achieve this objective, the adversary trains an adversarial policy $\nu$ that makes the victim observe a falsified state at each time step. In this context, the adversary's influence against the victim is characterized as follows:

**Access to the victim's policy.** We assume the adversary does not have full access to the victim's policy $\pi$, including its neural network parameters and training algorithm. Specifically, we define two access models: black-box and no-box. In the black-box, the adversary can only observe the victim's policy inputs (i.e., observed states) and the corresponding outputs (i.e., actions taken) at each time step. In the no-box, the adversary observes only the policy inputs at each time step. Since the no-box setting provides less information than the black-box, it presents a greater challenge to the adversary.

**Intervention ability on victim's state observations.** We assume that the adversary can alter the victim's state observation from the true state $s$ to the false state $\hat{s} \in \mathcal{B}(s)$. This assumption is common in most adversarial attacks on RL (Zhang et al., 2020b; 2021; Sun et al., 2022).

## 5 ATTACK METHOD

**Formulation of adversary's objective.** First, we define a composite policy of the victim's policy and the adversarial policy to define the adversarial objective. Let $s \in \mathcal{S}$ be a true state. The adversarial policy provides the falsified observation $\hat{s} \sim \nu(\cdot|s)$, and then the victim selects the next action by $a \sim \pi(\cdot|\hat{s})$. Consequently, the victim's resulting behavior policy under the adversary's influence is represented by the composite policy $\pi \circ \nu$:

$$\pi \circ \nu(a|s) \triangleq \sum_{\hat{s} \in \mathcal{S}} \nu(\hat{s}|s)\pi(a|\hat{s}). \tag{1}$$

Building on this definition, we define the adversary's objective as finding the optimal adversarial policy that aligns $\pi \circ \nu$ with $\pi_{\text{tgt}}$:

$$\arg\min_{\nu} J_{\text{adv}}(\nu) \triangleq \mathcal{D}(\pi \circ \nu, \pi_{\text{tgt}}), \tag{2}$$

where $\mathcal{D}$ is some divergence measure between two policies.

Unfortunately, existing imitation learning algorithms are not directly applicable for optimizing equation 2 under our threat model. Typical imitation learning methods treat the composite policy $\pi_\nu$ as the optimizable policy and assume it is directly updatable via gradients. However, since the adversary lacks white-box access to the victim policy $\pi$, it cannot backpropagate through $\pi$ to compute the gradient of the objective with respect to $\nu$. Therefore, this assumption does not hold.

To overcome this difficulty, we provide a novel theoretical result that, under a mild assumption, equation 2 can be reformulated into a problem that maximizes the cumulative reward in another MDP $\hat{M}$ in which the adversarial policy itself serves as the policy to be optimized:

**Theorem 5.1.** *Consider an SA-MDP $M = (\mathcal{S}, \mathcal{A}, R, \mathcal{B}, p, \gamma)$ with adversarial policy $\nu$. Let $\pi$ denote the victim's policy and $\pi_{tgt}$ the target policy. Assume that the divergence $\mathcal{D}$ admits the following variational representation:*

$$\mathcal{D}(\pi \circ \nu, \pi_{tgt}) = \max_{d} \Big[ \mathbb{E}_{\pi \circ \nu}[g(d(s, a, s'))] + \mathbb{E}_{\pi_{tgt}}[-f(d(s, a, s'))] \Big], \tag{3}$$

*where $f$ and $g$ are arbitrary convex and concave functions, respectively, and $d : \mathcal{S} \times \mathcal{A} \times \mathcal{S} \to \mathbb{R}$ is a discriminator. Let $d_\star$ be the optimal discriminator in equation 3. Under this assumption, define the reward function $\hat{R}_d$ and the state transition probability $\hat{p}$ as follows:*

$$\hat{R}_d(s, \hat{s}, s') = \begin{cases} -\frac{\sum_{a \in \mathcal{A}} \pi(a|\hat{s}) p(s'|s,a) g(d_\star(s,a,s'))}{\sum_{a \in \mathcal{A}} \pi(a|\hat{s}) p(s'|s,a)} & \text{if } \hat{s} \in \mathcal{B}(s) \\ C & \text{otherwise}, \end{cases} \tag{4}$$

$$\hat{p}(s'|s, \hat{s}) = \sum_{a \in \mathcal{A}} \pi(a|\hat{s}) p(s'|s, a), \tag{5}$$

*where $C$ is a large negative constant. Then, for the MDP $\hat{M} = (\mathcal{S}, \mathcal{S}, \hat{R}_d, \hat{p}, \gamma)$, it holds that:*

$$\arg\min_{\nu} \mathcal{D}(\pi \circ \nu, \pi_{tgt}) = \arg\max_{\nu} \mathbb{E}_{\nu}^{\hat{M}} \big[ \hat{R}_d(s, \hat{s}, s') \big]. \tag{6}$$

The proof is provided in Appendix B.1. Crucially, any standard RL algorithm can maximize the cumulative reward under MDP $\hat{M}$ without requiring white-box access to $\pi$ since the policy for MDP $\hat{M}$ depends solely on $\nu$ rather than on the composite $\pi \circ \nu$. The dynamics of the MDP $\hat{M}$ encapsulate the victim's policy, and this idea has been considered in (Schott et al., 2024; Bai et al., 2025). However, their idea was not theoretically justified because (Schott et al., 2024) only introduces the concept abstractly without empirical validation or theoretical analysis, and (Bai et al., 2025) lacks a proof and concrete reward design. We provide a complete proof and present a concrete reward design for learning the optimal adversarial policy for the first time.

**Behavior Imitation Attack (BIA).**    Building on the above insight, we propose Behavior Imitation Attack (BIA). Specifically, we bridge the behavior-targeted attacks and established Imitation Learning (IL) (Ho & Ermon, 2016; Torabi et al., 2019; Chang et al., 2024) via Theorem 5.1. We then present a practical algorithm for implementing BIA.

IL trains a policy that mimics an expert policy $\pi_E$ from given demonstrations without requiring rewards from the environment. Its objective can be expressed analogously equation 2:

$$\arg\min_{\pi} J_{\text{IL}}(\pi) \triangleq \mathcal{D}(\pi, \pi_E). \tag{7}$$

We distinguish two IL settings: IL from demonstration (ILfD) and IL from observation (ILfO). In ILfD, demonstrations are provided as sequences of state–action pairs of length $T$: $\tau_E = \{s_0, a_0, \ldots, s_{T-1}, a_{T-1} \mid a_t \sim \pi_E(\cdot|s_t), s_{t+1} \sim p(\cdot|s_t, a_t)\}$. In contrast, in ILfO, demonstrations consist of only state sequences. Due to the lack of information about the expert's actions, ILfO is

---

**Algorithm 1** Behavior Imitation Attack (BIA) in GAIL

---
1: **Input:** victim's policy $\pi$, initial adversarial policy $\nu_\theta$, initial discriminator $D_\phi$, demonstration $\tau_{\text{tgt}}$, Batch size $B$, learning rates $\alpha_D, \alpha_\nu$
2: **Output:** Optimized adversarial policy $\nu_\theta$
3: **for** $n = 0, 1, 2, \ldots$ **do**
4: $\quad \tau \leftarrow \emptyset$
5: $\quad$ **for** $b = 1$ to $B$ **do**
6: $\quad\quad \hat{s} \leftarrow \text{Project}(\nu_\theta(s), \mathcal{B}(s))$
7: $\quad\quad a \sim \pi(\cdot|\hat{s}), s' \sim p(\cdot|s, a)$
8: $\quad\quad \tau \leftarrow \tau \cup \{(s, \hat{s}, a, s')\}, s \leftarrow s'$
9: $\quad$ **end for**
10: $\quad \phi \leftarrow \phi + \alpha_D \nabla_\phi [\sum_{(s,a) \in \tau} \log D_\phi(s, a) - \sum_{(s,a) \in \tau_{\text{tgt}}} \log(1 - D_\phi(s, a))]$
11: $\quad \theta \leftarrow \theta + \alpha_\nu \nabla_\theta \mathbb{E}_{(s,a) \sim \tau}[-\log D_\phi(s, a)]$
12: **end for**

---

known to be empirically more challenging than ILfD. In our attack setting, the adversary has access to a demonstration $\tau_{\text{tgt}}$ generated by the target policy $\pi_{\text{tgt}}$.

By using IL algorithms that satisfy the assumptions of Theorem 5.1, we can carry out attacks even under black-box and no-box settings. Specifically, many IL algorithms rely on divergences that admit a variational representation. Thus, by reformulating the adversary's objective using Theorem 5.1, we can leverage these well-established methods to train the adversarial policy.

As a concrete example of an IL algorithm that holds the assumption, we illustrate how to apply BIA to GAIL(Ho & Ermon, 2016), a representative ILfD method. In GAIL, the adversary's objective function can be rewritten in the following variational form using a discriminator $D \colon \mathcal{S} \times \mathcal{A} \to [0, 1]$:

$$\arg\min_\nu \mathcal{D}(\pi \circ \nu, \pi_{\text{tgt}}) = \max_D \ \mathbb{E}_{\pi \circ \nu}^M[\log D(s, a)] \ + \ \mathbb{E}_{\pi_{\text{tgt}}}^M[\log(1 - D(s, a))]. \tag{8}$$

The second term is computed with demonstration $\tau_{\text{tgt}}$ instead of $\pi_{\text{tgt}}$ in practice. In Appendix C, we show that under our problem setting, the adversary's objective function can be reduced to a distribution matching problem. Consequently, we confirm that equation 8 holds exactly. By Theorem 5.1, this then reduces to the following optimization problem:

$$\arg\max_\nu \mathbb{E}_\nu^{\hat{M}}\big[\hat{R}_D(s, a)\big], \tag{9}$$

where $\hat{R}_D$ is the function obtained by replacing the discriminator $g(d_\star)$ with $\log D_\star$ in equation 4. This is a standard RL problem and can be solved using any RL algorithm. In Appendix D, we further show how BIA extends to other approaches, including ILfO (Torabi et al., 2019) and the state-of-the-art ILfD algorithm (Chang et al., 2024).

We present a practical algorithm for BIA that utilizes GAIL to train an adversarial policy in Algorithm 1. Since $\hat{R}$ in equation 4 represents the conditional expectation of $-g(d_\star(s, a, s'))$ for the adversarial policy, we simply approximate it by a single sample $g(d_\star(s, a, s')) = \log D(s, a)$ at each step. This approximation preserves exactness in expectation, so there is no discrepancy between Theorem 5.1 and Algorithm 1. As our approach does not require direct access to the victim's policy, BIA can be applied in a black-box setting. Furthermore, when ILfO is used, the discriminator $D$ only needs states and the next states to train the adversarial policy. Since it does not require the victim's policy's selected actions, our method is applicable even in a no-box setting. Also, to stabilize the learning process, we avoid imposing large negative rewards; instead, we limit the range of false states, ensuring that the adversarial policy is enforced to select falsified states in $\mathcal{B}(s)$ in our experiments.

## 6 DEFENSE METHOD

In this section, we propose a countermeasure against behavior-targeted attacks. In this attack, the adversary's objective is defined independently of the victim's reward. As a result, the victim can be steered toward unintended and potentially malicious behaviors even when the victim's cumulative reward does not decrease. Therefore, existing defenses that aim to prevent reward degradation may

fail to provide sufficient robustness under this threat. To address this limitation, we incorporate the adversary's gain as a safety metric.

**Formulation of defender's objective.** Let $R_{\text{tgt}} \colon \mathcal{S} \times \mathcal{A} \to \mathbb{R}$ be the reward function that is maximized when the victim's behavior exactly matches the adversary's specification. We then formulate the defender's objective as follows:

$$\arg\min_{\pi} J_{\text{def}}(\pi) = -J_{\text{RL}}(\pi) + \lambda \left( \max_{\nu} \mathbb{E}^M_{\pi \circ \nu}[R_{\text{tgt}}(s,a)] - \mathbb{E}^M_{\pi}[R_{\text{tgt}}(s,a)] \right), \tag{10}$$

where $\lambda$ is a hyperparameter that adjusts the trade-off between performance and robustness. $R_{\text{tgt}}$ represents the adversary's gain, and the second term represents the additional gain achievable by the worst-case adversarial policy. Thus, equation 10 seeks to minimize this worst-case increase in the adversary's gain. We note that equation 10 subsumes prior defense objectives against reward-minimization attacks via the choice of $R_{\text{tgt}}$. In particular, setting $R_{\text{tgt}} = -R$ makes the second term a regularizer that suppresses reward drops under attack, yielding a standard robustness objective.

Since the defender cannot access the adversary's intended behavior, equation 10 cannot be optimized directly. In particular, the adversary's objective is specified independently of the victim's reward, so the defender cannot compute the target reward $R_{\text{tgt}}$. Consequently, the gradients required to optimize equation 10 are unavailable.

**Time-Discounted Robust Training (TDRT).** To optimize equation 10, we prove that the increase in the cumulative reward from the reward function $R_{\text{tgt}}$ in equation 10 is bounded by the sensitivity of the policy's action outputs to state changes:

**Theorem 6.1.** *Let $R_{tgt} \colon \mathcal{S} \times \mathcal{A} \to \mathbb{R}$ be the adversary's objective reward function, and the discount factor be $\gamma \in (0,1)$. Assume that there exists an upper bound $\bar{R}_{tgt} \in \mathbb{R}$ for $R_{tgt}$. Then:*

$$\left( \frac{1}{\sqrt{2}\bar{R}_{tgt}} \left( \mathbb{E}^M_{\pi \circ \nu}[R_{tgt}(s,a)] - \mathbb{E}^M_{\pi}[R_{tgt}(s,a)] \right) \right)^2 \leq \sum_{t=0}^{\infty} \frac{\gamma^t}{1-\gamma} \mathbb{E}_{s \sim d^t_{\pi}}[D_{KL}(\pi(\cdot|s)||\pi \circ \nu(\cdot|s))],$$
$$\tag{11}$$

*where $d^t_{\pi}(s) = \Pr(s_t = s|\pi)$ represents the state distribution of $\pi$ at time $t$.*

The proof is provided in Appendix B.2. Theorem 6.1 implies that the smaller the sensitivity of the policy's action outputs to state changes, the smaller the increase in cumulative reward under attack. Thus, policy smoothing is an effective defense not only against reward-minimization attacks but also against behavior-targeted attacks. Furthermore, the theorem reveals a key insight unique to defending against behavior-targeted attacks: the sensitivity of action outputs in the early stages of trajectories has a greater influence on the victim's overall defense performance. Therefore, reducing early-stage action sensitivity further strengthens robustness against behavior-targeted attacks.

Based on Theorem 6.1, we propose a robust training framework, Time-Discounted Robust Training (TDRT). Let $B = \{(s_t, t)\}_{t=1}^N$ be a mini-batch of state-time pairs, where $t$ is the timestep at which state $s_t$ was observed. The defender's objective is redefined as:

$$J_{\text{def}}(\pi) = -J_{\text{RL}}(\pi) + \lambda \max_{\nu} \sum_{s_t \in B} \gamma^t D_{\text{KL}}(\pi(\cdot|s_t)||\pi \circ \nu(\cdot|s_t)). \tag{12}$$

The complete training procedure, implemented with PPO, is given in Algorithm 2. In our implementation, the timestep $t$ enters only through the discount factor $\gamma^t$ and is not included in the policy input. Furthermore, since direct calculation of the KL term is computationally expensive, we apply convex relaxation methods to obtain tight upper bounds (see Appendix E in details).

The primary advantage of time-discounting is to suppress performance degradation on the victim's original task. Existing uniform smoothing methods tend to overly restrict the policy's expressiveness in the later stages of a trajectory to achieve sufficient robustness. In contrast, TDRT concentrates the effect of regularization at the early stages by applying time-discounting. Consequently, TDRT achieves sufficient robustness while preserving the policy's expressive capacity, thereby maintaining high performance on the victim's original task.

# 7 EXPERIMENTS

We empirically evaluate our proposed method on Meta-World (Yu et al., 2020) (continuous action spaces), MuJoCo (Todorov et al., 2012) (continuous action spaces), and MiniGrid (Chevalier-Boisvert et al., 2018) (discrete action spaces, vision-based control). Due to space limitations, this section only reports results for the Meta-World environment. Other experiments can be found in Appendix F.

We focus on three key aspects: **(i) Attack performance:** Can the adversarial policy learned through BIA effectively manipulate the victim's behavior? **(ii) Robustness:** Does the policy smoothing in TDRT offer greater robustness against behavior-targeted attacks compared to existing defense methods such as adversarial training? **(iii) Original task performance:** Can the time-discounting component of TDRT mitigate performance degradation on the original tasks compared to traditional policy smoothing without time discounting?

**Set up.** As illustrated in the scenarios in Section 1, we suppose an adversary aims to force a victim to perform an adversary's task that is entirely different from the victim's original task. Accordingly, we set the adversary's objective to force the victim to execute an adversary's task that is opposite to the one the victim originally learned. In the following, we refer to the reward function in the adversary's target task as *the adversary's reward function* and the reward function in the victim's original task as *the victim's reward function*.

We consider five opposing task pairs: {window-close, window-open}, {drawer-close, drawer-open}, {faucet-close, faucet-open}, {handle-press-side, handle-pull-side}, and {door-lock, door-unlock}. For example, if the victim originally learned the window-close task, the target policy is defined as a policy that completes the window-open task. Since the reward structures of the Meta-World benchmark are relatively complex, the adversary's reward function (e.g., reward function of window-open) cannot be obtained by simply negating the victim's reward function (e.g., reward function of window-close). In this sense, forcing a victim agent trained to perform window-close to execute window-open cannot be achieved through a reward-minimization attack but requires a behavior-targeted attack.

Following the prior works (Zhang et al., 2021; Sun et al., 2022), we constrain the set of adversarial states $\mathcal{B}(s)$ using the $L_\infty$ norm: $\mathcal{B}(s) \triangleq \{\hat{s} \mid \|\hat{s} - s\|_\infty \leq \epsilon\}$, where $\epsilon$ represents the attack budget. All experiments are performed with $\epsilon = 0.3$. See G.2 for the results of changing the attack budgets. States are standardized across all tasks, with standardization coefficients calculated during the training of the victim agent. To learn adversarial policies with BIA, we employed DAC (Kostrikov et al., 2018) of ILfD and OPOLO (Zhu et al., 2020) for ILfO as IL algorithms, which are variants of GAIL. In BIA, the demonstrations consist of 20 episodes of trajectories generated by the target policy, which is fully trained on the adversary's target task. We also vary the number of demonstrations in Appendix G.1. The results show that performance changes little even when the number of episodes is reduced to four. Appendix G.3 also provides a comparison of attack performance across different target policies.

**Attack Baselines.** We compare our proposed attack method with three baselines. **(i) Random Attack:** This attack perturbs the victim's state observation by random noise drawn from a uniform distribution. This attack works with no-box access and requires no knowledge about the victim. **(ii) Targeted PGD Attack:** This naive attack method optimizes falsified states $\hat{s}$ using PGD at each time independently to align the victim's actions with those of the target policy's at each time: $\hat{s} = \arg\min_{\hat{s}} d(\pi(\cdot|\hat{s}), \pi_{\text{tgt}}(\cdot|s))$. *PGD requires white-box access to the victim's policy, giving the adversary an advantage not available in our proposed method.* The detailed explanation and analysis of Targeted PGD are provided in Appendix I.4.2. **(iii) Target Reward Maximization Attack:** This attack leverages Lemma B.1 to learn an adversarial policy that maximizes the cumulative reward obtained by the victim from the adversary's reward function $R_{\text{adv}}$: $\nu^* = \arg\max_{\pi\circ\nu}^M [R_{\text{adv}}(s, a, s')]$. We used two methods for training the adversarial policy: SA-RL(Zhang et al., 2021) (black-box attack) and PA-AD(Sun et al., 2022) (white-box attack). These attacks require access to the adversary's reward function. *Thus, they give the adversary an advantage not available in our proposed method.*

**Defense Baselines.** Defense methods against behavior-targeted attacks have not yet been proposed. Therefore, as baselines, we employ defense methods against untargeted attacks: ATLA-PPO(Zhang et al., 2021), PA-ATLA-PPO(Sun et al., 2022), RAD-PPO(Belaire et al., 2024), WocaR-PPO(Liang et al., 2022), and SA-PPO(Zhang et al., 2020b). ATLA-PPO and PA-ATLA-PPO are methods

Table 1: Comparison of attack performances. Each value represents the average episode reward ± standard deviation over 50 episodes. Each parenthesis indicates (*Access model*, *Adversary's knowledge*). **Under the limited-knowledge setting, BIA's attack performance is competitive with that of baseline methods that assume greater adversary knowledge.**

| Adv Task | Target Reward | Attack Rewards (↑) | | | | | |
|---|---|---|---|---|---|---|---|
| | | Adversary with full knowledge ← | | | | → Adversary with limited knowledge | |
| | | Targeted PGD (white-box, target policy) | Rew Max (PA-AD) (white-box, reward function) | Rew Max (SA-RL) (black-box, reward function) | BIA-ILfD (ours) (black-box, demonstrations) | BIA-ILfO (ours) (no-box, demonstrations) | Random (no-box, no knowledge) |
| window-close | 4543 ± 39 | 1666 ± 936 | 4255 ± 300 | **4505 ± 65** | 3962 ± 666 | 4036 ± 510 | 947 ± 529 |
| window-open | 4508 ± 121 | 515 ± 651 | 493 ± 562 | 506 ± 444 | **566 ± 523** | 557 ± 679 | 322 ± 261 |
| drawer-close | 4868 ± 6 | 2891 ± 150 | 3768 ± 1733 | 4658 ± 747 | **4760 ± 640** | 4626 ± 791 | 1069 ± 1585 |
| drawer-open | 4713 ± 16 | 953 ± 450 | **1607 ± 355** | 1499 ± 536 | 1556 ± 607 | 1445 ± 610 | 841 ± 357 |
| faucet-close | 4754 ± 15 | 1092 ± 192 | 1241 ± 501 | **3409 ± 652** | 3316 ± 648 | 3041 ± 502 | 897 ± 171 |
| faucet-open | 4544 ± 800 | 2541 ± 86 | 1420 ± 85 | 1448 ± 64 | **3031 ± 1493** | 2718 ± 1293 | 1372 ± 81 |
| handle-press-side | 4546 ± 721 | 1994 ± 1225 | **4726 ± 175** | 4625 ± 175 | 4631 ± 408 | 4627 ± 586 | 1865 ± 1340 |
| handle-pull-side | 4442 ± 732 | 2198 ± 1524 | 2065 ± 1501 | 3617 ± 1363 | **4268 ± 740** | 4193 ± 517 | 1426 ± 1617 |
| door-lock | 3845 ± 79 | 640 ± 664 | 763 ± 768 | 1937 ± 1186 | **2043 ± 1229** | 1906 ± 1045 | 589 ± 494 |
| door-unlock | 4690 ± 33 | 531 ± 61 | 3295 ± 1111 | **3421 ± 974** | 3336 ± 932 | 3123 ± 1123 | 391 ± 59 |

that utilize adversarial training. RAD-PPO is a regret-based defense method that learns policies to minimize regret, defined as the difference between the rewards under non-attack and attack conditions. WocaR-PPO is a defense method that learns policies to maximize the worst-case rewards and applies regularization to the smoothness of policies, specifically in critical states where rewards significantly decrease. SA-PPO aims to increase the smoothness of the policy's action outputs by a regularizer. *The difference between TDRT-PPO and SA-PPO is that SA-PPO does not apply time discounting in the regularization.* For detailed explanations of the baselines, see Appendix I.2.

**Attack Performance Comparison.** We present the results in Table 1. The *attack rewards* represent the cumulative reward of the adversary's task obtained by the victim under attacks. The *target rewards* are the cumulative reward obtained by the target policy, which is directly trained with the adversary's reward function, serving as the upper bound for attack rewards. We also report attack success rates in Table 10 by using the task-success criterion in Meta-World. These results exhibit the same trend.

BIA achieves an attack performance comparable to more advantaged attacks, such as Target Reward Maximization (Rew Max/SA-RL, PA-AD), which has access to the adversary's reward function. This demonstrates that BIA can effectively attack using demonstrations alone, without requiring any reward modeling of the adversary's task. The attack performance of BIA-ILfO is nearly identical to that of BIA-ILfD. We attribute this to the deterministic nature of state transitions in the Meta-World tasks: without access to the victim's actions, the transitions can be sufficiently predicted. While we used 20 episodes as demonstrations for BIA, results in Appendix G.1 confirm that the attack performance remains nearly unchanged even when the demonstrations are reduced to four episodes.

The targeted PGD attack achieves significantly lower attack rewards than BIA across all tasks. This is because targeted PGD optimizes adversarial perturbations independently at each time step without accounting for future decisions. Consequently, under a limited $\epsilon$, targeted PGD fails to achieve sufficient attack effectiveness. Our analysis revealed that, even after PGD optimization, the loss remains high at multiple states along a trajectory.

All attack methods exhibit relatively low attack performance on the window-open, drawer-open, and door-lock tasks. This can be attributed to the greater disparity in the state-action distributions between the victim's policy and the target policy in these tasks compared to others. In such cases, their behaviors differ so significantly that orchestrating a successful attack becomes extremely difficult.

**Defense Performance Comparison.** The results appear in Table 2. The *best attack rewards* refer to the largest attack reward obtained by the victim among the six attacks listed in Table 1; lower values indicate greater robustness. Complete results for all attack methods are reported in Table 12.

In all tasks, TDRT-PPO and SA-PPO achieve superior robustness against attacks, indicating that policy smoothing is effective against behavior-targeted attacks. For the drawer-close task, SA-PPO attains exceptionally high robustness, albeit with a substantial drop in original task performance (see Table 3). Additional experiments in Appendix H.2 examine how varying the smoothing coefficient trades off robustness and original task performance.

Adversarial training methods, including ATLA-PPO and PA-ATLA-PPO, are still vulnerable to behavior-targeted attacks in most tasks. As noted by (Korkmaz, 2021; 2023), adversarial training is

Table 2: Comparison of robustness. Each value represents the average episode reward $\pm$ standard deviation over 50 episodes. **Policy smoothing is very effective against behavior-targeted attacks.**

| | | | | | | | |
|---|---|---|---|---|---|---|---|
| | | | | **Best Attack Reward ($\downarrow$)** | | | |
| **Task** | **PPO** (No defense) | **ATLA-PPO** (AdvTraining) | **PA-ATLA-PPO** (AdvTraining) | **RAD-PPO** (Regret) | **WocaR-PPO** (Partial smoothing) | **SA-PPO** Smoothing (w/o time-discounting) | **TDRT-PPO (ours)** Smoothing (w/ time-discounting) |
| window-close | $4505 \pm 65$ | $4270 \pm 188$ | $4041 \pm 96$ | $4261 \pm 208$ | $575 \pm 135$ | $485 \pm 61$ | $\mathbf{482 \pm 3}$ |
| window-open | $566 \pm 523$ | $586 \pm 649$ | $671 \pm 589$ | $501 \pm 132$ | $295 \pm 18$ | $272 \pm 37$ | $\mathbf{254 \pm 214}$ |
| drawer-close | $4760 \pm 640$ | $4858 \pm 6$ | $4868 \pm 3$ | $4588 \pm 923$ | $4867 \pm 8$ | $\mathbf{4 \pm 2}$ | $4860 \pm 4$ |
| drawer-open | $1556 \pm 607$ | $1158 \pm 1026$ | $954 \pm 219$ | $736 \pm 22$ | $579 \pm 15$ | $403 \pm 49$ | $\mathbf{378 \pm 10}$ |
| faucet-close | $3409 \pm 652$ | $4108 \pm 790$ | $4012 \pm 123$ | $2235 \pm 528$ | $2829 \pm 1264$ | $\mathbf{1559 \pm 406}$ | $1789 \pm 610$ |
| faucet-open | $3031 \pm 1493$ | $4383 \pm 449$ | $2358 \pm 976$ | $4254 \pm 625$ | $3012 \pm 1301$ | $\mathbf{1763 \pm 255}$ | $1942 \pm 261$ |
| handle-press-side | $4726 \pm 175$ | $4302 \pm 799$ | $3318 \pm 1539$ | $2375 \pm 1440$ | $3042 \pm 1193$ | $\mathbf{1888 \pm 1169}$ | $1928 \pm 736$ |
| handle-pull-side | $4268 \pm 740$ | $532 \pm 534$ | $512 \pm 982$ | $1086 \pm 1256$ | $33 \pm 6$ | $10 \pm 1$ | $\mathbf{7 \pm 1}$ |
| door-lock | $2043 \pm 1229$ | $1020 \pm 805$ | $992 \pm 19$ | $712 \pm 392$ | $562 \pm 14$ | $\mathbf{478 \pm 7}$ | $487 \pm 11$ |
| door-unlock | $3421 \pm 974$ | $3277 \pm 1265$ | $2806 \pm 1437$ | $2743 \pm 1386$ | $1073 \pm 161$ | $787 \pm 1001$ | $\mathbf{691 \pm 356}$ |

effective only against the reward-minimization attack anticipated during training. Thus, it remains vulnerable to behavior-targeted attacks. Although the regret-based approach RAD-PPO demonstrates relatively higher robustness in some tasks than adversarial training, it still relies on reward-based metrics and, therefore, does not offer sufficient protection against behavior-targeted attacks. WocaR-PPO, which applies smoothing to a subset of states, achieves moderate robustness but underperforms SA-PPO and TDRT-PPO, both of which uniformly smooth across the entire state space. Further experiments on robust training efficiency are presented in Appendix H.3.

**Original-Task Performance Comparison.**
The results appear in Table 3. The *clean rewards* represent the reward obtained by the victim in its original task without attacks; higher values indicate less performance degradation due to robust training.

TDRT-PPO achieves higher clean rewards than SA-PPO in all tasks. Together with the results in Table 2, this demonstrates that while uniform regularization across the entire trajectory in SA-PPO sacrifices original performance, time discounting in TDRT-PPO mitigates performance degradation while preserving robustness. In Appendix H.2, we further study the effect of the regularization coefficient. Across a wide range of coefficients, TDRT-PPO consistently achieves higher clean rewards than SA-PPO.

Table 3: Clean rewards comparison: TDRT-PPO vs. SA-PPO. **Time-discounting greatly improves the performance on the original task.**

| | | |
|---|---|---|
| | **Clean Reward ($\uparrow$)** | |
| **Task** | **SA-PPO** (w/o time-discounting) | **TDRT-PPO (ours)** (w/ time-discounting) |
| window-close | $4367 \pm 107$ | $\mathbf{4412 \pm 55}\,(\uparrow 1.0\%)$ |
| window-open | $4092 \pm 461$ | $\mathbf{4383 \pm 57}\,(\uparrow 7.1\%)$ |
| drawer-close | $2156 \pm 453$ | $\mathbf{4237 \pm 93}\,(\uparrow 96.5\%)$ |
| drawer-open | $4161 \pm 1537$ | $\mathbf{4802 \pm 27}\,(\uparrow 15.4\%)$ |
| faucet-close | $4304 \pm 42$ | $\mathbf{4740 \pm 17}\,(\uparrow 10.1\%)$ |
| faucet-open | $4380 \pm 43$ | $\mathbf{4630 \pm 11}\,(\uparrow 5.7\%)$ |
| handle-press-side | $3226 \pm 806$ | $\mathbf{4321 \pm 215}\,(\uparrow 33.9\%)$ |
| handle-pull-side | $4094 \pm 350$ | $\mathbf{4468 \pm 126}\,(\uparrow 9.1\%)$ |
| door-lock | $2299 \pm 1491$ | $\mathbf{2769 \pm 1411}\,(\uparrow 20.4\%)$ |
| door-unlock | $2017 \pm 497$ | $\mathbf{3680 \pm 290}\,(\uparrow 82.5\%)$ |
| avg. improvement | | **28.2%** |

## 8    CONCLUSION AND LIMITATIONS

This work introduces the Behavior Imitation Attack (BIA), which manipulates victim behavior through perturbed state observations under limited access to the victim's policy. We also introduced Time-Discounted Robust Training (TDRT), the first defense method specifically designed for behavior-targeted attacks, which achieved robustness without compromising original performance.

**Limitation.**    While our work contributes valuable insights, it also has limitations. Adversarial policy is known to be less effective in high-dimensional state spaces, such as image inputs (Sun et al., 2022) (see Appendix F.2). Efforts to overcome this require white-box access (Sun et al., 2022) and remain unresolved. Moreover, although TDRT empirically exhibits robustness against behavior-targeted attacks, it lacks certified guarantees. In scenarios where higher reliability requirements are imposed, TDRT may prove insufficient.

## 9 ACKNOWLEDGEMENTS

This work is partially supported by JST JPMJNX25C2, JPMJKP24C3, JPMJCR23M4, JPMJCR21D3, JSPS 23H00483, and 120251002. We gratefully acknowledge the insightful comments and suggestions provided by the anonymous reviewers.

## 10 ETHICS STATEMENTS

This paper focused on adversarial attacks on reinforcement learning and their defense methods, aiming to improve the reliability of deep reinforcement learning. Our contribution lies in proposing attack methods and defenses that are envisioned for real-world scenarios. However, our proposed methods still face challenges in practical real-world applications. Our research fully complies with legal and ethical standards, and there are no conflicts of interest. Throughout this study, we utilized only publicly available benchmarks. No private datasets were used in this research.

We have provided detailed algorithms for our proposed methods. The benchmarks used in our experiments are public and accessible to everyone. As described above, we have made substantial efforts to ensure the reproducibility of our results.

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

# A    RELATED WORKS

In this section, we discuss prior works on attacks and defenses on DRL, which primarily consider an adversary that perturbs the victim's state observations.

## A.1    ATTACK METHODS

We classify existing attack methods on fixed reinforcement learning agents into three categories: the reward-minimization attack, the enchanting attack, and the behavior-targeted attack. Furthermore, we discuss the poisoning attacks that occur during the victim's training phase.

**Reward-minimization attack.**    In reward-minimization attacks, the adversary's goal is to minimize the cumulative reward received by the victim. The most basic approaches are gradient-based methods. Huang et al. (2017) proposed an attack method that uses the Fast Gradient Signed Method (FGSM) (Goodfellow et al., 2014) to compute adversarial perturbations that prevent the victim from choosing optimal actions. Pattanaik et al. (2018) performed a more powerful attack by creating adversarial perturbations to minimize the value estimated by the Q-function. Gleave et al. (2020) proposed an attack that assumes a two-agent environment. This method creates an adversarial agent that severely degrades the victim's performance. Sun et al. (2020) point out that previous works lack stealth and proposed the Critical Point Attack and Antagonist Attack, which achieve effective attacks within very few steps. Qiaoben et al. (2024) classify existing adversarial attacks against RL agents in the function space and propose an attack method based on a two-stage optimization derived from the theoretical analyses. Sun et al. (2020) and Qiaoben et al. (2024) are similar to ours in that they attempt to bring the victim's policy closer to a specific policy. However, these objectives differ from ours, and they are implemented via a PGD attack that requires white-box access.Duan et al. (2025) proposed an attack that targets the distribution of the victim's policy. However, their method is a reward-minimization attack and differs from ours. Specifically, they intended to induce large deviations from the original victim's distribution and indirectly reduce the victim's cumulative reward. Because their goal is not to shift the policy toward any specific target distribution, their attack differs from the behavior-targeted attack.

**Enchanting attack.**    The enchanting attack aims to lure the victim agent to reach a predetermined target state. Lin et al. (2017) first proposed this type of attack. In their approach, the adversary generates a sequence of states and actions that cause the victim to reach the target state. The adversary then crafts a sequence of perturbations to make the victim perform the sequence of actions. Tretschk et al. (2018) proposed an enchanting attack where adversarial perturbations to maximize adversarial rewards are heuristically designed for the attack purpose, thereby leading the victim to the specified target state. Buddareddygari et al. (2022) proposed a different enchanting attack using visual patterns placed on physical objects in the environment so that the victim agent is directed to the target state. Unlike (Lin et al., 2017) and (Tretschk et al., 2018), which perturb the victim's state observations, this attack alters the environmental dynamics. Ying et al. (2023) proposed an enchanting attack with a universal adversarial perturbation. When any state observations are modified with this perturbation, the victim agent is forced to be guided to the target state. All of these enchanting attacks require white-box access to the victim's policy.

**Behavior-targeted attack.**    The behavior-targeted attack aims to manipulate not only the final destination but also the victim's behavior in a more detailed manner. Hussenot et al. (2020) proposed a behavior-targeted attack that forces the victim to select the same actions as the policy specified by the adversary. More specifically, this attack precomputes a universal perturbation for each action so that the victim who observes the perturbed states takes the same action as the adversarially specified policy. One limitation of this attack is that the computational cost of precomputing such universal perturbations can be high when the action space is large or continuous. Since the cost required for pre-computing perturbations is significant, applying this attack is challenging. Boloor et al. (2020) proposed a heuristic attack specifically designed for autonomous vehicles. They formulated an objective function with a detailed knowledge of the target task, which is not applicable to behavior-targeted attacks for general tasks. Bai et al. (2025; 2024) investigated an attack to manipulate the victim's behavior that follows the adversary's preference for the behavior. This attack method can be applied to any environment. However, one limitation of this method is that it usually requires

thousands of labeled preference state-action sequences to specify the behavior that the adversary requests to follow. Therefore, this attack is not practical. We remark that all existing behavior-targeted attacks also require white-box access to the victim's policy.

**Poisoning attack.** Several studies have proposed poisoning attacks against RL agents, which intervene during the victim's training phase. Kiourti et al. (2020) introduced TrojDRL, a framework for evaluating backdoor attacks in DRL, where the adversary intercepts and manipulates states during training to steer the agent toward a predefined target policy. Sun et al. (2021) proposed VA2C-P that adapts without access to environment dynamics and supports both untargeted attacks and targeted attacks that align the victim's policy with a target policy. Rakhsha et al. (2020) theoretically analyzed poisoning attacks in both offline planning and online RL with tabular MDPs, while Rangi et al. (2022) further clarified the fundamental limits of poisoning attacks in episodic reinforcement learning. Other works (Xu et al., 2023; Xu & Singh, 2023; Li et al., 2024b) developed more practical black-box poisoning attacks that do not require knowledge of the environment dynamics or the victim's learning algorithm. More recently, Cui et al. (2024) proposed BadRL, a sparse poisoning attack that dynamically generates sample-specific triggers and injects them only at high attack-value states. All of these methods aim to steer the victim's policy toward an attacker-specified target policy. However, our threat model explicitly prohibits the adversary from accessing or intervening in the victim's training process, whereas poisoning attacks inherently rely on such access. Consequently, these attack settings fall outside the scope of our problem formulation.

## A.2 DEFENSE METHODS

One approach to learning robust policies against reward-minimization attacks is policy smoothing. Shen et al. (2020) introduced a regularization term to smooth the policy and demonstrated increased sample efficiency and robustness. Zhang et al. (2020b) formulated the State-Adversarial Markov Decision Process (SA-MDP) to represent situations where an adversary interferes with the victim's state observations. Based on SA-MDP, they showed that regularization to smooth the policy is effective in resisting reward-minimization attacks. However, their theoretical robustness guarantees are limited to reward-minimization attacks and not directly extended to behavior-targeted attacks. Furthermore, they often degrade performance on the victim's original tasks, as it imposes excessive constraints on the policy's representational capacity.

Another approach for defense is adversarial training. Zhang et al. (2021) showed that the optimal adversarial policy can be learned as a policy in MDP and proposed ATLA, an adversarial training framework that exploits this insight. Sun et al. (2022) proposed a theoretically optimal attack method that finds the optimal direction of perturbation and proposed PA-ATLA as an extension of ATLA, which is efficient even in large state spaces. Oikarinen et al. (2021) proposed a framework for training a robust RL by incorporating an adversarial loss that accounts for the worst-case input perturbations. Additionally, they introduced a new metric to efficiently evaluate the robustness of the victim. Liang et al. (2022) efficiently estimated the lower bound of cumulative rewards under adversarial attacks and performed adversarial learning with partial smoothness regularization. Li et al. (2024a) theoretically motivate minimizing the Bellman infinity-error in state-adversarial MDPs and propose CAR-DQN as a robust value-based algorithm under reward-minimization attacks. However, Korkmaz (2021; 2023) pointed out that the victims trained by adversarial training are still vulnerable to attacks that were not anticipated during training.

McMahan et al. (2024) proposed a comprehensive framework for computing optimal attacks and defenses, modeling the attack problem as a meta-MDP and the defense problem as a partially observable turn-based stochastic game. Bukharin et al. (2024) proposed a robust Multi-Agent RL (MARL) framework that uses adversarial regularization to promote Lipschitz continuity of policies, thereby enhancing robustness against environmental changes, observation noise, and malicious agent actions. Liang et al. (2024) introduced a new concept called temporally-coupled perturbations, where consecutive perturbations are constrained. Their proposed method, GRAD, demonstrates strong robustness against standard and temporally-coupled perturbations. Some works (Jin et al., 2018; Rigter et al., 2021; Belaire et al., 2024) proposed a novel defense method based on regret minimization. Additionally, another approach in robust reinforcement learning is certified defense (Wu et al., 2022; Kumar et al., 2022; Mu et al., 2024; Sun et al., 2024; Wang et al., 2025). These studies guarantee a lower bound on the rewards obtained by the victim under adversarial attacks. Liu

et al. (2024) proposed an adaptive defense method robust against multiple types of attacks, not limited to worst-case reward-minimization. In their method, the defender prepares multiple robust policies in advance and then selects the policy that maximizes the victim's reward under attack based on rewards obtained in previous episodes. However, our approach focuses on a single-agent RL problem, as the adversary usually targets a single static victim in a stationary environment. As a result, their adaptive setting differs from ours.

Recent works (Sun & Zheng, 2024; YANG & Xu, 2024) have proposed defense methods based on diffusion models. These methods use a diffusion model to denoise perturbed states and recover the original clean state as much as possible. A main advantage is that, once the diffusion model is trained, the same model can be applied to many different victim policies without retraining. However, these approaches require extra computation at inference time, because the agent must run the diffusion process before choosing an action. Our regularization method is complementary to these diffusion-based defenses and can be combined with them.

# B  PROOFS

## B.1  PROOF OF THEOREM 5.1

**Theorem 5.1.** *Consider an SA-MDP $M = (\mathcal{S}, \mathcal{A}, R, \mathcal{B}, p, \gamma)$ with adversarial policy $\nu$. Let $\pi$ denote the victim's policy and $\pi_{tgt}$ the target policy. Assume that the divergence $\mathcal{D}$ admits the following variational representation:*

$$\mathcal{D}(\pi \circ \nu, \pi_{tgt}) = \max_{d}\Big[\mathbb{E}_{\pi \circ \nu}[g(d(s, a, s'))] + \mathbb{E}_{\pi_{tgt}}[-f(d(s, a, s'))]\Big], \tag{3}$$

*where $f$ and $g$ are arbitrary convex and concave functions, respectively, and $d : \mathcal{S} \times \mathcal{A} \times \mathcal{S} \to \mathbb{R}$ is a discriminator. Let $d_\star$ be the optimal discriminator in equation 3. Under this assumption, define the reward function $\hat{R}_d$ and the state transition probability $\hat{p}$ as follows:*

$$\hat{R}_d(s, \hat{s}, s') = \begin{cases} -\dfrac{\sum_{a \in \mathcal{A}} \pi(a|\hat{s})p(s'|s,a)g(d_\star(s,a,s'))}{\sum_{a \in \mathcal{A}} \pi(a|\hat{s})p(s'|s,a)} & \textit{if } \hat{s} \in \mathcal{B}(s) \\ C & \textit{otherwise,} \end{cases} \tag{4}$$

$$\hat{p}(s'|s, \hat{s}) = \sum_{a \in \mathcal{A}} \pi(a|\hat{s})p(s'|s, a), \tag{5}$$

*where $C$ is a large negative constant. Then, for the MDP $\hat{M} = (\mathcal{S}, \mathcal{S}, \hat{R}_d, \hat{p}, \gamma)$, it holds that:*

$$\arg\min_{\nu} \mathcal{D}(\pi \circ \nu, \pi_{tgt}) = \arg\max_{\nu} \mathbb{E}_{\nu}^{\hat{M}}\big[\hat{R}_d(s, \hat{s}, s')\big]. \tag{6}$$

*Proof.* Before proving this theorem, we first establish the following lemma:

**Lemma B.1.** *Consider an SA-MDP $M = (\mathcal{S}, \mathcal{A}, R, \mathcal{B}, p, \gamma)$, and let $\pi$ be the victim's policy. Given the reward function $\bar{R}$ specified by the adversary, the reward function $\hat{R}$ and state transition probability $\hat{p}$ are defined as follows:*

$$\hat{R}(s, \hat{s}, s') = \mathbb{E}\left[\hat{r} \mid s, \hat{s}, s'\right] = \begin{cases} \dfrac{\sum_{a \in \mathcal{A}} \pi(a|\hat{s})p(s'|s,a)\bar{R}(s,a,s')}{\sum_{a \in \mathcal{A}} \pi(a|\hat{s})p(s'|s,a)} & \textit{if } \hat{s} \in \mathcal{B}(s) \\ C & \textit{otherwise,} \end{cases} \tag{13}$$

$$\hat{p}(s'|s, \hat{s}) = \sum_{a \in \mathcal{A}} \pi(a|\hat{s})p(s'|s, a). \tag{14}$$

*where $C$ is a large negative constant. Then, for the MDP $\hat{M} = (\mathcal{S}, \mathcal{S}, \hat{R}, \hat{p}, \gamma)$, the following equality holds:*

$$\arg\max_{\nu} \mathbb{E}_{\pi \circ \nu}^{M}[\bar{R}(s, a, s')] = \arg\max_{\nu} \mathbb{E}_{\nu}^{\hat{M}}[\hat{R}(s, \hat{s}, s')]. \tag{15}$$

*Proof.* This proof follows the approach outlined in the proof of Lemma 1 in (Zhang et al., 2020b), with some modifications to account for the differences in our setting.

In the proof of Lemma 1 presented in (Zhang et al., 2020b), by substituting $-R(s, a, s')$ with $\bar{R}(s, a, s')$, we can derive the subsequent results:

$$\hat{R}(s, \hat{a}, s') = \frac{\sum_a \bar{R}(s, a, s')p(s'|a, s)\pi(a|\hat{a})}{\sum_a p(s'|a, s)\pi(a|\hat{a})}. \tag{16}$$

Let $\overline{M} = \max_{s,a,s'} \bar{R}(s, a, s')$ and $\underline{M} = \min_{s,a,s'} \bar{R}(s, a, s')$. We Define the reward $C$ for when the adversarial policy selects an action $\hat{a} \notin \mathcal{B}$ as follows:

$$C < \min \left\{ \underline{M}, \frac{1}{1 - \gamma}\underline{M} - \frac{\gamma}{1 - \gamma}\overline{M} \right\}. \tag{17}$$

From the definition of $C$ and $\overline{M}$, we have for $\forall(s, \hat{a}, s')$,

$$C < \hat{R}(s, \hat{a}, s') \leq \overline{M}, \tag{18}$$

and, for $\forall \hat{a} \in \mathcal{B}(s)$, according to equation 16,

$$\underline{M} \leq \hat{R}(s, \hat{a}, s') \leq \overline{M}. \tag{19}$$

MDP has at least one optimal policy, so the $\hat{M}$ has an optimal adversarial policy $\nu^*$, which satisfies $\hat{V}_{\pi \circ \nu^*}(s) \geq \hat{V}_{\pi \circ \nu}(s)$ for $\forall s, \forall \nu$. From the property of the optimal policy, $\nu^*$ is deterministic. Let $\mathfrak{R} \triangleq \{\nu \mid \forall s, \exists \hat{a} \in \mathcal{B}(s), \nu(\hat{a} \mid s) = 1\}$. This restricts that the adversarial policy does not take actions $\hat{a} \notin B(s)$, so $\nu^* \in \mathfrak{R}$. If $\nu^* \notin \mathfrak{R}$ at state $s^0$,

$$\hat{V}_{\pi \circ \nu^*}(s^0) = \mathbb{E}_{\hat{p}, \nu^*} \left[ \sum_{k=0}^{\infty} \gamma^k \hat{r}_{t+k+1} \mid s_t = s^0 \right] \tag{20}$$

$$= C + \mathbb{E}_{\hat{p}, \nu^*} \left[ \sum_{k=1}^{\infty} \gamma^k \hat{r}_{t+k+1} \mid s_t = s^0 \right] \tag{21}$$

$$\leq C + \frac{\gamma}{1 - \gamma}\overline{M} \tag{22}$$

$$< \frac{1}{1 - \gamma}\overline{M} \tag{23}$$

$$\leq \mathbb{E}_{\hat{p}, \nu'} \left[ \sum_{k=0}^{\infty} \gamma^k \hat{r}_{t+k+1} \mid s_t = s^0 \right] = \hat{V}_{\pi \circ \nu'}(s^0). \tag{24}$$

The last inequality holds for any $\nu' \in \mathfrak{R}$. This contradicts the assumption that $\nu^*$ is optimal. Hence, the following analysis will only consider policies included in $\mathcal{N}$.

For any policy $\nu \in \mathfrak{R}$:

$$\hat{V}_{\pi \circ \nu}(s) = \mathbb{E}_{\hat{p}, \nu} \left[ \sum_{k=0}^{\infty} \gamma^k \hat{r}_{t+k+1} \mid s_t = s \right] \tag{25}$$

$$= \mathbb{E}_{\hat{p}, \nu} \left[ \hat{r}_{t+1} + \gamma \sum_{k=0}^{\infty} \gamma^k \hat{r}_{t+k+2} \mid s_t = s \right] \tag{26}$$

$$= \sum_{\hat{a} \in \mathcal{S}} \nu(\hat{a}|s) \sum_{s' \in \mathcal{S}} \hat{p}(s'|s, \hat{a}) \left[ \hat{R}(s, \hat{a}, s') + \gamma \mathbb{E}_{\hat{p}, \nu} \left[ \sum_{k=0}^{\infty} \gamma^k \hat{r}_{t+k+2} \mid s_{t+1} = s' \right] \right] \tag{27}$$

$$= \sum_{\hat{a} \in \mathcal{S}} \nu(\hat{a}|s) \sum_{s' \in \mathcal{S}} \hat{p}(s'|s, \hat{a}) \left[ \hat{R}(s, \hat{a}, s') + \gamma \hat{V}_{\pi \circ \nu}(s') \right]. \tag{28}$$

All policies in $\mathfrak{R}$ are deterministic, so we denote the deterministic action $\hat{a}$ chosen by a $\nu \in \mathfrak{R}$ at $s$ as $\nu(s)$. Then for $\forall \nu \in \mathfrak{R}$, we have

$$\hat{V}_{\pi \circ \nu}(s) = \sum_{s' \in \mathcal{S}} \hat{p}(s'|s, \hat{a}) \left[ \hat{R}(s, \hat{a}, s') + \gamma \hat{V}_{\pi \circ \nu}(s') \right] \tag{29}$$

$$= \sum_{s' \in \mathcal{S}} \sum_{a \in \mathcal{A}} \pi(a|\hat{a})p(s'|s, a) \left[ \frac{\sum_a \bar{R}(s, a, s')p(s'|a, s)\pi(a|\hat{a})}{\sum_a p(s'|a, s)\pi(a|\hat{a})} + \gamma \hat{V}_{\pi \circ \nu}(s') \right] \tag{30}$$

$$= \sum_{a \in \mathcal{A}} \pi(a|\nu(s)) \sum_{s' \in \mathcal{S}} p(s'|s, a) \left[ \bar{R}(s, a, s') + \gamma \hat{V}_{\pi \circ \nu}(s') \right]. \tag{31}$$

Thus, the optimal value function is

$$\hat{V}_{\pi \circ \nu^*}(s) = \max_{\nu^*(s) \in \mathcal{B}(s)} \sum_{a \in \mathcal{A}} \pi(a|\nu^*(s)) \sum_{s' \in \mathcal{S}} p(s'|s,a) \left[ \bar{R}(s,a,s') + \gamma \hat{V}_{\pi \circ \nu^*}(s') \right]. \quad (32)$$

The Bellman equation for the state value function $\tilde{V}_{\pi \circ \nu}(s)$ of the SA-MDP $M = (\mathcal{S}, \mathcal{A}, \mathcal{B}, R, p, \gamma)$ is given as follows:

**Lemma B.2** (Theorem 1 of (Zhang et al., 2020b)). *Given* $\pi : \mathcal{S} \to \mathcal{P}(\mathcal{A})$ *and* $\nu : \mathcal{S} \to \mathcal{S}$, *we have*

$$\tilde{V}_{\pi \circ \nu}(s) = \sum_{a \in \mathcal{A}} \pi(a|\nu(s)) \sum_{s' \in \mathcal{S}} p(s'|s,a) \left[ R(s,a,s') + \gamma \tilde{V}_{\pi \circ \nu}(s') \right] \quad (33)$$

Therefore, if the reward function is $\bar{R}$, then $\hat{V}_{\pi \circ \nu^*} = \tilde{V}_{\pi \circ \nu^*}$. So, we have

$$\tilde{V}_{\pi \circ \nu^*}(s) = \max_{\nu^*(s) \in \mathcal{B}(s)} \sum_{a \in \mathcal{A}} \pi(a|\nu^*(s)) \sum_{s' \in \mathcal{S}} p(s'|s,a) \left[ \bar{R}(s,a,s') + \gamma \tilde{V}_{\pi \circ \nu^*}(s') \right], \quad (34)$$

and $\tilde{V}_{\pi \circ \nu^*}(s) \geq \tilde{V}_{\pi \circ \nu}(s)$ for $\forall s, \forall \nu \in \mathfrak{R}$. Hence, $\nu^*$ is also the optimal $\nu$ for $\tilde{V}_{\pi \circ \nu}$.

$\square$

This lemma shows that an optimal policy on the MDP $\hat{M}$ matches the optimal adversarial policy that maximizes the cumulative reward obtained from $\bar{R}$ specified by the adversary.

We now turn to the proof of the theorem. Under the assumption that the divergence admits a variational form, the adversary's objective can be expressed as follows:

$$\arg \min_{\nu} \max_{d} \left[ \mathbb{E}_{\pi \circ \nu}[g(d(s,a,s'))] + \mathbb{E}_{\pi_{\text{tgt}}}[-f(d(s,a,s'))] \right]. \quad (35)$$

Let $d_\star$ be the optimal discriminator. Focusing on optimizing the adversarial policy, the problem reduces to:

$$\arg \min_{\nu} \mathbb{E}_{\pi \circ \nu}[g(d_\star(s,a,s'))] = \arg \max_{\nu} \mathbb{E}_{\pi \circ \nu}[-g(d_\star(s,a,s'))]. \quad (36)$$

This is regarded as a cumulative reward maximization problem in which $-g(d_\star)$ serves as the reward function. Therefore, for the MDP $\hat{M} = (\mathcal{S}, \mathcal{S}, \hat{R}_d, \hat{p}, \gamma)$, the following result holds by Lemma B.1:

$$\arg \min_{\nu} \mathcal{D}(\pi \circ \nu, \pi_{\text{tgt}}) = \arg \max_{\nu} \mathbb{E}_{\nu}^{\hat{M}} \left[ \hat{R}_d(s, \hat{s}, s') \right]. \quad (37)$$

$\square$

## B.2 PROOF OF THEOREM 6.1

**Theorem 6.1.** *Let* $R_{tgt} : \mathcal{S} \times \mathcal{A} \to \mathbb{R}$ *be the adversary's objective reward function, and the discount factor be* $\gamma \in (0,1)$. *Assume that there exists an upper bound* $\bar{R}_{tgt} \in \mathbb{R}$ *for* $R_{tgt}$. *Then:*

$$\left( \frac{1}{\sqrt{2}\bar{R}_{tgt}} \left( \mathbb{E}_{\pi \circ \nu}^M[R_{tgt}(s,a)] - \mathbb{E}_{\pi}^M[R_{tgt}(s,a)] \right) \right)^2 \leq \sum_{t=0}^{\infty} \frac{\gamma^t}{1-\gamma} \mathbb{E}_{s \sim d_\pi^t}[D_{KL}(\pi(\cdot|s)||\pi \circ \nu(\cdot|s))],$$

$$(11)$$

*where* $d_\pi^t(s) = \Pr(s_t = s|\pi)$ *represents the state distribution of* $\pi$ *at time* $t$.

*Proof.* We begin by defining the state–action distribution. The state-action distribution $\rho_\pi : \mathcal{S} \times \mathcal{A} \to [0,1]$ is defined by the probability of encountering specific state-action pairs when transitioning according to a policy $\pi$:

$$\rho_\pi(s,a) = (1-\gamma) \sum_{t=0}^{\infty} \gamma^t \Pr\left(s_t = s, a_t = a \mid s_0 \sim p_0(\cdot), a_t \sim \pi(\cdot|s_t), s_{t+1} \sim p(\cdot|s_t, a_t)\right). \quad (38)$$

The state-action distribution allows us to express the expected cumulative reward under any reward function $R$ as:

$$\mathbb{E}_{\pi}^{M}[R(s,a)] = \sum_{s,a} \rho_{\pi}(s,a) R(s,a). \tag{39}$$

Using this representation, we can rewrite the left-hand side of equation 10 and bound it via Pinsker's inequality:

$$\left( \frac{1}{\sqrt{2}\bar{R}_{\text{tgt}}} \left( \mathbb{E}_{\pi \circ \nu}^{M}[R_{\text{tgt}}(s,a)] - \mathbb{E}_{\pi}^{M}[R_{\text{tgt}}(s,a)] \right) \right)^{2} = \left( \frac{1}{\sqrt{2}\bar{R}_{\text{tgt}}} \left( \sum_{s,a} \rho_{\pi \circ \nu}(s,a) R(s,a) - \sum_{s,a} \rho_{\pi}(s,a) R(s,a) \right) \right)^{2} \tag{40}$$

$$\leq \left( \frac{1}{\sqrt{2}\bar{R}_{\text{tgt}}} \left( \bar{R}_{\text{tgt}} \sum_{s,a} |\rho_{\pi \circ \nu}(s,a) - \rho_{\pi}(s,a)| \right) \right)^{2} \tag{41}$$

$$= \left( \sqrt{2} D_{\text{TV}}(\rho_{\pi \circ \nu}, \rho_{\pi}) \right)^{2} \tag{42}$$

$$\leq D_{\text{KL}}(\rho_{\pi \circ \nu} \| \rho_{\pi}), \tag{43}$$

where, $D_{\text{TV}}$ represents the Total Variation distance, and $D_{\text{KL}}$ represents the Kullback-Leibler divergence.

Next, we introduce the state distribution. The state distribution $d_{\pi} : \mathcal{S} \to [0,1]$ represents the probability of encountering a specific state when transitioning according to a policy $\pi$:

$$d_{\pi}(s) = (1-\gamma) \sum_{t=0}^{\infty} \gamma^{t} \Pr\left( s_{t} = s \mid s_{0} \sim p_{0}(\cdot), a_{t} \sim \pi(\cdot|s_{t}), s_{t+1} \sim p(\cdot|s_{t}, a_{t}) \right). \tag{44}$$

Building on the definition, we establish the following lemma:

**Lemma B.3.** *Given two policies $\pi, \pi \circ \nu : \mathcal{S} \to \mathcal{P}(\mathcal{A})$ and their state distribution $d_{\pi}, d_{\pi \circ \nu}$, the following inequality holds:*

$$D_{KL}(d_{\pi} \| d_{\pi \circ \nu}) \leq \frac{\gamma^{2}}{1-\gamma^{2}} \sum_{t=1}^{\infty} \gamma^{t} \mathbb{E}_{s \sim d_{\pi}^{t}} [D_{KL}(\pi(\cdot|s) \| \pi \circ \nu(\cdot|s))] \tag{45}$$

*Proof.* This proof is based on Theorem 4.1 of (Belkhale et al., 2024). Regarding the distance between the state distributions of $\pi$ and $\pi \circ \nu$ at time $t$ $D_{\text{KL}}(d_{\pi}^{t}, d_{\pi \circ \nu}^{t})$, the following inequality holds for

$t \geq 1$ by using KL's joint convexity and Jensen's inequality:

$$D_{\text{KL}}(d_\pi^t \| d_{\pi \circ \nu}^t) = \int_{s'} d_\pi^t(s') \log \frac{d_\pi^t(s')}{d_{\pi \circ \nu}^t(s')} \tag{46}$$

$$= \int_{s'} \left( \int_{s,a} \gamma d_\pi^{t-1}(s) \pi(a|s) p(s'|s,a) \right) \log \frac{\int_{s,a} \gamma d_\pi^{t-1}(s) \pi(a|s) p(s'|s,a)}{\int_{s,a} \gamma d_{\pi \circ \nu}^{t-1}(s) \pi \circ \nu(a|s) p(s'|s,a)} \tag{47}$$

$$\leq \int_{s'} \int_{s,a} \gamma d_\pi^{t-1}(s) \pi(a|s) p(s'|s,a) \log \frac{\gamma d_\pi^{t-1}(s) \pi(a|s) p(s'|s,a)}{\gamma d_{\pi \circ \nu}^{t-1}(s) \pi \circ \nu(a|s) p(s'|s,a)} \tag{48}$$

$$\leq \int_{s'} \int_{s,a} \gamma d_\pi^{t-1}(s) \pi(a|s) p(s'|s,a) \left( \log \frac{d_\pi^{t-1}(s)}{d_{\pi \circ \nu}^{t-1}(s)} + \log \frac{\pi(a|s)}{\pi \circ \nu(a|s)} \right) \tag{49}$$

$$\leq \gamma \int_{s,a} d_\pi^{t-1}(s) \pi(a|s) \left( \log \frac{d_\pi^{t-1}(s)}{d_{\pi \circ \nu}^{t-1}(s)} + \log \frac{\pi(a|s)}{\pi \circ \nu(a|s)} \right) \tag{50}$$

$$\leq \gamma \int_s d_\pi^{t-1}(s) \log \frac{d_\pi^{t-1}(s)}{d_{\pi \circ \nu}^{t-1}(s)} + \gamma \int_{s,a} d_\pi^{t-1}(s) \pi(a|s) \log \frac{\pi(a|s)}{\pi \circ \nu(a|s)} \tag{51}$$

$$\leq \gamma D_{\text{KL}}(d_\pi^{t-1} \| d_{\pi \circ \nu}^{t-1}) + \gamma \mathbb{E}_{s \sim d_\pi^{t-1}} [D_{\text{KL}}(\pi(\cdot|s) \| \pi \circ \nu(\cdot|s))] \tag{52}$$

$$\leq \gamma^t D_{\text{KL}}(d_\pi^0 \| d_{\pi \circ \nu}^0) + \sum_{j=0}^{t-1} \gamma^{t-j} \mathbb{E}_{s \sim d_\pi^j} [D_{\text{KL}}(\pi(\cdot|s) \| \pi \circ \nu(\cdot|s))] \tag{53}$$

$$\leq \sum_{j=0}^{t-1} \gamma^{t-j} \mathbb{E}_{s \sim d_\pi^j} [D_{\text{KL}}(\pi(\cdot|s) \| \pi \circ \nu(\cdot|s))] \tag{54}$$

Thus, we obtain the following inequality:

$$D_{\text{KL}}(d_\pi \| d_{\pi \circ \nu}) = \int_s (\sum_{t=0}^\infty \gamma^t d_\pi^t(s)) \log \frac{\sum_{t=1}^\infty \gamma^t d_\pi^t(s)}{\sum_{t=1}^\infty \gamma^t d_{\pi \circ \nu}^t(s)} \tag{55}$$

$$\leq \int_s \sum_{t=0}^\infty \gamma^t d_\pi^t(s) \log \frac{\gamma^t d_\pi^t(s)}{\gamma^t d_{\pi \circ \nu}^t(s)} \tag{56}$$

$$\leq \sum_{t=0}^\infty \gamma^t \int_s d_\pi^t(s) \log \frac{d_\pi^t(s)}{d_{\pi \circ \nu}^t(s)} \tag{57}$$

$$\leq \sum_{t=1}^\infty \gamma^t D_{\text{KL}}(d_\pi^t \| d_{\pi \circ \nu}^t) \tag{58}$$

$$\leq \sum_{t=1}^\infty \gamma^t \sum_{j=0}^{t-1} \gamma^{t-j} \mathbb{E}_{s \sim d_\pi^j} [D_{\text{KL}}(\pi(\cdot|s) \| \pi \circ \nu(\cdot|s))] \tag{59}$$

$$\leq \sum_{t=1}^\infty \sum_{j=0}^{t-1} \gamma^{2t-j} \mathbb{E}_{s \sim d_\pi^j} [D_{\text{KL}}(\pi(\cdot|s) \| \pi \circ \nu(\cdot|s))] \tag{60}$$

$$\leq \frac{\gamma^2}{1 - \gamma^2} \sum_{t=1}^\infty \gamma^t \mathbb{E}_{s \sim d_\pi^t} [D_{\text{KL}}(\pi(\cdot|s) \| \pi \circ \nu(\cdot|s))] \tag{61}$$

$\square$

Finally, by using Lemma B.3, the following inequality holds for the distance between state-action distributions:

$$D_{\text{KL}}(\rho_\pi \| \rho_{\pi \circ \nu}) = \int_{s,a} \rho_\pi(s,a) \log \frac{\rho_\pi(s,a)}{\rho_{\pi \circ \nu}(s,a)} \tag{62}$$

$$= \int_{s,a} \pi(a|s) d_\pi(s) \log \frac{\pi(a|s) d_\pi(s)}{\pi \circ \nu(a|s) d_{\pi \circ \nu}(s)} \tag{63}$$

$$= \int_{s,a} \pi(a|s) d_\pi(s) \log \frac{\pi(a|s)}{\pi \circ \nu(a|s)} + \int_{s,a} \pi(a|s) d_\pi(s) \log \frac{d_\pi(s)}{d_{\pi \circ \nu}(s)} \tag{64}$$

$$= \int_{s,a} \pi(a|s) \left( \sum_{t=0}^{\infty} \gamma^t d_\pi^t(s) \right) \log \frac{\pi(a|s)}{\pi \circ \nu(a|s)} + \int_s d_\pi(s) \log \frac{d_\pi(s)}{d_{\pi \circ \nu}(s)} \tag{65}$$

$$= \sum_{t=0}^{\infty} \gamma^t \int_{s,a} d_\pi^t(s) \pi(a|s) \log \frac{\pi(a|s)}{\pi \circ \nu(a|s)} + D_{\text{KL}}(d_\pi \| d_{\pi \circ \nu}) \tag{66}$$

$$\leq \sum_{t=0}^{\infty} \gamma^t \mathbb{E}_{s \sim d_\pi^t}[D_{\text{KL}}(\pi(\cdot|s) \| \pi \circ \nu(\cdot|s))] + \frac{\gamma^2}{1-\gamma^2} \sum_{t=1}^{\infty} \gamma^t \mathbb{E}_{s \sim d_\pi^t}[D_{\text{KL}}(\pi(\cdot|s) \| \pi \circ \nu(\cdot|s))] \tag{67}$$

$$\leq \sum_{t=0}^{\infty} \gamma^t \mathbb{E}_{s \sim d_\pi^t}[D_{\text{KL}}(\pi(\cdot|s) \| \pi \circ \nu(\cdot|s))] + \frac{\gamma^2}{1-\gamma^2} \sum_{t=0}^{\infty} \gamma^t \mathbb{E}_{s \sim d_\pi^t}[D_{\text{KL}}(\pi(\cdot|s) \| \pi \circ \nu(\cdot|s))] \tag{68}$$

$$= \frac{1}{1-\gamma^2} \sum_{t=0}^{\infty} \gamma^t \mathbb{E}_{s \sim d_\pi^t}[D_{\text{KL}}(\pi(\cdot|s) \| \pi \circ \nu(\cdot|s))] \tag{69}$$

$$\square$$

# C   THEORETICAL ANALYSIS OF THE DISTRIBUTION MATCHING APPROACH IN OUR ADVERSARIAL ATTACK SETTING

In this section, we prove that the distribution matching approach in our problem setting is equivalent to the problem of minimizing the distance between policies through inverse reinforcement learning. This shows that the adversarial policy, which is learned through our distribution matching approach, rigorously mimics the target policy. Our procedures follow the proof of Proposition 3.1 in (Ho & Ermon, 2016).

Let a set of all stationary stochastic policies as $\Pi$ and a set of all stationary stochastic adversarial policies as $\mathcal{N}$. Also, we write $\bar{\mathbb{R}}$ for extended real numbers $\mathbb{R} \cup \{\infty\}$. The goal of inverse reinforcement learning is to find a reward function such that when a policy is learned to maximize the rewards obtained from this function, it matches the expert's policy. This process aims to derive a reward function from the expert's trajectories. We formulate the adversary's objective as learning an adversarial policy that maximizes the victim's cumulative reward from the reward function estimated by IRL:

$$\text{IRL}_{\psi,\nu}(\pi_{\text{tgt}}) = \underset{c \in \mathbb{R}^{\mathcal{S} \times \mathcal{A}}}{\arg\max} \, -\psi(c) + \left( \min_{\nu \in \mathcal{N}} \mathbb{E}_{\pi \circ \nu}^M[c(s,a)] \right) - \mathbb{E}_{\pi_{\text{tgt}}}^M[c(s,a)], \tag{70}$$

$$\text{RL}(c) = \underset{\nu \in \mathcal{N}}{\arg\min} \, \mathbb{E}_{\pi \circ \nu}[c(s,a)], \tag{71}$$

where $c : \mathcal{S} \times \mathcal{A} \to \mathbb{R}$ is a cost function, $\psi : \mathbb{R}^{\mathcal{S} \times \mathcal{A}} \to \bar{\mathbb{R}}$ is a convex cost function regularization. Note that we use a cost function instead of a reward function to represent reinforcement learning as a minimization problem. Let $c^* \in \text{IRL}_{\psi,\nu}(\pi_{\text{tgt}})$ be the optimal cost function through IRL. The optimal adversarial policy is learned with respect to the optimal cost function: $\nu^* \in \text{RL}(c^*)$.

For the proof, we define the occupancy measure. The occupancy measure is an unnormalized state-action distribution:

$$\hat{\rho}_\pi(s,a) = \sum_{t=0}^{\infty} \gamma^t \Pr\left(s_t = s, a_t = a | s_0 \sim p_0(\cdot), a_t \sim \pi(\cdot|s_t), s_{t+1} \sim p(\cdot|s_t, a_t)\right). \tag{72}$$

Therefore, we present the Theorem C.1. The Theorem shows that the optimal adversarial policy obtained via inverse reinforcement learning coincides with the optimal adversarial policy obtained through the distribution matching approach:

**Theorem C.1.** *Let $\mathcal{C}$ be the set of cost functions, $\psi$ be a convex function, and $\psi^*$ be the conjugate of $\psi$. When $\pi$ is fixed and $\mathcal{C}$ is a compact convex set, the following holds:*

$$\text{RL} \circ \text{IRL}_{\psi,\nu}(\pi_{tgt}) = \arg\min_{\nu \in \mathcal{N}} \psi^*(\hat{\rho}_{\pi \circ \nu} - \hat{\rho}_{\pi_{tgt}}). \tag{73}$$

*Proof.* Let $\mathcal{D}_{\pi \circ \nu} = \{\hat{\rho}_{\pi \circ \nu} | \nu \in \mathcal{N}\}$. If $\mathcal{D}_{\pi \circ \nu}$ is a compact and convex set, equation 73 is valid according to the proof of Theorem 3.1 in (Ho & Ermon, 2016). Thus, we prove that $\mathcal{D}_{\pi \circ \nu}$ is a compact and convex set.

**Compactness:** The mapping from $\nu$ to $\pi \circ \nu$ is linear, and $\mathcal{N}$ is compact. Therefore, by (Arkhangel'skiĭ & Fedorchuk, 1990), $\Pi_\nu$ is a compact set. From (Ho & Ermon, 2016), when $\Pi_\nu$ is compact, $\mathcal{D}_{\pi \circ \nu}$ is also compact. Consequently, $\mathcal{D}_{\pi \circ \nu}$ is a compact set.

Next, we show that $\mathcal{D}_{\pi \circ \nu}$ is closed. Policy $\pi \in \Pi$ and occupancy measure $\hat{\rho} \in \mathcal{D}$ have a one-to-one correspondence by Lemma 1 in (Ho & Ermon, 2016). Let $\Pi_\nu = \{\pi' \mid \nu \in \mathcal{N}, \pi' = \pi \circ \nu\}$ be the set of all behavior policies. Since $\Pi_\nu \subseteq \Pi$, $\pi \circ \nu \in \Pi_\nu$ and $\hat{\rho}_{\pi \circ \nu} \in \mathcal{D}_{\pi \circ \nu}$ also have a one-to-one correspondence. Thus, let $\{\hat{\rho}_{\pi \circ \nu_1}, \hat{\rho}_{\pi \circ \nu_2}, \dots\}$ be any cauchy sequence with $\hat{\rho}_{\pi \circ \nu_n} \in \mathcal{D}_{\pi \circ \nu}$. Due to the one-to-one correspondence, there exists a corresponding sequence $\{\pi \circ \nu_1, \pi \circ \nu_2, \dots\}$. Since $\Pi_\nu$ is compact, the sequence $\{\pi \circ \nu_1, \pi \circ \nu_2, \dots\}$ converges to $\pi \circ \nu \in \Pi_\nu$. Hence, the sequence $\{\hat{\rho}_{\pi \circ \nu_1}, \hat{\rho}_{\pi \circ \nu_2}, \dots\}$ also converges to the occupancy measure $\hat{\rho}_{\pi \circ \nu}$ corresponding to $\pi \circ \nu$. Therefore, $\mathcal{D}_{\pi \circ \nu}$ is closed.

**Convexity:** First, we show that $\Pi_\nu$ is a convex set. For $\forall \nu_1 \in \mathcal{N}, \forall \nu_2 \in \mathcal{N}$ and $\lambda \in [0,1]$, we define $\pi \circ \nu$ as

$$\pi \circ \nu = \lambda \pi \circ \nu_1 + (1 - \lambda)\pi \circ \nu_2. \tag{74}$$

Then, we have

$$\pi \circ \nu = \lambda \pi \circ \nu_1 + (1 - \lambda)\pi \circ \nu_2 \tag{75}$$

$$= t \sum_{\hat{a} \in \mathcal{S}} \nu_1(\hat{a}|s)\pi(a|\hat{a}) + (1 - t) \sum_{\hat{a} \in \mathcal{S}} \nu_2(\hat{a}|s)\pi(a|\hat{a}) \tag{76}$$

$$= \sum_{\hat{a} \in \mathcal{S}} (t\nu_1(\hat{a}|s) + (1 - t)\nu_2(\hat{a}|s))\pi(a|\hat{a}) \tag{77}$$

$$= \pi \circ (\lambda \nu_1 + (1 - \lambda)\nu_2) \tag{78}$$

Using the convexity of $\mathcal{N}$, we have $t\nu_1 + (1 - t)\nu_2 \in \mathcal{N}$. Thus, $\pi \circ \nu \in \Pi_\nu$ holds for any $\nu_1 \in \mathcal{N}, \nu_2 \in \mathcal{N}$, and $\lambda \in [0,1]$ and $\Pi_\nu$ is convex.

Noting that the one-to-one correspondence of $\hat{\rho}_{\pi \circ \nu} \in \mathcal{D}_{\pi \circ \nu}$ and $\pi \circ \nu \in \Pi_\nu$, for any mixture policy $\pi \circ \nu_{\text{m}} \in \Pi_\nu$, we have $\hat{\rho}_{\pi \circ \nu_{\text{m}}} \in \mathcal{D}_{\pi \circ \nu}$. Consequently, $\mathcal{D}_{\pi \circ \nu}$ is a convex set.

Based on the above, we prove the Theorem following the same procedure as the proof of Theorem 3.1 in (Ho & Ermon, 2016). Let $\tilde{c} \in \text{IRL}_{\psi,\nu}(\pi_{\text{tgt}}), \widetilde{\pi \circ \nu} \in \text{RL}(\tilde{c}) = \text{RL} \circ \text{IRL}_{\psi,\nu}(\pi_{\text{tgt}})$. The RHS of 73 is denoted by

$$\pi \circ \nu_A \in \arg\min_{\pi \circ \nu} \psi^*(\hat{\rho}_{\pi \circ \nu} - \hat{\rho}_{\pi_{\text{tgt}}}) = \arg\min_{\pi \circ \nu} \max_c -\psi(c) + \int_{s,a} (\hat{\rho}_{\pi \circ \nu}(s,a) - \hat{\rho}_{\pi_{\text{tgt}}}(s,a))c(s,a). \tag{79}$$

We define the RHS of 73 by $\bar{L} : \mathcal{D}_{\pi \circ \nu} \times \mathcal{C} \to \mathbb{R}$ as follow:

$$\bar{L}(\hat{\rho}, c) = -\psi(c) + \int_{s,a} \hat{\rho}(s,a)c(s,a) - \int_{s,a} \hat{\rho}_{\pi_{\text{adv}}}c(s,a). \tag{80}$$

We remark that $\bar{L}$ takes an occupancy measure as its argument.

To prove $\widetilde{\pi \circ \nu} = \pi \circ \nu_A$, we utilize the minimax duality of $\bar{L}$.

Policy $\pi \in \Pi$ and occupancy measure $\hat{\rho} \in \mathcal{D}$ have a one-to-one correspondence by Lemma 1 in (Ho & Ermon, 2016). Since $\Pi_\nu \subseteq \Pi$, $\pi \circ \nu \in \Pi_\nu$ and $\hat{\rho}_{\pi \circ \nu} \in \mathcal{D}_{\pi \circ \nu}$ also have a one-to-one correspondence. Thus, the following relationship is established:

$$\hat{\rho}_{\pi \circ \nu_A} \in \arg\min_{\hat{\rho} \in \mathcal{D}_{\pi \circ \nu}} \max_{c \in \mathcal{C}} \bar{L}(\hat{\rho}, c), \tag{81}$$

$$\tilde{c} \in \arg\max_{c \in \mathcal{C}} \min_{\hat{\rho} \in \mathcal{D}_{\pi \circ \nu}} \bar{L}(\hat{\rho}, c), \tag{82}$$

$$\hat{\rho}_{\widetilde{\pi \circ \nu}} \in \arg\min_{\hat{\rho} \in \mathcal{D}_{\pi \circ \nu}} \bar{L}(\hat{\rho}, \tilde{c}). \tag{83}$$

$\mathcal{D}_{\pi \circ \nu}$ is a compact convex set and $\mathcal{C}$ is also a compact convex set. Since $\psi$ is a convex function, we have that $\bar{L}(\cdot, c)$ is convex for all $c$, and that $\bar{L}(\hat{\rho}, \cdot)$ is liner for all $\hat{\rho}$, so $\bar{L}(\hat{\rho}, \cdot)$ is concave for all $\hat{\rho}$. Due to minimax duality(Fernique et al., 1983), we have the following equality:

$$\min_{\hat{\rho} \in \mathcal{D}_{\pi \circ \nu}} \max_{c \in \mathcal{C}} \bar{L}(\hat{\rho}, c) = \max_{c \in \mathcal{C}} \min_{\hat{\rho} \in \mathcal{D}_{\pi \circ \nu}} \bar{L}(\hat{\rho}, c). \tag{84}$$

Therefore, from equation 81 and equation 82, $(\hat{\rho}_{\pi \circ \nu_A}, \tilde{c})$ is a saddle point of $\bar{L}$, which implies that $\hat{\rho}_{\pi \circ \nu_A} \in \arg\min_{\hat{\rho} \in \mathcal{D}_{\pi \circ \nu}} \bar{L}(\hat{\rho}, \tilde{c})$ and so $\hat{\rho}_{\widetilde{\pi \circ \nu}} = \hat{\rho}_{\pi \circ \nu_A}$. $\qquad\square$

The right-hand side of equation 73 represents the optimal adversarial policy that minimizes the distance between occupancy measures as measured by $\psi^*$. Therefore, Theorem C.1 indicates that the optimal adversarial policy obtained through the distribution matching approach is equivalent to that via inverse reinforcement learning.

# D  EXTENSION OF BIA

In this section, we verify that the assumptions of Theorem 5.1 hold for other imitation learning methods beyond GAIL and demonstrate that BIA can be applied to a variety of existing algorithms. As concrete examples, we consider GAIfO (Torabi et al., 2019), an ILfO extension of GAIL, and AILBoost (Chang et al., 2024), a state-of-the-art ILfD method.

## D.1  GAIFO

GAIfO is an optimization method applicable in the ILfO setting, where only expert state transitions are provided. It extends GAIL to ILfO and optimizes the policy using a GAN-style algorithm. In GAIfO, we use a discriminator $D_o \colon \mathcal{S} \times \mathcal{S} \to \mathbb{R}$ that takes state $s \in \mathcal{S}$ and next state $s' \in \mathcal{S}$ as input, and reformulate the adversary's objective as:

$$\arg\min_\nu \max_{D_o} \mathbb{E}^M_{\pi \circ \nu} \left[ \left[ \log D_o(s, s') \right] + \mathbb{E}^M_{\pi_{\text{tgt}}} \left[ \log\left(1 - D_o(s, s')\right) \right] \right]. \tag{85}$$

This reformulation leverages that the objective can be reduced to a distribution matching problem similar to GAIL. By applying the discussion in Appendix C to the ILfO setting, we can also convert the adversary's objective in our problem setup to a distribution matching problem. Thus, it ensures that the reformulation of equation 85 holds. Consequently, by Theorem 5.1, it is transformed into the following objective:

$$\arg\max_\nu \mathbb{E}^{\hat{M}}_\nu \left[ \hat{R}_{D_o} \right], \tag{86}$$

where $\hat{R}_{D_o}$ is the reward function obtained by replacing $g(d_\star)$ with the optimal discriminator $\log D_{o\star}$ in Equation 4. equation 86 can be optimized by any reinforcement learning algorithm. Therefore, BIA can also be applied to GAIfO.

## D.2  AILBOOST

AILBoost is a state-of-the-art ILfD method based on gradient boosting, which allows the use of more efficient off-policy algorithms compared to the on-policy methods used in GAIL. In AILBoost, the objective is expressed in a variational form using a weighted ensemble of policies rather than a single policy:

$$\arg\min_\nu \max_D \left[ \mathbb{E}^M_{\boldsymbol{\pi} \circ \nu} \left[ D(s, a) \right] + \mathbb{E}^M_{\pi_{\text{tgt}}} \left[ -\exp\left( D(s, a) \right) \right] \right], \tag{87}$$

where $\boldsymbol{\pi} \circ \boldsymbol{\nu} \triangleq \{\alpha_i, \pi \circ \nu_i\}$ denotes the weighted ensemble with $\alpha_i \geq 0$ and $\sum_i \alpha_i = 1$. When executing $\boldsymbol{\pi} \circ \boldsymbol{\nu}$, at the beginning of an episode, a single policy $\pi \circ \nu_i$ is sampled with probability $\alpha_i$, and then $\pi \circ \nu_i$ is executed for the entire episode.

Using Lemma B.1, we show that Theorem 5.1 also applies to this ensemble policy. First, given the ensemble $\boldsymbol{\pi} \circ \boldsymbol{\nu}^{(t)} = \{\alpha_i, \pi \circ \nu_i\}_{i \leq t}$ at iteration $t$, we train a discriminator $D_t$ on the experiences collected by $\boldsymbol{\pi} \circ \boldsymbol{\nu}^{(t)}$. Then we optimize the next weak policy $\pi \circ \nu_{t+1}$ as

$$\pi \circ \nu_{t+1} = \arg\max_{\pi \circ \nu} \mathbb{E}_{\pi \circ \nu}^M \big[ -D_t(s, a) \big]. \tag{88}$$

After the optimization, the existing weights $\alpha_i$ are rescaled by the weighting parameter $\alpha$ to $\alpha_i(1-\alpha)$, and the newly obtained $\pi \circ \nu_{t+1}$ is added to the ensemble with weight $\alpha$.

By Lemma B.1, this is equivalent to the following reinforcement learning problem:

$$\nu_{t+1} = \arg\max_{\nu} \mathbb{E}_{\nu}^{\hat{M}} \big[ \hat{R}_{D_t}(s, a) \big], \tag{89}$$

where $\hat{R}_{D_t}$ is the reward function obtained by replacing $g(d_\star)$ with $D_{t\star}$ in equation 4. Since equation 89 defines a standard reinforcement learning problem, the ensemble policy can be learned accordingly. Thus, Theorem 5.1 holds for AILBoost as well, demonstrating that BIA is applicable.

## E   TDRT-PPO ALGORITHM

---

**Algorithm 2** Time-Discounted Robust Training in PPO (TDRT-PPO)

---

1: **Input:** Number of iterations $T$, clipping parameter $\epsilon_c$, Minibatch size $M$, regularization coefficient $\lambda$, learning rates $\alpha_\pi, \alpha_V$
2: **Output:** Optimized policy $\pi_\theta$
3: Initialize actor network $\pi_\theta(a|s)$ and critic network $V_\phi(s)$
4: **for** $t = 1$ to $T$ **do**
5: $\quad \mathcal{D} \leftarrow$ Collect trajectories using current policy $\pi_\theta$
6: $\quad$ **for** each $(s_t, a_t, r_t, s_{t+1})$ in $\mathcal{D}$ **do**
7: $\quad\quad \hat{R}_t \leftarrow \sum_{l=0}^{\infty} \gamma^l r_{t+l}$
8: $\quad\quad \hat{A}_t \leftarrow \hat{R}_t - V_\phi(s_t)$
9: $\quad$ **end for**
10: $\quad$ **for** $K$ epochs **do**
11: $\quad\quad B \leftarrow \{(s_{t_i}, a_{t_i}, \hat{R}_{t_i}, \hat{A}_{t_i})\}_{i=1}^M \sim \mathcal{D}$
12: $\quad\quad \phi \leftarrow \phi - \alpha_V \nabla_\phi \frac{1}{M} \sum_{i=1}^M (V_\phi(s_i) - \hat{R}_i)^2$
13: $\quad\quad$ # Compute the time-discounted regularization term
14: $\quad\quad \mathcal{R}_\theta \leftarrow \sum_{s_{t_i} \in B} \max_{\hat{s}_{t_i} \in \mathcal{B}(s_{t_i})} \gamma^{t_i} D_{\mathrm{KL}}(\pi_\theta(\cdot|s_{t_i}) \| \pi_\theta(\cdot|\hat{s}_{t_i}))$
15: $\quad\quad r_i(\theta) = \frac{\pi_\theta(a_i|s_i)}{\pi_{\theta_{\mathrm{old}}}(a_i|s_i)}$
16: $\quad\quad$ # Update actor-network with regularization
17: $\quad\quad \theta \leftarrow \theta + \alpha_\pi \nabla_\theta \Big( \frac{1}{M} \sum_{i=1}^M \min(r_i(\theta)\hat{A}_i, \mathrm{clip}(r_i(\theta), 1 - \epsilon_c, 1 + \epsilon_c)\hat{A}_i) + \lambda \mathcal{R}_\theta \Big)$
18: $\quad$ **end for**
19: **end for**

---

We present the complete algorithm for TDRT-PPO in Algorithm 2. This algorithm extends the standard PPO algorithm by incorporating time-discounted regularization $\mathcal{R}_\theta$.

Directly computing the maximum value of the KL term in line 14 of Algorithm 2 is computationally expensive. Following SA-PPO (Zhang et al., 2020b), we therefore apply convex-relaxation methods (Zhang et al., 2018; Wong & Kolter, 2018; Salman et al., 2019; Zhang et al., 2020a) to obtain tight upper bounds. We implemented this using the auto_LiRPA toolkit (Xu et al., 2020), reducing the computational cost of the regularization term and enabling an efficient implementation of TDRT-PPO. This regularization-based approach achieves shorter training times compared to adversarial training methods that require learning adversarial policies (Liang et al., 2022). A detailed analysis of the training time efficiency is provided in Section H.3.

Table 4: Comparison of attack and defense performances in MuJoCo environments. Each value represents the average episode reward $\pm$ standard deviation over 50 episodes. Each parenthesis indicates (*Access model*, *Adversary's knowledge*). For defense methods, better defense performance is achieved with smaller attack rewards. For attack methods, higher attack rewards lead to better performance. **Our proposed attack and defense methods are also effective for MuJoCo environments.**

| Task | Target Reward | Method | Clean Reward | Attack Reward | | | |
|------|--------------|--------|--------------|---------------|--|--|--|
| | | | | **Targeted PGD** (White-box, target policy) | **Rew Max (SA-RL)** (Black-box, reward function) | **BIA-ILfD (ours)** (Black-box, demonstration) | **Avg.** |
| **Ant** | $4574 \pm 311$ | PPO (No defense) | $3027 \pm 35$ | $3975 \pm 281$ | $4681 \pm 202$ | $4423 \pm 123$ | 4359 |
| | | TDRT-PPO (Ours) | $2944 \pm 243$ | $2763 \pm 87$ | $3425 \pm 12$ | $3278 \pm 83$ | 3155 |
| **HalfCheetah** | $3688 \pm 291$ | PPO (No defense) | $2178 \pm 4$ | $2823 \pm 519$ | $4023 \pm 98$ | $3602 \pm 75$ | 3482 |
| | | TDRT-PPO (Ours) | $1970 \pm 38$ | $2023 \pm 274$ | $2606 \pm 48$ | $2084 \pm 39$ | 2237 |
| **Hopper** | $3513 \pm 182$ | PPO (No defense) | $1405 \pm 54$ | $1673 \pm 54$ | $3056 \pm 42$ | $3198 \pm 9$ | 2642 |
| | | TDRT-PPO (Ours) | $1310 \pm 191$ | $1324 \pm 321$ | $2190 \pm 234$ | $1481 \pm 2$ | 1665 |

During the KL computation in line 17, gradients are back-propagated only through the output associated with the perturbed state, while the output for the original state is held constant. This design confines the perturbed output to remain close to the original output.

For clarity, we denote perturbed states by $\hat{s} \in \mathcal{B}(s)$. However, since the defender lacks knowledge of the adversary's true ball $\mathcal{B}$, $\hat{s}$ is optimized within a defender-specified region instead.

# F  ADDITIONAL EXPERIMENTS

In this section, we evaluate the generalizability of our proposed method by conducting additional experiments in two distinct environments characterized by diverse state and action spaces. The first environment is MuJoCo (Todorov et al., 2012), which features both continuous states and continuous actions in a robotic locomotion task. The second is the MiniGrid environment(Chevalier-Boisvert et al., 2018), which involves a maze task with a discrete action space. For MiniGrid, we conduct experiments under two configurations: one where the agent's state is represented as coordinates and another where it is represented as images.

## F.1  EXPERIMENTS ON MUJOCO ENVIRONMENTS

**Setup.** As a possible attack in the real world, the adversary aims to increase the victim's cumulative reward, which the victim intentionally constrained during training. For instance, in an autonomous driving scenario where reaching the destination sooner yields higher rewards, the agent's speed may increase significantly, which may compromise safety by increasing the risk of accidents. To mitigate this, the victim may adjust the reward function during training to maintain a balanced level of performance. However, the adversary's goal here is to force exceptionally high rewards, thereby inducing the victim to adopt excessively fast and potentially unsafe driving behaviors.

We set the adversary's objective to maximize the victim's reward, while the victim's policy is intentionally constrained to achieve moderate rewards. We define the target policy as one that attains very high rewards on the same tasks. We use the Ant, HalfCheetah, and Hopper tasks, each with a reward function that increases as the agent moves to the right. All other experimental settings are the same as those described in Section 7.

**Attack and Defense Performance Results.** Table 4 presents the results on an attack budget $\epsilon = 0.3$. The attack reward represents the reward obtained by the victim under the attack. When evaluating attack methods, a higher attack reward indicates stronger attack performance. Conversely, when evaluating defense methods, a lower attack reward implies greater robustness against attacks. The target reward is an upper bound for the attack reward, the reward obtained by a well-trained target policy. The clean reward denotes the reward obtained by the victim in non-attack. Unlike in the experiments in Section 7, the victim is intentionally not fully trained. As a result, the clean reward does not indicate whether the defense method affects the original performance.

The targeted PGD attack does not exhibit sufficient attack performance, which is consistent with the attack performance in the Meta-World experiments, where it also failed to manipulate the victim

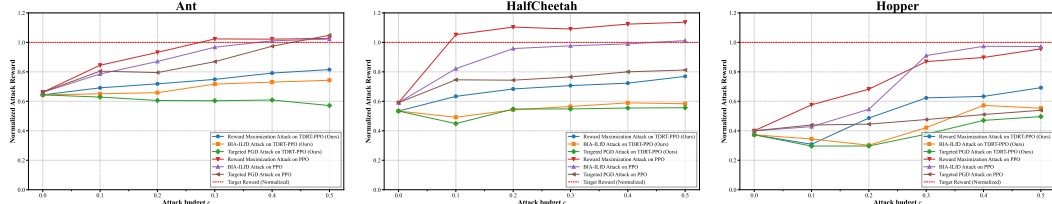

Figure 2: Attack and defense performances under various attack budgets $\epsilon$ in MuJoCo environments. The horizontal axis represents the attack budget, which indicates the adversary's intervention capability. The vertical axis shows the attack reward, which represents the reward obtained during the attack. Each value represents the average reward over 50 episodes.

Table 5: Comparison of defense performances under the best attack in MuJoCo environments. Each value represents the average episode reward $\pm$ standard deviation over 50 episodes. Higher clean rewards and lower best attack rewards indicate better defense performance.

| Task | Method | Clean Reward ($\uparrow$) | Best Attack Reward ($\uparrow$) |
|---|---|---|---|
| **Ant** | PPO | $4720 \pm {}_{255}$ | $850 \pm {}_{612}$ |
| | SA-PPO (smoothing w/o time-discounting) | $4268 \pm {}_{342}$ | $\mathbf{3318 \pm {}_{391}}$ |
| | **TDRT-PPO (ours, smoothing w/ time-discounting)** | $\mathbf{4612 \pm {}_{219}}$ | $3206 \pm {}_{781}$ |
| **HalfCheetah** | PPO | $3680 \pm {}_{220}$ | $160 \pm {}_{540}$ |
| | SA-PPO (smoothing w/o time-discounting) | $2963 \pm {}_{398}$ | $2826 \pm {}_{513}$ |
| | **TDRT-PPO (ours, smoothing w/ time-discounting)** | $\mathbf{3577 \pm {}_{197}}$ | $\mathbf{2981 \pm {}_{462}}$ |
| **Hopper** | PPO | $3290 \pm {}_{150}$ | $820 \pm {}_{430}$ |
| | SA-PPO (smoothing w/o time-discounting) | $3198 \pm {}_{97}$ | $\mathbf{1579 \pm {}_{672}}$ |
| | **TDRT-PPO (ours, smoothing w/ time-discounting)** | $\mathbf{3214 \pm {}_{163}}$ | $1345 \pm {}_{787}$ |

effectively. This suggests that single-step optimization is ineffective in manipulating overall behavior in this experimental setting. Comparing the reward maximization attack and BIA, we observe that the reward maximization attack demonstrates stronger attack performance. We argue that this occurs because BIA excessively alters the victim's behavior to match that of the target policy. In MuJoCo tasks, moving to the right yields higher rewards. Therefore, in the reward maximization attack, since the adversary's objective is to maximize the victim's reward, no perturbation is applied when the victim is already moving to the right. Thus, the attack does not interfere with the victim's movement. On the other hand, in BIA, the attack aims to make the victim's behavior close to the target policy's behavior, regardless of the reward. Consequently, even if the victim is already moving to the right, perturbations are still applied to change the victim's behavior. This excessive modification disrupts stable locomotion, ultimately reducing the attack rewards. In evaluation of defense performance, TDRT-PPO results in a lower attack reward than vanilla PPO, indicating that it is more robust against behavior-targeted attacks.

**Attack Performance on Various Attack Budgets.** To further analyze attack performance, we conduct experiments with various attack budgets. The experimental results are shown in Figure 2. To standardize the scale across different tasks, the attack reward is normalized so that the target reward is set to 1.

In the Ant task, the targeted PGD attack demonstrated high attack performance at large attack budgets. However, in the remaining tasks, it failed to achieve sufficient attack performance even with a large attack budget. Similar to the experiments in Meta-World, this result suggests that there are no falsified states where the victim's chosen actions perfectly match those of the target policy. In the HalfCheetah task, under the reward maximization attack, the attack reward for vanilla PPO exceeds the target reward. This occurs because the reward maximization attack only observes the victim's obtained rewards and performs attacks independently of the target policy's rewards. As a result, the victim under attack achieves higher rewards than the target policy. In contrast, in BIA, the target reward serves as the upper bound for the attack, meaning that the attack reward never exceeds this limit.

**Defense Performance Comparison.** We also compare TDRT with the baseline method in the MuJoCo environment. Since we cannot directly compare clean performance in the experimental setup

Table 6: Comparison of attack and defense performances for the setting where the state is represented as coordinates in MiniGrid. Each value represents the average episode reward $\pm$ standard deviation over 50 episodes. Each parenthesis indicates (*Access model*, *Adversary's knowledge*). For defense methods, better defense performance is achieved with smaller attack rewards. For attack methods, higher attack rewards lead to better performance. **Our proposed attack and defense methods are also effective for discrete action spaces.**

| Task | Target Reward | Method | Clean Reward | Attack Reward | | | |
|------|---------------|--------|--------------|---------------|---|---|---|
| | | | | **Rew Max (SA-RL)** (Black-box, reward function) | **Rew Max (PA-AD)** (White-box, reward function) | **BIA-ILfD (ours)** (Black-box, demonstration) | **Avg.** |
| $8 \times 8$ | $0.96 \pm 0$ | PPO (No defense) | $0.44 \pm 0.17$ | $0.91 \pm 0.03$ | $0.89 \pm 0.02$ | $0.94 \pm 0.05$ | $0.91$ |
| | | TDRT-PPO (Ours) | $0.39 \pm 0.12$ | $0.65 \pm 0.01$ | $0.58 \pm 0.04$ | $0.54 \pm 0.03$ | $0.59$ |
| $16 \times 16$ | $0.98 \pm 0$ | PPO (No defense) | $0.46 \pm 0.21$ | $0.95 \pm 0.02$ | $0.93 \pm 0.03$ | $0.96 \pm 0.01$ | $0.95$ |
| | | TDRT-PPO (Ours) | $0.50 \pm 0.04$ | $0.53 \pm 0.04$ | $0.47 \pm 0.01$ | $0.38 \pm 0.05$ | $0.46$ |

Table 7: Comparison of attack and defense performances for the setting where the state is represented as an image in MiniGrid. Each value represents the average episode reward $\pm$ standard deviation over 50 episodes. **Our proposed defense method is effective for image inputs. However, the adversarial policy without white-box access is ineffective for image inputs.**

| Task | Target Reward | Method | Clean Reward | Attack Reward | | | |
|------|---------------|--------|--------------|---------------|---|---|---|
| | | | | **Rew Max (SA-RL)** (Black-box, reward function) | **Rew Max (PA-AD)** (White-box, reward function) | **BIA-ILfD (ours)** (Black-box, demonstration) | **Avg.** |
| $8 \times 8$ (RGB image input) | $0.96 \pm 0$ | PPO (No defense) | $0.37 \pm 0.19$ | $0.50 \pm 0.15$ | $0.92 \pm 0.03$ | $0.48 \pm 0.20$ | $0.63$ |
| | | TDRT-PPO (Ours) | $0.43 \pm 0.17$ | $0.32 \pm 0.18$ | $0.58 \pm 0.02$ | $0.29 \pm 0.16$ | $0.40$ |
| $16 \times 16$ (RGB image input) | $0.98 \pm 0$ | PPO (No defense) | $0.38 \pm 0.56$ | $0.49 \pm 0.26$ | $0.94 \pm 0.02$ | $0.48 \pm 0.24$ | $0.64$ |
| | | TDRT-PPO (Ours) | $0.44 \pm 0.27$ | $0.47 \pm 0.19$ | $0.63 \pm 0.01$ | $0.39 \pm 0.22$ | $0.50$ |

described above, we apply a reward-minimization setting. Specifically, we define the target policy as the policy trained with a reward function whose sign is flipped from the original task reward. Unlike the previous experiments, the victim policy is sufficiently trained before the attack is applied. The results in Table 5 show that TDRT-PPO achieves robustness comparable to SA-PPO while attaining higher clean rewards, which is consistent with the findings in Section 7. These results demonstrate the versatility of TDRT across different environments.

## F.2 EXPERIMENTS ON MINIGRID ENVIRONMENTS

**Setup.** MiniGrid is 2D grid-world environments, where the agent's objective is to reach a designated goal coordinate. The agent's actions are defined over a discrete space that includes movements and interactions such as picking up keys. We evaluate tasks on both $8 \times 8$ and $16 \times 16$ grids. Similar to experiments on MuJoCo in Section F.1, e set the adversary's objective to maximize the victim's reward, while the victim's policy is intentionally constrained to achieve moderate rewards. We define the target policy as one that attains very high rewards on the same tasks. We use the Ant, HalfCheetah, and Hopper tasks, each with a reward function that increases as the agent moves to the right. All other experimental settings are the same as those described in Section 7.

**Results in coordinate states.** Table 6 shows the results for the setting in which the state is represented as coordinates, with the attack budget set to $\epsilon = 0.3$. Our experimental results show that our attack and defense methods are effective even in the discrete action spaces. Specifically, BIA-ILfD exhibited higher attack rewards against PPO (No defense), and TDRT-PPO achieved lower attack rewards than PPO (No defense). Since the attack is executed on the victim policy's state space, it is as effective in discrete action spaces as it is in continuous ones.

**Results in vision-based states.** Next, we evaluate the effectiveness of our proposed method in vision-based control tasks. We modify the MiniGrid environment to use RGB image inputs. Table 7 shows the results for the setting in which the state is represented as an image, with the attack budget set to $\epsilon = \frac{3}{255}$. Our experimental results show that the effectiveness of our attack method decreases with image inputs. When the states are represented as RGB images, the dimensionality of the state space increases significantly. Consequently, the action space of the adversarial policy expands, making learning much more challenging. This problem is not unique to our method but is common

Table 8: Comparison of defense performances under the best attack in MinGrid environments. Each value represents the average episode reward $\pm$ standard deviation over 50 episodes. Higher clean rewards and lower best attack rewards indicate better defense performance.

| Task | Method | Clean Reward ($\uparrow$) | Best Attack Reward ($\uparrow$) |
|---|---|---|---|
| $8 \times 8$ | PPO | $0.93 \pm {}_{0.04}$ | $0.12 \pm {}_{0.32}$ |
| | SA-PPO (smoothing w/o time-discounting) | $0.93 \pm {}_{0.05}$ | $0.78 \pm {}_{0.11}$ |
| | **TDRT-PPO (ours, smoothing w/ time-discounting)** | $0.92 \pm {}_{0.08}$ | $0.74 \pm {}_{0.15}$ |
| $16 \times 16$ | PPO | $0.92 \pm {}_{0.03}$ | $0.24 \pm {}_{0.22}$ |
| | SA-PPO (smoothing w/o time-discounting) | $0.92 \pm {}_{0.01}$ | $0.71 \pm {}_{0.12}$ |
| | **TDRT-PPO (ours, smoothing w/ time-discounting)** | $0.93 \pm {}_{0.04}$ | $0.74 \pm {}_{0.04}$ |

among adversarial policy-based attacks such as SA-RL. PA-AD overcomes this limitation but requires white-box access. Thus, targeted attacks under black-box access in vision-based control tasks remain as future work.

**Defense Performance Comparison.** Analogously to the MuJoCo experiments, we compare TDRT with the baseline method in the MiniGrid environment. Because attack performance is insufficient when using vision-based observations, we restrict our evaluation to coordinate-state settings, while keeping all other experimental configurations identical to the MuJoCo setup. The results are summarized in Table 8. TDRT-PPO and SA-PPO exhibit similar robustness and clean performance, which we conjecture is because MiniGrid is a relatively easy task in which both methods nearly saturate the achievable performance, leaving little room for noticeable differences in clean performance.

# G  ADDITIONAL ANALYSIS IN ATTACK METHODS

## G.1  DEMONSTRATION EFFICIENCY

In this section, we conduct additional experiments to investigate how the quantity and quality of target policy demonstrations affect the attack performance of BIA-ILfD/ILfO. We evaluate the performance of BIA-ILfD/ILfO using different amounts of demonstrations: 1, 4, 8, 12, 16, and 20 episodes. The victim is a vanilla PPO agent without any defense method. All experimental settings remain the same as those in Section 7, with the attack budget $\epsilon$ set to 0.3.

Figure 3 shows the experimental results. Across all tasks, we observed no significant performance degradation when using up to four demonstration episodes. However, performance declined when only one demonstration episode was provided. This decline can be attributed to the variability in initial states, where the discriminator cannot effectively handle such diversity with extremely limited demonstrations.

We argue that the number of demonstrations required for successful behavior-targeted attacks depends on the environment's characteristics. In environments with deterministic state transitions and initial state distributions, where the target policy exhibits similar behavior across episodes, fewer demonstrations may suffice. Conversely, environments with more randomness in state transitions and initial state distributions require more demonstrations to ensure proper generalization of the discriminator.

## G.2  ATTACK BUDGET EFFICIENCY

In this section, we present a more comprehensive analysis of the attack efficiency of BIA-ILfD/ILfO. We conduct experiments across different attack budgets $\epsilon$ = 0.05, 0.1, 0.2, 0.3, 0.4, 0.5 and evaluate the attack rewards. The victim is a vanilla PPO agent without any defense methods. All experimental parameters remain consistent with those in Section 7, and the results for $\epsilon$ = 0.3 correspond to those presented in Table 1.

The experimental results are shown in Figure 4. Across all tasks, we observe that attack rewards increase proportionally as the attack budget increases. Notably, in the window-close and drawer-close tasks, the attack rewards nearly match the Target Rewards when given larger attack budgets, indicating that the attack successfully guides the victim to almost perfectly replicate the target behavior.

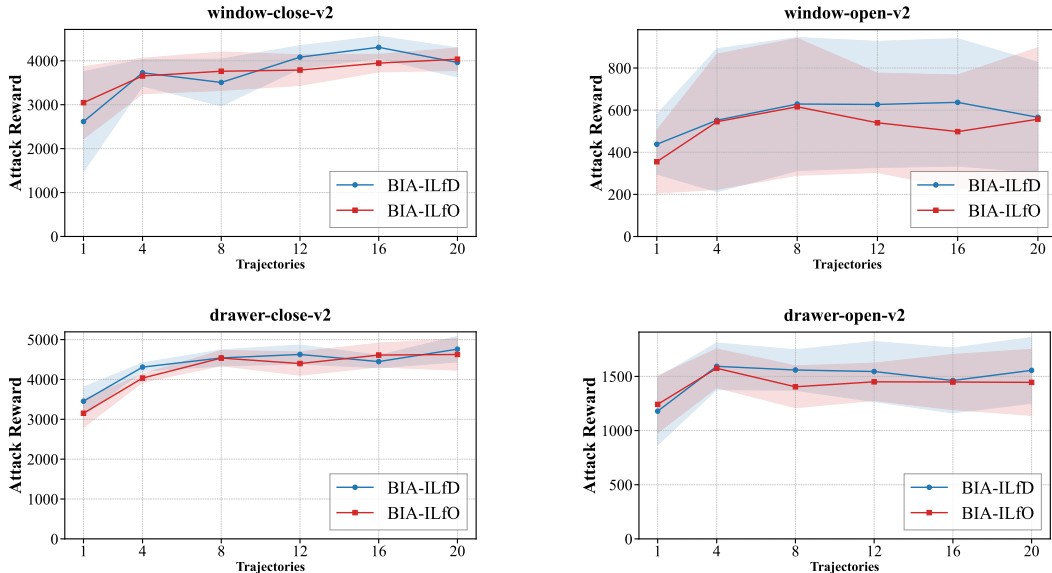

Figure 3: Attack performance of BIA-ILfD/ILfO with varying amounts of demonstrations. The x-axis shows the number of demonstration episodes, and the y-axis represents the attack reward. The attack budget $\epsilon = 0.3$. Each environment name represents an adversarial task. The solid line and shaded area denote the mean and the standard deviation / 2 over 50 episodes.

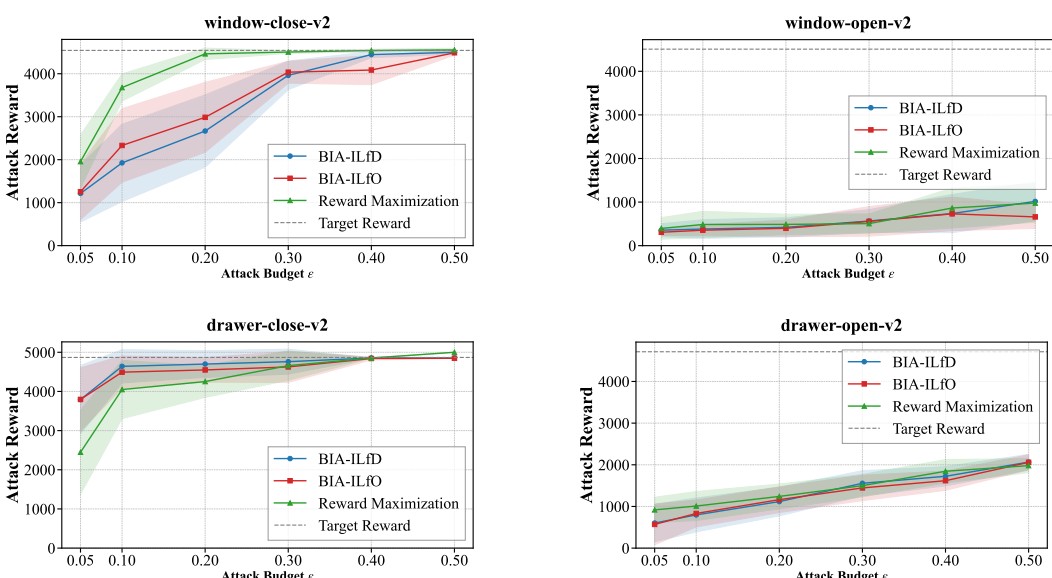

Figure 4: Attack performance of BIA-ILfD/ILfO with varying attack budget $\epsilon$. The x-axis shows the value of the attack budget, and the y-axis represents the attack reward. The target reward represents the cumulative reward obtained by the target policy and serves as the upper bound for the attack rewards of BIA-ILfD/ILfO. Each environment name represents an adversarial task. The solid line and shaded area denote the mean and the standard deviation / 2 over 50 episodes.

In the window-close and drawer-close tasks, where the attacks are particularly successful, we observed an interesting phenomenon. When the attack budget is small, there are high variances in attack rewards, but this variance decreases as the attack budget increases. The high variances indicate that among the 50 evaluation episodes, some attacks achieve perfect success while others completely fail. This finding suggests that the initial state significantly influences the attack success rate. We

Table 9: Comparison of attack and defense performances for the setting where the state is represented as an image in MiniGrid. Each value represents the average episode reward $\pm$ standard deviation over 50 episodes. A higher attack reward indicates higher attack performance and lower defense effectiveness. Each parenthesis indicates (*Access model*, *Adversary's knowledge*).

| Task | Target Reward | Attack Reward (BIA-ILfD) |
|---|---|---|
| window-close | $803 \pm_{129}$ | $777 \pm_{106}$ |
| | $3389 \pm_{219}$ | $3298 \pm_{396}$ |
| | $4543 \pm_{39}$ | $3962 \pm_{666}$ |
| window-open | $461 \pm_{514}$ | $358 \pm_{318}$ |
| | $2076 \pm_{1599}$ | $585 \pm_{439}$ |
| | $4508 \pm_{121}$ | $566 \pm_{523}$ |
| drawer-close | $1029 \pm_{864}$ | $1470 \pm_{1016}$ |
| | $3480 \pm_{1744}$ | $4252 \pm_{9}$ |
| | $4868 \pm_{6}$ | $4760 \pm_{640}$ |
| drawer-open | $1175 \pm_{510}$ | $1278 \pm_{637}$ |
| | $2544 \pm_{89}$ | $1169 \pm_{645}$ |
| | $4713 \pm_{16}$ | $1556 \pm_{607}$ |

Table 10: Comparison of attack performances. Each value represents the average attack success rate (ASR) over 50 episodes. Each parenthesis indicates (*Access model*, *Adversary's knowledge*). We set the attack budget $\epsilon = 0.3$.

| Task | Targeted PGD (white-box) | Rew Max (PA-AD) (black-box) | Rew Max (SA-RL) (black-box) | BIA-ILfD (ours) (black-box) | BIA-ILfO (ours) (no-box) | Random (no-box) |
|---|---|---|---|---|---|---|
| window-close | 0.08 | 0.82 | **1.00** | 0.72 | 0.74 | 0.00 |
| window-open | 0.08 | 0.02 | 0.06 | **0.12** | 0.06 | 0.00 |
| drawer-close | 0.68 | 0.68 | **1.00** | **1.00** | **1.00** | 0.22 |
| drawer-open | 0.00 | 0.00 | 0.00 | 0.00 | 0.00 | 0.00 |
| faucet-close | 0.00 | 0.04 | **0.68** | **0.68** | 0.60 | 0.00 |
| faucet-open | 0.00 | 0.00 | 0.00 | **0.12** | 0.08 | 0.00 |
| handle-press-side | 0.16 | **1.00** | 0.98 | **1.00** | **1.00** | 0.02 |
| handle-pull-side | 0.06 | 0.70 | 0.92 | **1.00** | **1.00** | 0.00 |
| door-lock | 0.04 | 0.04 | 0.46 | **0.50** | 0.42 | 0.00 |
| door-unlock | 0.00 | 0.62 | **0.68** | 0.64 | 0.62 | 0.00 |

hypothesize that this occurs because certain initial states require the victim to perform actions that are rarely selected in their normal behavior, making the attack more challenging in these scenarios.

### G.3 RELATION BETWEEN THE PERFORMANCE OF THESE DEMONSTRATIONS AND THE ATTACKING PERFORMANCE.

In this section, we evaluate how the performance of the target policy influences the attack performance of BIA. We use three types of target policies. Each target policy achieved a different target reward. All other settings remain the same as described in the experimental section.

Table 9 shows the results. We confirm that lower-performing target policies lead to lower attack performance. This is because the adversarial policy is not trained to maximize the attack reward but rather to mimic the target policy. However, even if the target policy is suboptimal, selecting one that closely resembles the victim's original behavior may still lead to strong attack performance.

### G.4 EVALUATING ATTACK PERFORMANCE WITH ASR

In this section, we evaluate the performance of the attack methods using the attack success rate (ASR). Success is determined by the task success flag provided by the Meta-World environment. For each setting, we run 50 episodes and report ASR as the fraction of episodes marked as successful. All other experimental settings follow the main text.

The results are shown in Table 10. Consistent with the evaluation based on episode reward, the results indicate that BIA is effective, supporting that the attack performance of our proposed method generalizes across evaluation metrics.

## G.5 COMPUTATIONAL COST OF TRAINING ADVERSARIAL POLICIES

In this section, we compare the computational cost of different adversarial attack methods. We divide computational cost into two components: *training-time cost* and *test-time cost*. This distinction is important because different attack families place the computational burden at different phases. For example, BIA requires training an adversarial policy, but its test-time execution is simply a single forward pass. In contrast, Targeted PGD requires no training, but incurs optimization overhead at every timestep during test-time rollout. Table 11 summarizes the computational cost measured on a single NVIDIA H100 GPU.

Table 11: Comparison of computational cost.

|               | Targeted PGD     | Rew Max (PA-AD) | Rew Max (SA-RL) | BIA-ILfD (ours) | BIA-ILfO (ours) |
| ------------- | ---------------- | --------------- | --------------- | --------------- | --------------- |
| Training time | –                | 3.6 h           | 4.3 h           | 6.2 h           | 6.1 h           |
| Test time     | 37 sec / episode | –               | –               | –               | –               |

**BIA vs. reward-based white-box attacks (Rew Max).** BIA has a higher training cost than Rew Max (PA-AD). This is mainly because imitation learning–based BIA must train both a policy and a discriminator, while Rew Max only optimizes the adversarial policy. On the other hand, BIA only requires demonstrations of the target behavior, whereas Rew Max assumes access to the target reward function, which is often harder to obtain in practice.

**BIA vs. Targeted PGD.** As described above, Targeted PGD has no training-time cost, but incurs a test-time optimization cost. When many episodes need to be attacked (e.g., long deployments or large-scale evaluation), BIA's training cost is spread out over many episodes, and BIA becomes more computationally efficient overall than repeatedly running Targeted PGD.

## H ADDITIONAL ANALYSIS IN DEFENSE METHODS

### H.1 FULL RESULTS IN DEFENSE PERFORMANCE EVALUATION

In our evaluation of defense methods in Section 7, we only provide the results of the best attack. In this section, we present the results of all attacks in Table 12. Targeted PGD is ineffective against robustly trained victims. We observed a trend where the Reward Maximization Attack tended to be slightly more effective than BIA when attacking highly robust victims. When the victim's robustness is high, it becomes difficult to make the victim behave like the target policy, which may cause the learning process in BIA to fail or collapse. On the other hand, in the Reward Maximization Attack, the reward serves as a good guidepost, allowing learning to proceed even when the victim's robustness is high.

### H.2 IMPACT OF REGULARIZATION COEFFICIENT

We analyze how policy smoothing affects both robustness and original performance. The strength of the regularization is determined by the coefficient $\lambda$. Accordingly, we evaluate the clean rewards and best attack rewards for SA-PPO and TDRT-PPO with $\lambda \in \{0.03, 0.1, 0.2, 0.3, 0.5\}$. The results are shown in Table 13, and all other settings follow those in Section 7 (where $\lambda$=0.3).

**Robustness.** For $\lambda$=0.5, the best attack rewards remain unchanged compared to $\gamma$=0.3 for both TDRT-PPO and SA-PPO, indicating that $\lambda$=0.3 provides sufficient regularization, and that further increasing it does not enhance robustness. Conversely, as $\lambda$ decreases, the best attack rewards decline. Notably, TDRT-PPO fails to maintain robustness when $\lambda$=0.03, highlighting the need for a certain level of regularization to achieve robust performance.

Table 12: Comparison of Defense Methods. Each value is the average episode rewards $\pm$ standard deviation over 50 episodes. Clean Rewards are the rewards for the victim's tasks (no attack). The best attack reward is the highest reward among the five types of adversarial attacks. The attack budget is set to $\epsilon = 0.3$.

| Adv Task | Methods | Clean Rewards (↑) | Attack Rewards (↓) | | | | | | |
|---|---|---|---|---|---|---|---|---|---|
| | | | Random | Targeted PGD | Rew Max (SA-RL) | Rew Max (PA-AD) | BIA-ILfD (ours) | BIA-ILfO (ours) | Best Attack |
| window-close | PPO | 4508 ± 121 | 947 ± 529 | 1666 ± 936 | 4505 ± 65 | 4255 ± 300 | 3962 ± 666 | 4036 ± 510 | 4505 ± 65 |
| | ATLA-PPO | 4169 ± 467 | 1706 ± 1097 | 2028 ± 1387 | 4270 ± 188 | 4378 ± 319 | 3063 ± 1515 | 2564 ± 787 | 4270 ± 188 |
| | PA-ATLA-PPO | 4353 ± 89 | 482 ± 3 | 483 ± 3 | 4041 ± 96 | 3978 ± 193 | 2183 ± 567 | 1932 ± 663 | 4041 ± 96 |
| | RAD-PPO | 4432 ± 80 | 511 ± 77 | 626 ± 49 | 4261 ± 208 | 2724 ± 1237 | 3704 ± 373 | 3018 ± 789 | 4261 ± 208 |
| | WocaR-PPO | 2879 ± 1256 | 480 ± 5 | 480 ± 5 | 575 ± 135 | 481 ± 5 | 381 ± 30 | 403 ± 29 | 575 ± 135 |
| | SA-PPO | 4367 ± 107 | 478 ± 5 | 477 ± 5 | 485 ± 61 | 478 ± 5 | 21 ± 12 | 21 ± 12 | 485 ± 61 |
| | TDRT-PPO (ours) | 4412 ± 55 | 422 ± 56 | 409 ± 44 | 482 ± 3 | 429 ± 56 | 376 ± 43 | 377 ± 43 | **482 ± 3** |
| window-open | PPO | 4543 ± 39 | 322 ± 261 | 515 ± 651 | 506 ± 444 | 493 ± 562 | 566 ± 523 | 557 ± 679 | 566 ± 523 |
| | ATLA-PPO | 4566 ± 80 | 354 ± 257 | 319 ± 250 | 547 ± 611 | 406 ± 349 | 586 ± 649 | 532 ± 444 | 586 ± 649 |
| | PA-ATLA-PPO | 4332 ± 109 | 397 ± 59 | 224 ± 65 | 671 ± 589 | 452 ± 492 | 521 ± 712 | 524 ± 673 | 671 ± 589 |
| | RAD-PPO | 4269 ± 202 | 305 ± 73 | 302 ± 19 | 501 ± 132 | 338 ± 162 | 454 ± 226 | 493 ± 356 | 501 ± 132 |
| | WocaR-PPO | 3645 ± 1575 | 287 ± 24 | 287 ± 24 | 295 ± 18 | 289 ± 22 | 247 ± 64 | 253 ± 60 | 295 ± 18 |
| | SA-PPO | 4092 ± 461 | 259 ± 46 | 258 ± 47 | 272 ± 37 | 259 ± 46 | 200 ± 57 | 208 ± 60 | 272 ± 37 |
| | TDRT-PPO (ours) | 4383 ± 57 | 213 ± 54 | 213 ± 54 | 229 ± 54 | 217 ± 55 | 254 ± 214 | 229 ± 137 | **254 ± 214** |
| drawer-close | PPO | 4714 ± 16 | 1069 ± 1585 | 2891 ± 150 | 4658 ± 747 | 3768 ± 1733 | 4760 ± 640 | 4626 ± 791 | 4760 ± 640 |
| | ATLA-PPO | 4543 ± 102 | 1004 ± 892 | 962 ± 1532 | 4858 ± 6 | 4858 ± 11 | 3919 ± 1808 | 3919 ± 1808 | 4858 ± 6 |
| | PA-ATLA-PPO | 4543 ± 102 | 1204 ± 535 | 1434 ± 898 | 4858 ± 6 | 4868 ± 3 | 4865 ± 3 | 4865 ± 3 | 4868 ± 3 |
| | RAD-PPO | 4865 ± 5 | 868 ± 424 | 928 ± 678 | 2935 ± 2163 | 2106 ± 2126 | 4588 ± 923 | 2704 ± 2285 | 4588 ± 923 |
| | WocaR-PPO | 4193 ± 304 | 562 ± 1335 | 834 ± 1635 | 3654 ± 1976 | 1738 ± 2091 | 4867 ± 8 | 4838 ± 22 | 4867 ± 8 |
| | SA-PPO | 2156 ± 453 | 3 ± 1 | 3 ± 1 | 3 ± 1 | 3 ± 1 | 4 ± 2 | 4 ± 2 | 4 ± 2 |
| | TDRT-PPO (ours) | 4237 ± 93 | 1143 ± 1779 | 667 ± 1620 | 4770 ± 1 | 1498 ± 1965 | 4860 ± 4 | 4860 ± 4 | 4860 ± 4 |
| drawer-open | PPO | 4868 ± 6 | 841 ± 357 | 953 ± 450 | 1499 ± 536 | 1607 ± 355 | 1556 ± 607 | 1445 ± 610 | 1556 ± 607 |
| | ATLA-PPO | 4863 ± 7 | 464 ± 270 | 421 ± 129 | 1158 ± 1026 | 650 ± 518 | 831 ± 653 | 741 ± 561 | 1158 ± 1026 |
| | PA-ATLA-PPO | 4867 ± 7 | 434 ± 93 | 398 ± 9 | 954 ± 219 | 937 ± 251 | 752 ± 358 | 748 ± 343 | 954 ± 219 |
| | RAD-PPO | 4151 ± 489 | 441 ± 19 | 455 ± 10 | 727 ± 21 | 651 ± 25 | 736 ± 22 | 728 ± 13 | 736 ± 22 |
| | WocaR-PPO | 4704 ± 654 | 410 ± 9 | 405 ± 8 | 579 ± 15 | 515 ± 9 | 446 ± 28 | 442 ± 22 | 579 ± 15 |
| | SA-PPO | 4161 ± 1537 | 368 ± 9 | 368 ± 9 | 368 ± 9 | 368 ± 9 | 403 ± 49 | 403 ± 49 | 403 ± 49 |
| | TDRT-PPO (ours) | 4802 ± 27 | 378 ± 10 | 378 ± 10 | 378 ± 10 | 378 ± 10 | 357 ± 4 | 357 ± 4 | **378 ± 10** |
| faucet-close | PPO | 4544 ± 800 | 897 ± 171 | 1092 ± 192 | 3409 ± 652 | 1241 ± 501 | 3316 ± 648 | 3041 ± 502 | 3409 ± 652 |
| | ATLA-PPO | 4756 ± 18 | 1406 ± 118 | 1727 ± 137 | 3872 ± 732 | 3907 ± 726 | 4108 ± 790 | 4058 ± 791 | 4108 ± 790 |
| | PA-ATLA-PPO | 3716 ± 802 | 1278 ± 210 | 1562 ± 137 | 4012 ± 123 | 3292 ± 833 | 1746 ± 165 | 1827 ± 72 | 4012 ± 123 |
| | RAD-PPO | 4737 ± 46 | 1756 ± 169 | 1871 ± 159 | 2235 ± 528 | 1938 ± 486 | 1757 ± 48 | 1749 ± 74 | 2235 ± 528 |
| | WocaR-PPO | 3323 ± 974 | 1604 ± 743 | 1686 ± 770 | 2829 ± 1264 | 2115 ± 1171 | 2177 ± 931 | 2092 ± 1056 | 2829 ± 1264 |
| | SA-PPO | 4304 ± 42 | 1253 ± 372 | 1351 ± 295 | 1559 ± 406 | 1307 ± 397 | 457 ± 26 | 445 ± 31 | 1559 ± 406 |
| | TDRT-PPO (ours) | 4740 ± 17 | 1297 ± 743 | 1432 ± 507 | 1789 ± 610 | 1618 ± 750 | 1169 ± 243 | 1218 ± 262 | 1789 ± 610 |
| faucet-open | PPO | 4754 ± 15 | 1372 ± 81 | 2514 ± 86 | 1448 ± 64 | 1420 ± 35 | 3031 ± 1493 | 2718 ± 1293 | 3031 ± 1493 |
| | ATLA-PPO | 4742 ± 30 | 1231 ± 195 | 2729 ± 12 | 3952 ± 732 | 4383 ± 449 | 3695 ± 874 | 2736 ± 758 | 4383 ± 449 |
| | PA-ATLA-PPO | 3767 ± 10 | 1613 ± 555 | 2832 ± 357 | 1477 ± 184 | 1345 ± 206 | 2358 ± 976 | 1874 ± 341 | 2358 ± 976 |
| | RAD-PPO | 4713 ± 111 | 1598 ± 1045 | 1338 ± 87 | 4254 ± 625 | 3548 ± 1130 | 3133 ± 699 | 2924 ± 1148 | 4254 ± 625 |
| | WocaR-PPO | 3756 ± 16 | 2101 ± 1006 | 2693 ± 1533 | 2885 ± 1340 | 2465 ± 1196 | 2997 ± 1307 | 3012 ± 1301 | 3012 ± 1301 |
| | SA-PPO | 4380 ± 43 | 1582 ± 140 | 1690 ± 290 | 1763 ± 256 | 1635 ± 199 | 258 ± 25 | 257 ± 25 | 1763 ± 256 |
| | TDRT-PPO (ours) | 4630 ± 11 | 1469 ± 158 | 1881 ± 555 | 1942 ± 261 | 1554 ± 170 | 306 ± 21 | 308 ± 21 | 1942 ± 261 |
| handle-press-side | PPO | 4442 ± 732 | 1865 ± 1340 | 1994 ± 1225 | 4625 ± 175 | 4726 ± 175 | 4631 ± 408 | 4627 ± 586 | 4726 ± 175 |
| | ATLA-PPO | 4831 ± 29 | 1961 ± 1689 | 2243 ± 2071 | 4289 ± 852 | 4225 ± 757 | 3185 ± 1427 | 4302 ± 799 | 4302 ± 799 |
| | PA-ATLA-PPO | 4757 ± 71 | 1210 ± 611 | 2211 ± 1748 | 3318 ± 1539 | 1324 ± 1385 | 1638 ± 1924 | 1641 ± 1939 | 3318 ± 1539 |
| | RAD-PPO | 4725 ± 606 | 524 ± 814 | 1764 ± 1592 | 2375 ± 1440 | 927 ± 1340 | 833 ± 998 | 824 ± 597 | 2375 ± 1440 |
| | WocaR-PPO | 3724 ± 83 | 1673 ± 811 | 1784 ± 983 | 3042 ± 1193 | 2893 ± 1742 | 2184 ± 892 | 1984 ± 2132 | 3042 ± 1193 |
| | SA-PPO | 3226 ± 806 | 817 ± 1347 | 506 ± 1044 | 1051 ± 1627 | 978 ± 1527 | 1619 ± 2099 | 1888 ± 1169 | 1888 ± 1169 |
| | TDRT-PPO (ours) | 4321 ± 215 | 702 ± 515 | 891 ± 428 | 4067 ± 942 | 1215 ± 1386 | 1799 ± 1715 | 1928 ± 736 | 1928 ± 736 |
| handle-pull-side | PPO | 4546 ± 721 | 1426 ± 1617 | 2198 ± 1524 | 3617 ± 1363 | 2065 ± 1501 | 4268 ± 740 | 4193 ± 517 | 4268 ± 740 |
| | ATLA-PPO | 4608 ± 68 | 482 ± 424 | 534 ± 438 | 532 ± 534 | 157 ± 668 | 482 ± 1069 | 278 ± 969 | 532 ± 534 |
| | PA-ATLA-PPO | 3634 ± 1993 | 492 ± 783 | 483 ± 54 | 428 ± 324 | 232 ± 574 | 512 ± 982 | 382 ± 862 | 512 ± 982 |
| | RAD-PPO | 4480 ± 117 | 487 ± 342 | 564 ± 783 | 464 ± 1044 | 191 ± 453 | 1086 ± 1256 | 892 ± 1345 | 1086 ± 1256 |
| | WocaR-PPO | 3482 ± 432 | 5 ± 1 | 7 ± 1 | 33 ± 6 | 31 ± 7 | 4 ± 1 | 4 ± 1 | 33 ± 6 |
| | SA-PPO | 4094 ± 350 | 7 ± 0 | 10 ± 0 | 10 ± 1 | 7 ± 0 | 3 ± 0 | 3 ± 1 | 10 ± 1 |
| | TDRT-PPO (ours) | 4468 ± 126 | 30 ± 5 | 30 ± 6 | 7 ± 1 | 6 ± 1 | 5 ± 1 | 5 ± 1 | 7 ± 1 |
| door-lock | PPO | 4690 ± 33 | 589 ± 494 | 640 ± 664 | 1937 ± 1186 | 763 ± 769 | 2043 ± 1229 | 1906 ± 1045 | 2043 ± 1229 |
| | ATLA-PPO | 3790 ± 80 | 488 ± 25 | 977 ± 535 | 612 ± 584 | 594 ± 469 | 907 ± 895 | 1020 ± 805 | 1020 ± 805 |
| | PA-ATLA-PPO | 2385 ± 1211 | 486 ± 13 | 487 ± 14 | 721 ± 396 | 517 ± 202 | 893 ± 36 | 992 ± 19 | 992 ± 19 |
| | RAD-PPO | 2973 ± 1328 | 488 ± 10 | 489 ± 16 | 632 ± 298 | 712 ± 392 | 689 ± 123 | 593 ± 131 | 712 ± 392 |
| | WocaR-PPO | 2420 ± 1415 | 461 ± 7 | 461 ± 7 | 561 ± 7 | 561 ± 7 | 562 ± 14 | 562 ± 14 | 562 ± 14 |
| | SA-PPO | 2299 ± 1491 | 479 ± 8 | 479 ± 7 | 482 ± 8 | 480 ± 8 | 478 ± 8 | 478 ± 8 | 478 ± 8 |
| | TDRT-PPO (ours) | 2769 ± 1411 | 461 ± 21 | 477 ± 9 | 481 ± 10 | 471 ± 10 | 487 ± 11 | 487 ± 9 | 487 ± 11 |
| door-unlock | PPO | 3845 ± 79 | 391 ± 59 | 531 ± 61 | 3421 ± 974 | 3295 ± 1111 | 3336 ± 932 | 3123 ± 1123 | 3421 ± 974 |
| | ATLA-PPO | 4561 ± 283 | 695 ± 645 | 1166 ± 1372 | 3277 ± 1265 | 1550 ± 1225 | 3163 ± 1238 | 3163 ± 1186 | 3277 ± 1265 |
| | PA-ATLA-PPO | 4468 ± 323 | 875 ± 460 | 886 ± 473 | 2806 ± 1437 | 1819 ± 1456 | 2433 ± 1461 | 2247 ± 1488 | 2806 ± 1437 |
| | RAD-PPO | 3773 ± 56 | 635 ± 497 | 1694 ± 1412 | 2086 ± 1124 | 2482 ± 1643 | 2743 ± 1386 | 2562 ± 1788 | 2743 ± 1386 |
| | WocaR-PPO | 2545 ± 291 | 728 ± 171 | 761 ± 161 | 1073 ± 161 | 1044 ± 165 | 885 ± 120 | 928 ± 97 | 1073 ± 161 |
| | SA-PPO | 2017 ± 497 | 505 ± 123 | 513 ± 130 | 514 ± 133 | 509 ± 130 | 713 ± 1135 | 787 ± 1001 | 787 ± 1001 |
| | TDRT-PPO (ours) | 3680 ± 290 | 620 ± 212 | 712 ± 371 | 691 ± 360 | 660 ± 301 | 411 ± 60 | 402 ± 37 | 691 ± 360 |

Table 13: Comparison between TDRT-PPO and SA-PPO with different $\lambda$ values. Each value is the average episode rewards $\pm$ standard deviation over 50 episodes. Clean Rewards are the rewards for the victim's tasks (no attack). Best attack rewards represent the highest attack reward among the five types of adversarial attacks. The attack budget is set to $\epsilon = 0.3$. **In SA-PPO, which does not apply time discounting, ensuring sufficient robustness leads to a significant drop in performance on the original task. In contrast, TDRT-PPO, which applies time discounting, achieves high robustness while preserving the original-task performance.**

| Adv Task | $\lambda$ | TDRT-PPO (ours) | | SA-PPO | |
|---|---|---|---|---|---|
| | | Clean Rewards ($\uparrow$) | Best Attack Rewards ($\downarrow$) | Clean Rewards ($\uparrow$) | Best Attack Rewards ($\downarrow$) |
| window-close | 0.03 | **4500** $\pm$ **20** | 1853 $\pm$ 241 | 4482 $\pm$ 21 | **1452** $\pm$ **512** |
| | 0.1 | **4512** $\pm$ **12** | **712** $\pm$ **46** | 4324 $\pm$ 76 | 987 $\pm$ 62 |
| | 0.2 | **4403** $\pm$ **34** | 512 $\pm$ 4 | 4218 $\pm$ 129 | **472** $\pm$ **51** |
| | 0.3 | **4412** $\pm$ **55** | **482** $\pm$ **3** | 4367 $\pm$ 103 | 485 $\pm$ 61 |
| | 0.5 | **4351** $\pm$ **130** | **495** $\pm$ **9** | 4041 $\pm$ 293 | 499 $\pm$ 3 |
| window-open | 0.03 | **4512** $\pm$ **38** | 489 $\pm$ 526 | 4483 $\pm$ 23 | **401** $\pm$ **391** |
| | 0.1 | **4430** $\pm$ **47** | 397 $\pm$ 219 | 4219 $\pm$ 76 | **253** $\pm$ **31** |
| | 0.2 | **4403** $\pm$ **52** | **253** $\pm$ **321** | 4198 $\pm$ 87 | 284 $\pm$ 21 |
| | 0.3 | **4383** $\pm$ **57** | **254** $\pm$ **214** | 4092 $\pm$ 461 | 272 $\pm$ 32 |
| | 0.5 | **4313** $\pm$ **59** | **263** $\pm$ **298** | 4015 $\pm$ 212 | 268 $\pm$ 23 |
| drawer-close | 0.03 | 4610 $\pm$ 81 | **4660** $\pm$ **240** | **4709** $\pm$ **99** | 4792 $\pm$ 4 |
| | 0.1 | **4398** $\pm$ **72** | **4592** $\pm$ **414** | 4129 $\pm$ 498 | 4809 $\pm$ 9 |
| | 0.2 | **4442** $\pm$ **44** | 4890 $\pm$ 4 | 2183 $\pm$ 572 | **5** $\pm$ **3** |
| | 0.3 | **4237** $\pm$ **93** | 4860 $\pm$ 4 | 2156 $\pm$ 453 | **4** $\pm$ **2** |
| | 0.5 | **4184** $\pm$ **104** | 4592 $\pm$ 4 | 1952 $\pm$ 629 | **4** $\pm$ **2** |
| drawer-open | 0.03 | **4818** $\pm$ **9** | 1098 $\pm$ 192 | 4799 $\pm$ 42 | **809** $\pm$ **210** |
| | 0.1 | **4860** $\pm$ **1** | **792** $\pm$ **94** | 4801 $\pm$ 31 | 823 $\pm$ 194 |
| | 0.2 | **4843** $\pm$ **7** | **394** $\pm$ **10** | 4766 $\pm$ 31 | 670 $\pm$ 79 |
| | 0.3 | **4802** $\pm$ **27** | **378** $\pm$ **10** | 4161 $\pm$ 1537 | 403 $\pm$ 49 |
| | 0.5 | **4839** $\pm$ **25** | 413 $\pm$ 12 | 3984 $\pm$ 76 | **405** $\pm$ **24** |

**Original Performance.** Both TDRT-PPO and SA-PPO experience a slight drop in performance as $\lambda$ increases, likely because stronger regularization reduces the expressiveness of the policy. For SA-PPO, lowering $\lambda$ improves performance, suggesting that $\lambda$=0.3 may be overly stringent and that reducing it helps recover the policy's representational capacity.

Crucially, while SA-PPO avoids performance degradation at weaker regularization levels, it fails to achieve sufficient robustness in those settings. In contrast, TDRT-PPO can remain robust without compromising original performance. These findings indicate that time discounting in TDRT effectively curtails behavioral shifts throughout the entire episode.

## H.3 TRAINING TIME EFFICIENCY

We compare the training time of TDRT-PPO in the window-close and window-open tasks with ATLA-PPO, PA-ATLA-PPO, RAD-PPO, WocaR-PPO, and SA-PPO. For all methods, training is conducted for about 3,000,000 steps. To ensure a fair comparison, we use an Nvidia H100 Tensor Core GPU for the training of all methods.

Table 14: Comparison of training time on window-close and window-open tasks. For all methods, training is conducted for 3,000,000 steps. Each value represents the training time in hours.

| Task | Method | | | | | | |
|---|---|---|---|---|---|---|---|
| | Vanilla PPO | ATLA-PPO | PA-ATLA-PPO | RAD-PPO | WocaR-PPO | SA-PPO | TDRT-PPO (ours) |
| window-close | 2.2h | 7.8h | 8.0h | 4.9h | 6.0h | 3.8h | 4.5h |
| window-open | 2.0h | 7.7h | 8.5h | 5.0h | 7.1h | 4.2h | 4.6h |

Table 14 shows the training time of each method for each task. TDRT-PPO shows superior time efficiency compared to other methods. However, its training cost is higher than that of SA-PPO. This difference is probably due to the fact that TDRT-PPO requires recording the time step of each state in collecting experience.

ATLA-PPO and PA-ATLA-PPO use adversarial training methods. As a result, their training process requires learning not only a robust agent but also an adversarial agent, leading to significantly higher training costs. RAD-PPO incurs higher training costs than vanilla PPO due to the need to compute an approximate minimum reward for regret computation.

In addition, WocaR-PPO must estimate both the regularization term for policy smoothness and the worst-case value. While these computations are not excessively costly, the training cost increases compared to SA-PPO and TDRT-PPO, which only compute the policy smoothness regularization term. SA-PPO achieves a lower training cost because it focuses only on calculating the regularization term for policy smoothness.

# I IMPLEMENTATION DETAILS

In this section, we provide a detailed explanation of the implementation.

## I.1 ENVIRONMENT DETAILS

We conducted the experiments described in Section 7 using Meta-World(Yu et al., 2020), a benchmark that simulates robotic arm manipulation. All tasks in Meta-World share a common 39-dimensional continuous state space and a 4-dimensional continuous action space. In our experiments, we use four tasks: window-close, window-open, drawer-close, drawer-open, faucet-close, faucet-open, handle-press-side, handle-pull-side, door-lock, and door-unlock. The objective of each task is to move a specific object to a designated position.

**Reward Design** The reward design is specific to each task in Meta-World. For example, the reward functions for window-close and window-open tasks are designed independently. Therefore, when attacking a victim trained on the window-close task to perform a window-open task, this differs from an untargeted attack since it does not simply minimize the victim's reward.

## I.2 DEFENSE BASELINE DETAILS

This section provides a detailed explanation of the baseline defense methods used in our experiments.

**(i) SA-PPO** (Zhang et al., 2020b): This method aims to increase the smoothness of the policy's action outputs by a regularizer. *The difference between TDRT-PPO and SA-PPO is that SA-PPO does not apply time discounting in the regularization.*

**(ii) ATLA-PPO** (Zhang et al., 2021): ATLA-PPO is an adversarial training method that alternates between training a victim and an adversary. The adversary is trained to generate perturbations that produce the worst-case cumulative reward for the victim by leveraging the SA-MDP framework. The agent is then trained to optimize its policy against this strong adversary, resulting in improved robustness to adversarial attacks.

**(iii) PA-ATLA-PPO** (Sun et al., 2022): PA-ATLA-PPO extends ATLA-PPO by integrating the Policy Adversarial Actor Director (PA-AD) framework, which improves adversarial attack generation. PA-AD utilizes a "director" to determine optimal policy perturbation directions and an "actor" to generate corresponding state perturbations, ensuring efficient and theoretically optimal adversarial attacks. Unlike ATLA-PPO, which trains an adversary directly using reinforcement learning, PA-ATLA-PPO's decoupled approach enhances efficiency and scalability in environments with large state spaces.

**(iv) WocaR-PPO** (Liang et al., 2022): WocaR-PPO trains the policy to maximize the worst-case cumulative reward under adversarial attacks. Unlike adversarial training methods such as ATLA-PPO, which involve learning alongside an adversary, WocaR-PPO uses a computationally feasible approach to estimate the worst-case cumulative reward without requiring additional interaction with the environment. Additionally, regularization is applied to improve the smoothness of the policy, focusing specifically on critical states where significant reward drops are likely to occur based on state importance weights.

**(v) RAD-PPO** (Belaire et al., 2024): RAD-PPO is a regret-based defense approach. Regret represents the difference between the value without an attack and the value under an attack. RAD-PPO aims to achieve robustness against adversarial attacks by learning a policy that minimizes regret at each step. Since regret is defined based on the rewards obtained by the victim, regret-based defense methods primarily assume untargeted adversaries. Therefore, the robustness of this approach cannot be fully guaranteed against the behavior-targeted attack. It is worth noting that the implementation code for RAD-PPO is not publicly available, so we implemented it ourselves based on the details provided in the paper.

## I.3 HYPERPARAMETER DETAILS

In this section, we discuss the hyperparameters used in our experiments. In general, our hyperparameter settings follow (Zhang et al., 2021).

**Architecture.** For TDRT-PPO and all defense baselines, we used 3-layer MLPs with hidden layer sizes of [256, 256] as the policy network. Similarly, 3-layer MLPs with [256, 256] were used as the Discriminator network for training BIA-ILfD/ILfO. This configuration is commonly employed in imitation learning.

**Parameter Search.** We conducted hyperparameter tuning for the victim agents using grid search. Specifically, we explored the following parameter ranges and selected the models that achieved the highest clean reward (cumulative reward for the original task in the absence of attacks): policy learning rate: {1e-3, 3e-4, 1e-4, 3e-5}, value function learning rate: {1e-3, 3e-4, 1e-4, 3e-5}, entropy coefficient: 1e-5, 0. For RAD-PPO and WocaR-PPO, which require training a Q-function, we also searched the Q-function learning rate within the range {0.0004, 0.004, 0.00004}. Regarding attack methods, during the training of BIA-ILfD/ILfO, we explored the following ranges: adversarial policy learning rate: {1e-5, 3e-5, 1e-4, 2e-4, 3e-4}, discriminator learning rate: {1e-5, 3e-5, 5e-5, 1e-4, 2e-4, 3e-4}. We observed that if the balance between policy learning and discriminator learning deteriorates, attack performance significantly decreases. Thus, hyperparameter tuning for BIA-ILfD/ILfO is crucial for achieving high attack performance. Furthermore, for the Target Reward Maximization Attack, we explored the following ranges for adversarial policy training: policy learning rate: {1e-3, 3e-4, 1e-4, 3e-5}, value function learning rate: {1e-3, 3e-4, 1e-4, 3e-5}. We select the models that recorded the highest attack reward (cumulative reward for adversarial tasks under attack).

## I.4 THE DETAILS OF TARGETED PGD ATTACK

In this section, we provide a detailed explanation of targeted PGD attacks and present additional experiments. In Section I.4.1, we show the pseudo-code for targeted PGD attacks and provide specific details about their implementation. In Section I.4.2, we conduct experiments with various attack budgets and analyze the results to gain deeper insights into targeted PGD attacks.

### I.4.1 IMPLEMENTATION DETAILS OF THE TARGETED PGD ATTACK

We present the algorithm used to optimize the false state at each step for the targeted PGD attack in Algorithm 3. The attack aims to find optimal false states that minimize the difference between the victim's action and the target policy's action at each step. The algorithm performs $T$ iterations, during which it updates states using FGSM in each iteration. Specifically, it uses the L2 distance between current victim actions and target actions as the loss function and updates states by scaling in the direction of gradient signs by $\epsilon_{\text{step}}$. In all experiments, we set $T = 30$. We also implement random initialization for stable optimization.

### I.4.2 PERFORMANCE ANALYSIS OF TARGETED PGD ATTACKS UNDER DIFFERENT ATTACK BUDGETS

We evaluate the performance of targeted PGD attacks across various attack budgets $\epsilon$. We conduct attacks against victims trained with PPO without any defense method, using $\epsilon$ values of $[0.3, 0.5, 1.0, 3.0, 5.0, 10.0]$. The adversary's objectives are the same as in Section 7.

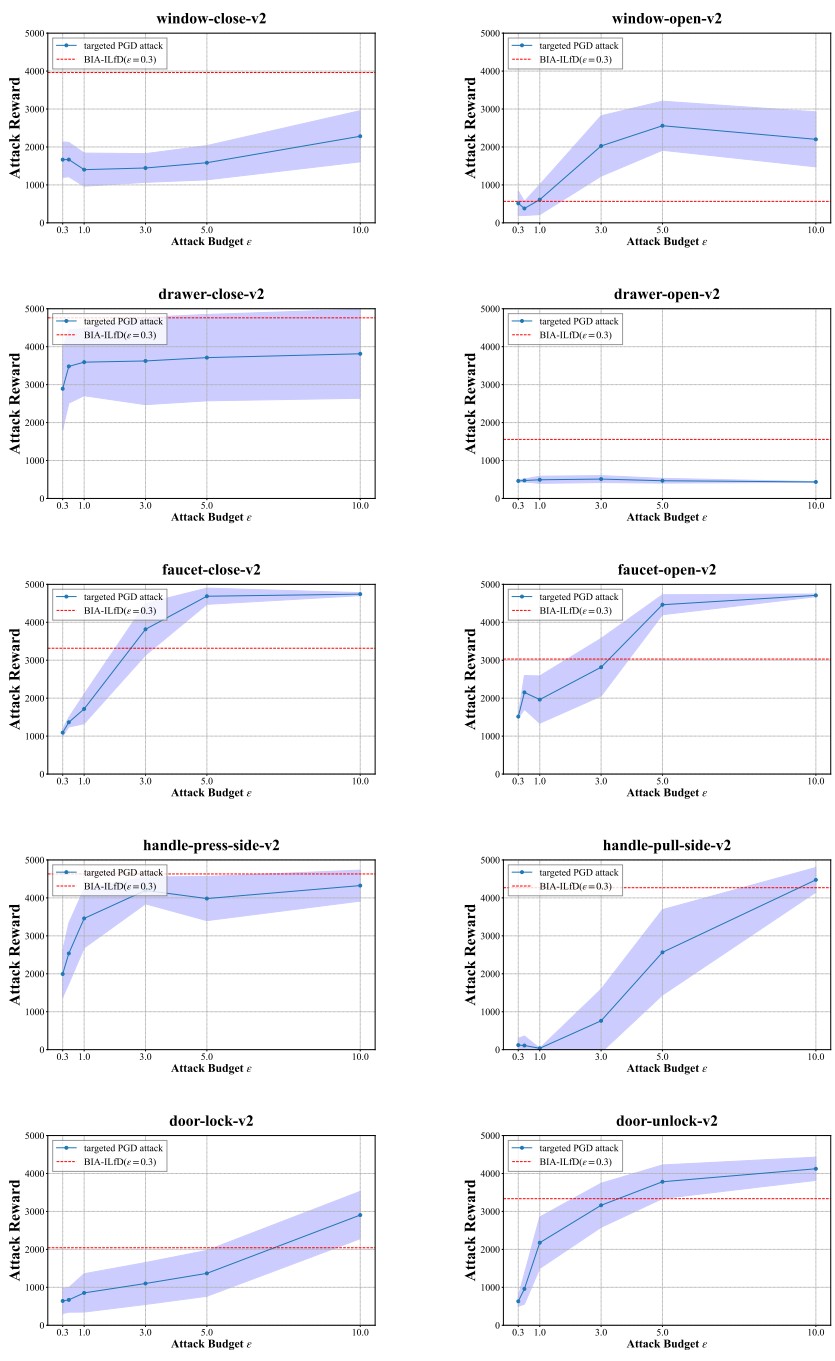

Figure 5: Performance of targeted PGD attacks under different attack budgets $\epsilon$. The x-axis represents the attack budget $\epsilon$, and the y-axis represents the attack reward. Each environment name represents an adversarial task. The solid line and shaded area denote the mean and the standard deviation / 2 over 50 episodes.

---

**Algorithm 3** Optimization of adversarial states at each step in the targeted PGD attack

---

1: **Input:** Initial state $s$, target policy network $\pi_{\text{tgt}}$, victim's policy network $\pi$, perturbation bound $\epsilon$, number of steps $T$
2: **Output:** Perturbed state $\hat{s}$
3: $\epsilon_{\text{step}} \leftarrow \epsilon/T$
4: $s_{\min} \leftarrow s - \epsilon$, $s_{\max} \leftarrow s + \epsilon$
5: $\hat{s} \leftarrow s + \text{Uniform}(-\epsilon_{\text{step}}, \epsilon_{\text{step}})$
6: $a_{\text{tgt}} \sim \pi_{\text{tgt}}(\hat{s}|\cdot)$
7: **for** $t = 1$ to $T$ **do**
8: $\quad a_{\text{current}} \sim \pi(\hat{s}|\cdot)$
9: $\quad \mathcal{L} \leftarrow \|a_{\text{current}} - a_{\text{tgt}}\|_2^2$
10: $\quad \hat{s} \leftarrow \hat{s} - \text{sign}(\nabla_{\hat{s}} \mathcal{L}) \cdot \epsilon_{\text{step}}$
11: $\quad \hat{s} \leftarrow \text{clip}(\hat{s}, s_{\min}, s_{\max})$
12: **end for**

---

Figure 5 shows the experimental results. The environment names in the graphs represent adversarial tasks. In most tasks, attack performance increased as the attack budget increased. We hypothesize that this is because, with a larger attack budget, it becomes easier to find fictitious states where the victim's chosen actions perfectly match those of the target policy. However, in the drawer-open task, attack performance did not improve even with a larger attack budget.

Comparing BIA with other attack methods at $\epsilon = 0.3$, we find that in some tasks, even with a large attack budget, the attack performance did not surpass that of BIA. This strongly suggests that single-step optimization is ineffective for behavior-targeted attacks. Therefore, we argue that in behavior-targeted attacks, it is crucial to train an adversarial policy that considers future behavior when performing attacks.

