# OpenReview forum: "Robust Deep Reinforcement Learning against Adversarial Behavior Manipulation"
_ICLR.cc/2026/Conference — ICLR 2026 Poster_

### Official Review · Reviewer_Yovk · 2025-10-29

**Soundness:** 3
**Presentation:** 3
**Contribution:** 2
**Rating:** 4
**Confidence:** 4

**Summary:**

This paper proposes a new state-perturbation attack against a trained RL agent, aiming to manipulate the agent into behaving according to a target policy chosen by the attacker. The main idea is to use imitation learning techniques, such as GAIL, to learn the attack policy from demonstrations of a target policy, so that the attacker does not need to access the agent's policy directly. Further, the paper proposes a new defense adapted from the SA-PPO algorithm, using time-discounted regularization to train a smoothed policy.

**Strengths:**

This problem of adversarial behavior maniplation through state perturbation has been considered recently. The main contribution of the paper is showing that the attacker's problem can be viewed as a MDP with carefully defined reward and transition functions derived from the agent's MDP, and utilizing behavior imitation to avoid white-box access to agent's policy. Another contribution of the paper is the obsevation that the state changes in the early stages of trajectories have a great influence on the agent's overall performane in continuing tasks and the time-discounted regularization scheme derived from that.

**Weaknesses:**

1. While the paper does not need white-box access to the agent's policy, it requires demonstrations from the target policy, which might not be easy to get in practice without strong domain knowledge, including the environment and the agent's policy. In particular, behavior manipulation includes reward minimization as a special case, treating a reward minimization policy as the target policy.
2. Algorithm 1 is a straightforward adaptation of GAIL, and it can be directly derived from the objective (2), without Theorem 5.1 and the discussions in Section 5. Why are those discussions useful?
3. The defense objective (10) does not make sense to me. From the agent's perspective, it should just try to maximize its own reward. Why should it try to minimize the attacker's gain? This only makes sense in the zero-sum setting. Further, since behavior manipulation includes reward minimization as a special case, the agent should consider the worst-case scenario to achieve robust defense. In that sense, there is no need to distinguish target behavior manipulation and reward minimization from the agent's perspective.
4. The observation that the agent's policy is more sensitive to early stages of trajectories looks like an easy consequence of using a discount factor in continuing tasks, and this should apply to the reward minimization case as well. However, this result does not apply to episodic tasks and when the discount factor is very close to 1, which are commonly considered in the previous work, such as SA-PPO.
5. The evaluation is not very convincing. The paper does provide baseline results for other defenses except in the Meta-World environment. Further, it ignores important recent defenses such as:
- Li et al., Towards Optimal Adversarial Robust Q-learning with Bellman Infinity-error, ICML 2024.
- Sun et al., Belief-Enriched Pessimistic Q-Learning against Adversarial State Perturbations, ICLR 2024.
- Yang et al., DMBP: Diffusion Model-Based Predictor for Robust Offline Reinforcement Learning against State Observation Perturbations, ICLR 2024.

**Questions:**

1. Algorithm 1 is a straightforward adaptation of GAIL, and it can be directly derived from the objective (2), without Theorem 5.1 and the discussions in Section 5. Why are those discussions useful?
2. Why should the agent consider target behavior manipulation? It seems that the agent should always consider the worst-case scenario to achieve robust defense, and there is no need to distinguish target behavior manipulation and reward minimization from the agent's perspective.
3. Is time-discounted regularization still useful when gamma is close to 1, as typically considered in previous work?

---

> ### Author Response · Authors · 2025-11-21
> **Response (1/3)**
>
> **Thank you very much for taking the time to review our paper.** In what follows, we aim to clarify several points. We have also **uploaded a revised version** of the paper, and all changes are highlighted in **red.**
>
> ---
>
> # W1. Practicality of the Demonstration Assumption
>
> While we recognize that there are scenarios in which collecting demonstrations can be challenging, we believe that **in many practical settings, it is still easier to obtain demonstrations than to design the reward function assumed by existing attack methods.**
>
> Indeed, **imitation learning has been widely studied as a way to avoid the cost of designing reward functions [A, B].** Reward design usually requires careful manual engineering and tuning, whereas demonstrations can be obtained by sampling a few trajectories that show the desired behavior. Thus, our threat model captures a realistic adversary.
>
> Furthermore, to the best of our knowledge, **this is the first work that studies how to learn an adversarial policy directly from demonstrations, which constitutes our novelty.** In addition, our theoretical analysis provides insights into general policy learning since behavior manipulation includes reward minimization as a special case, as the reviewer notes.
>
> Overall, we would be glad if the reviewer could view this assumption in light of these practical and conceptual advantages.
>
> [A] Ibarz, Borja, et al. "Reward learning from human preferences and demonstrations in atari." *Advances in neural information processing systems* 31 (2018)
>
> [B] Sontakke, Sumedh, et al. "Roboclip: One demonstration is enough to learn robot policies." *Advances in Neural Information Processing Systems* 36 (2023): 55681-55693.
>
> ---
>
>
> # W2, Q1. Why are Theorem 5.1 and the discussions in Section 5 useful?
>
> In short, Theorem 5.1 and Section 5 play an essential role in our paper to **provide a theoretical guarantee that a worst-case attacker for behavior-targeted attacks is learnable** by using a conventional imitation learning algorithm such as GAIL. Thus, **if Theorem 5.1 and Section 5 were absent, BIA would remain a method whose effectiveness is supported only empirically**, which would be insufficient as an evaluation of vulnerability under worst-case attacks.
>
> ### Why is it not sufficient to directly apply GAIL to objective (2)? (Significance of Section 5)
>
> The original GAIL formulation differs from our objective (2), and it is not theoretically obvious that GAIL can be applied directly.
>
> - GAIL assumes that the entire composition $\pi \circ \nu $ ($\pi$ is a victim policy,  $\nu$ is an adversarial policy) is jointly optimizable.
> - In our threat model, however, the victim policy $\pi$ is fixed, so this assumption does not hold.
> - Section 5 shows that, even under this constraint, the optimal adversarial policy is learnable by existing imitation-learning algorithms, which satisfy assumptions about variational representation.
>
> Thus, the use of GAIL in Algorithm 1 should be viewed as one concrete instantiation of this broader theoretical result. **The key contribution is theoretically showing that objective (2) is solvable through imitation learning.**
>
> ### Theorem 5.1 provides a worst-case guarantee (Significance of Theorem 5.1)
>
> Theorem 5.1 establishes that our attack method (BIA) provides a worst-case attack scenario for behavior-targeted attacks.
>
> - Theorem 5.1 shows that an optimal attacker for objective (2) can be characterized as the solution of an RL problem.
> - Consequently, the attacker learned by BIA approximates a worst-case attacker.
>
> **This property is highly valuable for vulnerability analysis.** In contrast, simply applying GAIL without this theoretical foundation does not guarantee worst-case optimality.

---

> ### Author Response · Authors · 2025-11-21
> **Response (2/3)**
>
> # W3, Q2. Why should the agent consider target behavior manipulation?
>
> Thank you for this insightful comment. We agree that if the defender only aims to maximize its own reward, then objective (10) would indeed appear unnecessary; however, **our motivation for introducing objective (10) is to prevent *unintended exploitation* of the policy that existing defense methods cannot handle, regardless of the victim’s reward.**
>
> ### **Exploitation can occur even if the victim’s reward does not decrease.**
>
> The victim’s reward is designed to reflect the victim’s own interests, but situations in which the adversary’s objective can be achieved simply by minimizing the victim’s reward are limited to special cases such as zero-sum settings. In contrast, **there are many situations where the attacker can achieve their objective without affecting the victim’s reward at all.**
>
> - For example, in a recommender system, the attacker could bias the system to recommend only those items that are highly profitable to the attacker, among the items the user is likely to prefer. In this case, the user’s satisfaction is largely preserved, so the victim’s reward does not drop significantly. However, the attacker still obtains substantial gain.
>
> **We regard such *unintended exploitation of the victim* as a serious security problem that should be addressed**. Therefore, we consider it reasonable to incorporate the attacker’s gain as a general safety metric, not only in zero-sum settings.
>
> ### The defense objective (10) explicitly addresses the threat.
>
> While existing defenses that only consider worst-case reward minimization cannot handle this threat, our defense objective (10) directly incorporates the attacker’s gain. Thus, TDRT, which is designed based on an upper bound of this objective, provides both robustness and a theoretical justification against behavior-targeted attacks.
>
> Furthermore, as the reviewer points out, behavior-manipulation attacks include reward minimization as a special case. Thus, our formulation generalizes reward minimization by including it as a special case. As a result, beyond its security implications, we believe our method offers a useful general framework for discussing future robust RL from a machine-learning perspective as well.
>
> Overall, we believe that **defenses that explicitly consider target behavior manipulation represent an important direction in robustness research for RL** and constitute a well-motivated problem setting. We would be pleased if the reviewer will find this direction both meaningful and valuable.
>
> ---
>
> # W4, Q3. On time-discounted regularization
>
> Thank you very much for raising this important point. We agree that the observation is intuitively a natural consequence of discounting. However, **our contribution is not to rediscover this intuition, but to formulate it explicitly as a theoretical upper bound on robustness against behavior-targeted attacks, and then to translate this bound into a principled defense algorithm.**
>
> - To the best of our knowledge, no prior work, including reward-minimization defenses, has actually carried out this analysis, and we are the first to establish it theoretically.
> - In particular, our theory provides a non-trivial and explicit guideline for how regularization should decay with the discount factor $\gamma$ and timestep $t$.
>
> As the reviewer points out, our observation that the agent's policy is more sensitive to early stages of trajectories does not apply to episodic tasks and when the discount factor is very close to 1. In such cases, our theorem implies that all timesteps contribute almost equally to the defender’s effectiveness, a behavior similar to that of SA-PPO. **However, the theoretical analyses of SA-PPO and TDRT-PPO are grounded in different defense objectives, and it is therefore non-trivial whether this property should also hold under behavior-targeted attacks.** We regard this insight as one of our contributions.
>
> To avoid misunderstanding, we briefly mention the assumptions underlying our theorem. We assume a discount factor $\gamma \in (0,1)$, which is the same assumption used in SA-PPO (Theorem 5 in [C]) and is therefore standard in this line of work.
>
> [C] Zhang, Huan, et al. "Robust deep reinforcement learning against adversarial perturbations on state observations." *Advances in neural information processing systems* 33 (2020): 21024-21037.

---

> > ### Author Response · Authors · 2025-11-21
> > **Response (3/3)**
> >
> > # W5-1. Additional experiments
> >
> > > The paper does provide baseline results for other defenses except in the Meta-World environment.
> > >
> >
> > For this comment, we interpret it as a typo for “The paper does **not** provide baseline results for other defenses except in the Meta-World environment,” and respond on that basis. If our interpretation is incorrect, we would appreciate your correction.
> >
> > Thank you for raising this important point. We agree that defense baselines should also be reported in environments other than Meta-World. In the revised version, we therefore add experimental results in the MuJoCo environment (Appendix F.1) and the MiniGrid environment (Appendix F.2). We compare SA-PPO, a representative defense baseline that uses the same smoothing mechanism as our method, with our proposed TDRT-PPO. For brevity, we reproduce a subset of the results below.
> >
> > | Task | Method | Clean Rewards (↑) | Best Attack Rewards (↑) |
> > | --- | --- | --- | --- |
> > | Ant | PPO | 4720 ± 255 | 850 ± 612 |
> > | Ant | SA-PPO (smoothing w/o time-discounting) | 4268 ± 342 | 3318 ± 391 |
> > | Ant | **TDRT-PPO (ours, smoothing w/ time-discounting)** | 4612 ± 219 | 3206 ± 781 |
> > | HalfCheetah | PPO | 3680 ± 220 | 160 ± 540 |
> > | HalfCheetah | SA-PPO (smoothing w/o time-discounting) | 2963 ± 398 | 2826 ± 513 |
> > | HalfCheetah | **TDRT-PPO (ours, smoothing w/ time-discounting)** | 3577 ± 197 | 2981 ± 462 |
> > | Hopper | PPO | 3290 ± 150 | 820 ± 430 |
> > | Hopper | SA-PPO (smoothing w/o time-discounting) | 3198 ± 97 | 1579 ± 672 |
> > | Hopper | **TDRT-PPO (ours, smoothing w/ time-discounting)** | 3214 ± 163 | 1345 ± 787 |
> >
> > These experiments show the same trend as in the Meta-World environment. Compared with SA-PPO, TDRT-PPO exhibits a smaller drop in clean reward from PPO, while maintaining a robustness level under attack that is comparable to SA-PPO.
> >
> > # W5-2. Additional Related works
> >
> > Thank you for this important comment. We have added a comparison with the suggested works in the Related Works section. For clarity, we summarize the differences between these methods and ours as follows.
> >
> > - **[D, E, F] have different defense objectives from ours.** Our defense is designed for behavior-targeted attacks, whereas all of [D, E, F] focus on reward-minimization attacks. Consequently, these defenses do not provide a theoretical justification of robustness against behavior-targeted attacks. We also note that the diffusion model-based defenses [E, F] can in principle be applied to our problem setting as well, but they are inference-time defenses and are therefore orthogonal to our approach as explained in detail later.
> > - **Diffusion model-based defenses [E, F] are orthogonal to our defense.** These methods purify perturbed states at inference time. Our defense instead introduces a regularization term during training and does not require any additional preprocessing at test time. As they act on different components of the overall pipeline, our defense and [E, F] are orthogonal and complementary.
> > - **[D, E] focus on DQN in discrete action spaces.** In contrast, our defense method is applicable to both continuous and discrete action spaces.
> >
> > Once again, we sincerely appreciate these very helpful suggestions.
> >
> > [D] Li et al., Towards Optimal Adversarial Robust Q-learning with Bellman Infinity-error, ICML 2024.
> >
> > [E] Sun et al., Belief-Enriched Pessimistic Q-Learning against Adversarial State Perturbations, ICLR 2024.
> >
> > [F] Yang et al., DMBP: Diffusion Model-Based Predictor for Robust Offline Reinforcement Learning against State Observation Perturbations, ICLR 2024.
> >
> > ---
> >
> > ### **Thank you again for taking the time to review our paper.**
> >
> > We hope this addresses your concerns, and we would greatly appreciate it if you could reconsider your score.

---

> ### Author Response · Authors · 2025-11-28
> **Follow-up on Rebuttal**
>
> Dear Reviewer Yovk,
>
> Thank you again for your thoughtful and constructive review. As the discussion period ends on December 2, we would greatly appreciate it if you could briefly review our responses and let us know whether they resolve your concerns or if any points remain unclear. We are happy to provide further clarification if needed.

---

### Official Review · Reviewer_kHJs · 2025-10-29

**Soundness:** 3
**Presentation:** 4
**Contribution:** 2
**Rating:** 6
**Confidence:** 3

**Summary:**

This paper introduces behavior-targeted attacks, which steer a victim RL agent’s policy toward an adversary-desired behavior rather than simply reducing reward. It proposes (1) Behavior Imitation Attack (BIA) — an imitation-learning–based attack operable in black-box or no-box settings, and (2) Time-Discounted Regularization Training (TDRT) — a defense that suppresses early-trajectory policy sensitivity via time-weighted regularization.

**Strengths:**

1. Innovative attack under limited access: BIA elegantly converts the problem into an imitation-learning formulation, removing dependence on victim parameters.

2. TDRT’s time-discounted regularization is motivated by a provable bound (Theorem 6.1) and empirically validated.

3. Experiments span multiple continuous-control and grid environments, with clear comparisons against strong baselines (ATLA-PPO, SA-PPO, RAD-PPO, WocaR-PPO).

4. The paper is well-structured, with a precise threat model, proofs in appendices, and reproducibility statements including algorithm pseudocode.

**Weaknesses:**

1. Diffusion model based defenses[1][2] have been proposed recently to fight against state adversarial perturbations. Could the author compare them with the proposed defense?

2. The proposed method currently do not scale to image based input RL environments such as Atari games.

[1] Z. Yang and Y. Xu. DMBP: Diffusion Model–Based Predictor for Robust Offline Reinforcement Learning against State Observation Perturbations. ICLR, 2024.

[2] X. Sun and Z. Zheng. Belief-Enriched Pessimistic Q-Learning against Adversarial State Perturbations. ICLR, 2024.

**Questions:**

1. I would like to see a discussion between the behavior imitation attack and the backdoor attack. This should be an important related work in this work.

2. I would like to see some results on diffusion model-based defenses against behavior imitation attacks.

3. Could the proposed methods extend to a large state space environment, such as images based environment, such as Atari games? I know it is a challenging extension, but do the authors have any potential ideas on this?

---

> ### Author Response · Authors · 2025-11-21
> **Response**
>
> **Thank you very much for taking the time to review our paper.** We also appreciate your positive assessment of our work. We have  **uploaded a revised version** of the paper, and clarifications are highlighted in **red.**
>
> ---
>
> # W1, Q2. Comparison with diffusion-based defenses
>
> Thank you very much for this insightful suggestion. We agree that a comparison between diffusion-based defenses and our method is important.
>
> To clarify the structural differences, we have added the following comparison to the Related Works section:
>
> > Several works [A, B] propose diffusion-based defenses, which perform purification at inference time to remove adversarial perturbations from corrupted states. In contrast, our approach is a training-time regularization method and requires no additional preprocessing during inference. As a result, these defenses operate on different components of the pipeline, making them orthogonal and complementary.
> >
>
> In addition, we are in the process of implementing the diffusion-based defense [A] to evaluate its effectiveness against behavior-targeted attacks. (Note that [B] assumes a discrete action space and therefore cannot be directly applied to our experimental setting.) Once the results are available, we will report them as an additional comparison.
>
> [A] Z. Yang and Y. Xu. DMBP: Diffusion Model–Based Predictor for Robust Offline Reinforcement Learning against State Observation Perturbations. ICLR, 2024.
>
> [B] X. Sun and Z. Zheng. Belief-Enriched Pessimistic Q-Learning against Adversarial State Perturbations. ICLR, 2024.
>
> ---
>
> # W2, Q3. Scalability to Large State Space Environments
>
> Thank you for this constructive question. As you pointed out, while our attack is in principle applicable to high-dimensional state spaces, achieving strong attack performance in such settings is practically challenging. This issue has also been discussed in prior work [C]. Below, we outline several potential directions for addressing this challenge.
>
> ### Toward Scaling to Large State Space Environments
>
> The main difficulty in large state space environments arises from a fundamental challenge in RL: **policy learning becomes significantly harder when the action space is high-dimensional**. Since the victim’s state space directly serves as the action space for the adversarial policy, increasing the dimensionality of the state space makes it harder to learn an effective adversarial policy, which in turn reduces attack performance.
>
> To address the difficulty, a promising direction is **to perturb only the most influential state dimensions, rather than all dimensions uniformly**. Several prior studies [D, E] suggest that not all state dimensions contribute equally to the victim’s action selection; instead, a small subset plays a disproportionately important role. Based on this insight, the adversary could extract **critical state dimensions,** either before or jointly with adversarial policy learning, and restrict the adversarial policy to operate only on this reduced **effective action space**. This may alleviate the difficulty of learning in a high-dimensional action space and improve attack effectiveness.
>
> **Such selective perturbation also has advantages from a practical standpoint.** In many environments, perturbing all dimensions of the state may be infeasible due to cost or implementation constraints. Limiting perturbations to dimensions that are easier to manipulate or more actionable for the adversary provides a more realistic attack model.
>
> [C] Sun, Yanchao, et al. "Who Is the Strongest Enemy? Towards Optimal and Efficient Evasion Attacks in Deep RL." ICLR2022.
>
> [D] Huber et al., “Enhancing Explainability of Deep Reinforcement Learning Through Selective Layer-Wise Relevance Propagation,” 2019
>
> [E] Hao et al., “Sparse Feature Selection Makes Batch Reinforcement Learning More Sample Efficient.” ICML 2021.
>
> ---
>
> # Q1. Discussion on Behavior Imitation Attacks vs. Backdoor Attacks
>
> We fully agree that the discussion between behavior imitation attacks and backdoor (poisoning) attacks is an important aspect of our work, and we are in fact already discussing this relationship in Appendix A.1. Following your suggestion, we have strengthened this discussion and added a pointer from the main text to the Appendix. In summary, our comparison is as follows:
>
> - Many poisoning attacks share a similar goal with our attack. They aim to steer the victim’s policy toward an attacker-specified target policy. However, in our problem setting, the adversary is **not allowed to intervene in the victim’s training process**. In contrast, poisoning attacks **require access to the victim’s training phase** in order to inject manipulated data or states. For this reason, these poisoning-based attack settings fall outside the scope of our threat model.
>
> ---
>
> ### **Thank you again for taking the time to review our paper.**
>
> We are also truly grateful for your favorable evaluation of our work.

---

> > ### Comment · Reviewer_kHJs · 2025-11-24
> >
> > Thank you to the authors for their responses. I am interested in seeing the results for the diffusion defenses. And I am keeping my positive rating.

---

> > > ### Author Response · Authors · 2025-11-27
> > > **Response**
> > >
> > > **Thank you for your response and for your positive evaluation of our work.**
> > >
> > > We additionally ran experiments to evaluate a diffusion model-based defense (DMBP) against behavior-targeted attacks. The table below reports the best attack rewards (lower is better) for each task.
> > >
> > > | Task | PPO (No defense) | ATLA-PPO (AdvTraining) | TDRT-PPO (ours) | PPO + DMBP (diffusion model-based defense) |
> > > | --- | --- | --- | --- | --- |
> > > | window-close | 4505 ± 65 | 4270 ± 188 | **482 ± 3** | 3442 ± 492 |
> > > | window-open | 566 ± 523 | 586 ± 649 | **254 ± 214** | 498 ± 411 |
> > > | drawer-close | 4760 ± 640 | 4858 ± 6 | 4860 ± 4 | **4792 ± 323** |
> > > | drawer-open | 1556 ± 607 | 1158 ± 1026 | **378 ± 10** | 1323 ± 891 |
> > > | faucet-close | 3409 ± 652 | 4108 ± 790 | 1789 ± 610 | **1642 ± 592** |
> > > | faucet-open | 3031 ± 1493 | 4383 ± 449 | 1942 ± 261 | **1837 ± 318** |
> > > | handle-press-side | 4726 ± 175 | 4302 ± 799 | **1928 ± 736** | 2002 ± 881 |
> > > | handle-pull-side | 4268 ± 740 | 532 ± 534 | **7 ± 1** | 512 ± 682 |
> > > | door-lock | 2043 ± 1229 | 1020 ± 805 | **487 ± 11** | 643 ± 194 |
> > > | door-unlock | 3421 ± 974 | 3277 ± 1265 | **691 ± 356** | 1501 ± 567 |
> > >
> > > From these results, we observe that the **diffusion model-based defense can still provide some robustness even against behavior-targeted attacks.** In particular, for tasks such as faucet-open and faucet-close, PPO + DMBP achieves stronger robustness than TDRT-PPO. This suggests that, unlike adversarial training methods that are tailored to a specific attack objective (e.g., reward minimization), the diffusion model-based defense learns to denoise perturbed states in a way that is less dependent on the attack goal. Consequently, it can mitigate the impact of behavior-targeted attacks.
> > >
> > > On the other hand, **the effect of the diffusion model-based defense is limited on some tasks, such as window-close and drawer-open.** We hypothesize that this is because the state distribution of the victim’s original task is very different from that of the adversary’s target task. During training, the diffusion model learns to map perturbed states back to clean states while conditioning on trajectories from the victim task. At test time under attack, however, the trajectories used for conditioning are replaced by those from the adversary’s target task.  In such cases, the diffusion model can fail to map adversarially perturbed states back to clean ones, which weakens the overall defensive effect.
> > >
> > > As discussed in the initial response, TDRT is a training-time regularization method and requires no additional preprocessing at inference. In contrast, diffusion model-based defenses perform denoising during inference, which incurs extra computational cost. In this sense, TDRT and diffusion model-based defenses are orthogonal and complementary. We believe that each of these defenses represents a promising direction for defending against behavior-targeted attacks.

---

### Official Review · Reviewer_vKMc · 2025-10-31

**Soundness:** 4
**Presentation:** 3
**Contribution:** 4
**Rating:** 8
**Confidence:** 3

**Summary:**

In this paper, the authors propose behavior imitation attack (BIA), a new class of behavior-targeted attack in DRL (where the adversary's goal is to manipulate the victim policy in such a way that it benefits the adversary instead of pure reward minimization) via adversarial manipulations to state observations. Unlike prior work, BIA leverages imitation learning and replaces the state observations of the victim with the falsified state generated by the adversary policy and works under black-box or no-box settings. The authors also present a defense strategy, Time-Dependent Robust Training (TDRT), against such attacks. TDRT introduces a regularization term to suppress the sensitivity of action outputs to state changes. Experiments on Meta-World, MuJoCo, and MiniGrid show that BIA can effectively manipulate victim policies, and TDRT outperforms existing adversarial training baselines in robustness-performance trade-offs.

I believe that the theoretical insights combined with the novelty is enough for acceptance.

**Strengths:**

S1. First black-box/no-box behavior-targeted attack: The introduction of BIA allows for attack generation with extremely limited victim access, which aligns with realistic threat models.

S2. Theoretical analysis: The authors provide a theoretical basis for both the attack and defense strategies proposed in the paper. I also find the motivation behind TDRT simple and elegant.

S3. Strong empirical evaluation: Although most of the experiments and valuable ablation studies are pushed to the appendix, results are comprehensive and multiple baselines are compared using different environment settings.

**Weaknesses:**

W1. The defense method is referred to as both Time-Discounted Robust Training (in introduction) and Time-Discounted Regularization Training (in later sections). The naming should be unified throughout the paper.

W2. The proposed attack may not scale to high-dimensional state spaces, which was also admitted by the authors in the limitations section.

**Questions:**

Q1. What is the computational cost of BIA compared to white-box attack baselines?

Q2. Is TDRT (or policy smoothing) vulnerable to reward minimization attacks? Can we combine adversarial training with policy smoothing to equip agents with a defense that can cover different attack types?

Q3.  When I click to repository in Appendix I, I get the following error: "The repository is expired". The authors should ensure that the code is accessible.

**Details Of Ethics Concerns:**

No ethics concerns

---

> ### Author Response · Authors · 2025-11-21
> **Response**
>
> **Thank you very much for taking the time to review our paper.** We also appreciate your positive assessment of our work. We have **uploaded a revised version** of the paper, and clarifications are highlighted in **red.**
>
> ---
>
> ## W1. Mixed Naming of Defense Methods
>
> Thank you very much for pointing this out. We apologize for the confusion. In the revised version, we have unified the name to **“Time-Discounted Robust Training.”**
>
> ---
>
> ## W2. Scalability to high-dimensional state spaces
>
> As you pointed out, our attack is in principle applicable to high-dimensional state spaces, but achieving strong attack performance in such settings is practically challenging. This difficulty has also been recognized in prior work [A] and and remains an open and important problem.
>
> Nonetheless, prompted by Reviewer kHJs’s question, we propose a potential direction to address this issue (see our response to Q3 from Reviewer kHJs). While we cannot guarantee that this idea will fully resolve the problem, we believe it points to a promising avenue for extending our attack to high-dimensional state spaces in future work.
>
> [A] Sun, Yanchao, et al. "Who Is the Strongest Enemy? Towards Optimal and Efficient Evasion Attacks in Deep RL." ICLR2022.
>
> ---
>
> ## Q1. Computational cost of BIA
>
> Thank you for pointing out this important issue. In Appendix G.5 of the revised version, we have added comparison experiments on the computational cost of BIA. We summarize these results below.
>
> A direct one-number comparison between BIA and a white-box attack (Targeted PGD) is inherently difficult, because they place the computational burden in different phases:
>
> - **BIA** incurs a **training-time cost** to learn an adversarial policy, but **test-time cost is just a forward pass** of that policy.
> - **Targeted PGD** has **no training-time cost**, but incurs a **test-time optimization cost at every timestep** of every attacked episode.
>
> Because of this difference, we report both training time and per-episode test time, measured on a single NVIDIA H100 GPU:
>
> |  | Targeted PGD (white-box) | Rew Max (PA-AD, white-box) | Rew Max (SA-RL) | BIA-ILfD (ours) | BIA-ILfO (ours) |
> | --- | --- | --- | --- | --- | --- |
> | **Training time** | – | 4.3 hours | 3.6 hours | 6.2 hours | 6.1 hours |
> | **Test time** | 37 sec / episode | – | – | – | – |
>
> **BIA vs. reward-based white-box attacks (Rew Max):** BIA has a higher training cost than Rew Max (PA-AD). This is mainly because imitation learning–based BIA must train both a policy and a discriminator, while Rew Max only optimizes the adversarial policy. On the other hand, BIA only requires demonstrations of the target behavior, whereas Rew Max assumes access to the target reward function, which is often harder to obtain in practice.
>
> **BIA vs. Targeted PGD:** As described above, Targeted PGD has no training-time cost, but incurs a test-time optimization cost. When many episodes need to be attacked (e.g., long deployments or large-scale evaluation), BIA’s training cost is spread out over many episodes, and BIA can become more computationally efficient overall than repeatedly running Targeted PGD.
>
> ---
>
> ## Q2. Is TDRT (or policy smoothing) vulnerable to reward-minimization attacks?
>
> Thank you for this valuable question. We clarify the robustness of policy smoothing against reward-minimization attacks.
>
> **Policy smoothing has been shown to be effective against reward-minimization attacks.** Existing works have proposed policy smoothing as a defense specifically for reward-minimization attacks, and its robustness has been demonstrated empirically [B, C]. Moreover, **the objective function of TDRT (Eq. (10)) includes reward minimization as a special case**. If the victim’s reward function is $R$, setting $R_{\text{tgt}} = -R$ makes the attacker’s objective equivalent to reward minimization. Thus, TDRT is theoretically justified to be effective against reward-minimization attacks as well.
>
> We also believe that TDRT can be combined with adversarial training. While this combination increases computational cost, it allows the victim to choose which type of attacker they want to be robust against by changing the assumed adversarial policy. For this reason, such a combination can be a practical option in real applications.
>
> [B] Zhang, Huan, et al. "Robust deep reinforcement learning against adversarial perturbations on state observations." *Advances in neural information processing systems* 33 (2020): 21024-21037.
>
> [C] Zhang, Chen, et al. "Robust Reinforcement Learning on State Observations with Learned Optimal Adversary." ICLR 2021.
>
> ---
>
> ## Q3. Code Availability
>
> We apologize that the repository has expired. We have generated a new link and included the correct one in the revised version of the paper. Thank you very much for bringing this to our attention.
>
> ---
>
> ### **Thank you again for taking the time to review our paper.**
> We are also truly grateful for your favorable evaluation of our work.

---

> ### Author Response · Authors · 2025-11-28
> **Follow-up on Rebuttal**
>
> Dear Reviewer vKMc,
>
> Thank you again for your thoughtful and constructive review. As the discussion period ends on December 2, we would greatly appreciate it if you could briefly review our responses and let us know whether they resolve your concerns or if any points remain unclear. We are happy to provide further clarification if needed.

---

### Official Review · Reviewer_185Q · 2025-11-01

**Soundness:** 2
**Presentation:** 2
**Contribution:** 2
**Rating:** 4
**Confidence:** 5

**Summary:**

This paper proposes TDRT, an adversarial training method for mitigating adversarial behavior manipulation, along with a novel black-box adversarial behavior manipulation method, BIA. BIA leverages adversarial imitation learning to train the adversarial policy using demonstrations of the adversary's target behaviors under the State-Adversarial MDP (SA-MDP) setting. TDRT then integrates BIA into the adversarial training process to learn a robust policy against behavior-targeted attack. Experiments show BIA and TDRT are comparable with previous works, SA-RL and SA-PPO, separately.

**Strengths:**

1. This paper is well written and easy to follow.
2. The motivation for developing black-box behavior-targeted attacks is clearly analyzed.
3. Experiments demonstrate the effectiveness of BIA and TDRT.

**Weaknesses:**

1. The novelty of this paper is limited. Firstly, regarding BIA, it is clear that BIA is equivalent to SA-RL; both approaches involve adversarial policy learning within the SA-MDP framework. The primary distinction is that BIA requires the adversary to provide several demonstrations of the target behavior, whereas SA-RL necessitates the adversary to develop a reward model for that behavior. Moreover, Theorem 5.1 in BIA closely resembles Lemma 1 from Zhang et al. (2020b), with the only difference being that the adversary's reward function in BIA is derived from the demonstration samples.
Secondly, concerning TDRT, it is also evident that TDRT is equivalent to SA-PPO. Theorem 6.1 is nearly identical to Theorem 5 from Zhang et al. (2020b), differing primarily in that TDRT employs a discounted form while SA-PPO uses a maximum form.

2. Due to the limited novelty, the performance of BIA and SA-RL, and the performance of TDRT and SA-PPO, are almost the same. Although the authors compare the clean task performance of TDRT and SA-PPO and show that TDRT achieves better clean task performance, it may be mainly due to the different hyperparameter sets. The authors should provide a clearer and fair comparison of the best performance of both SA-PPO and TDRT.

**Questions:**

Please refer to Weaknesses.

---

> ### Author Response · Authors · 2025-11-21
> **Response (1/2)**
>
> **Thank you very much for taking the time to review our paper.** We also appreciate your positive assessment that our approach is well motivated. To ensure full clarity, we would like to further articulate the novelty of our contributions.  We have also **uploaded a revised version** of the paper, and clarifications are highlighted in **red.**
>
> ---
>
> ### Clarification on the reviewer’s summary
>
> We would like to respectfully point out a misunderstanding in the reviewer’s summary, as it may affect the evaluation of our contributions.
>
> The reviewer writes:
>
> > TDRT then integrates BIA into the adversarial training process to learn a robust policy against behavior-targeted attack.
>
> However, BIA and TDRT are separate components in our framework. **TDRT does not perform adversarial training by using BIA**. Instead, TDRT applies time-discounted policy smoothing to the victim’s policy, independently of any specific attack algorithm such as BIA.
>
> ---
>
> # W1. Novelty concern
>
> The goals of our attack and defense cannot be achieved by a straightforward application of prior work. Our proposed methods are fundamentally different from existing methods in both **objectives and methodology**.
>
> ## Novelty of Our Attack Method (BIA)
>
> ### Difference in objectives and threat model (BIA vs. SA-RL)
>
> Although both BIA and SA-RL use SA-MDP framework, their objectives and threat models are different.
>
> - SA-RL aims to **minimize the victim’s cumulative reward** under the assumption that the victim’s reward function is given to the adversary.
> - In contrast, BIA aims to **steer the victim’s behavior toward a target policy**, irrespective of the victim’s reward function, assuming access only to demonstrations of the target behavior.
> - Consequently, **SA-RL cannot directly realize a behavior-targeted attack under our threat model, where only demonstrations are available.**
> - Moreover, in many complex environments, **designing a reward function is difficult, whereas collecting demonstrations of desired behaviors is often much easier [A, B]**. This distinction is practically important and clearly separates SA-RL from BIA.
>
> ### Methodological differences (Our Theorem 5.1 vs. Lemma 1 from Zhang et al. (2020b))
>
> Our Theorem 5.1 proves that the **worst-case attacker for a behavior-targeted attack is learnable**, which is conceptually and technically distinct from Lemma 1 in Zhang et al. (2020b).
>
> - Lemma 1 in Zhang et al. (2020b) only shows that the **optimal attacker for reward minimization is learnable**, but it does not guarantee that the adversary can construct an optimal attacker for a behavior-targeted attack objective.
> - In contrast, our Theorem 5.1 shows that the **optimal attacker for the behavior-targeted attack objective is learnable by using existing imitation learning algorithms.**
> - **This result is not a technically trivial generalization** of the reward-minimization setting. To obtain this extension, **we carefully exploit assumptions from imitation learning.**
>
> Taken together, **BIA provides a framework for evaluating robustness against worst-case behavior-targeted attackers.** We believe this constitutes a meaningful contribution to vulnerability analysis.
>
> [A] Ibarz, Borja, et al. "Reward learning from human preferences and demonstrations in atari." *Advances in neural information processing systems* 31 (2018)
>
> [B] Sontakke, Sumedh, et al. "Roboclip: One demonstration is enough to learn robot policies." *Advances in Neural Information Processing Systems* 36 (2023): 55681-55693.

---

> ### Author Response · Authors · 2025-11-21
> **Response (2/2)**
>
> ## Novelty of Our Defense Method (TDRT)
>
> ### Difference in objectives (TDRT vs. SA-PPO)
>
> TDRT and SA-PPO differ clearly in the type of attack they are designed to defend against.
>
> - SA-PPO is designed for the reward-minimization attack and aims to mitigate the decrease in the victim’s cumulative reward.
> - In contrast, TDRT is designed for the behavior-targeted attacks and aims to suppress the increase of the attacker’s target reward.
> - Because of this difference in objectives, our theoretical analysis is qualitatively different from that of SA-PPO. **SA-PPO does not address robustness to behavior-targeted attacks, whereas TDRT provides a theoretical justification of robustness specifically against behavior-targeted attacks.**
>
> ### Methodological differences (Our Theorem 6.1 vs. Theorem 5 from Zhang et al. (2020b))
>
> Our Theorem 6.1 focuses on robustness against behavior-targeted attacks, and is therefore different from Theorem 5 in Zhang et al. (2020b) which focuses on robustness against the reward-minimization attack.
>
> - Theorem 5 in Zhang et al. (2020b) does not consider the behavior-targeted attack objective, and their analysis does **not** lead to the insight that earlier time steps should be regularized more strongly.
> - In contrast, our Theorem 6.1 shows that the attacker’s gain can be upper-bounded, with a time-discounted weighting, by the sensitivity of the policy to state perturbations. **This result gives a direct theoretical foundation for robustness against behavior-targeted attacks.**
> - At the algorithmic level, and as the reviewer notes, we explicitly introduce a discount factor into the regularization term. **This design leverages the above theoretical insight and offers an advantage over SA-PPO,** as it helps maintain robustness while mitigating the degradation of clean performance.
>
> Taken together, we believe that the novelty of our work is clear and that our methods provide contributions that are distinct from prior work.
>
> ---
>
> # W2. Performance Difference
>
> ### BIA vs. SA-RL
>
> The similar attack performance of BIA and SA-RL should not be interpreted as a lack of novelty. Instead, it demonstrates that **our attack based solely on demonstrations can be as effective as a reward-based attack**, which is a central contribution of our work.
>
> - In many environments, learning a policy from demonstrations (as in BIA) is generally more challenging than learning a policy from an explicit reward function (as in SA-RL). To address this challenge, a large body of work has focused on imitation learning. Thus, **compared to SA-RL, BIA operates under disadvantageous assumptions for the attacker.**
> - Therefore, the empirical result that BIA achieves attack performance comparable to SA-RL is evidence that BIA is as strong as SA-RL **under weaker assumptions.**
>
> ### TDRT vs. SA-PPO
>
> The novelty of our defense lies in the fact that TDRT maintains robustness comparable to SA-PPO **while achieving higher clean rewards**. Consequently, **the similarity in robustness performance does not undermine our contribution.**
>
> We fully acknowledge that reviewers are not obliged to read the appendix. Nevertheless, in Appendix H.2 we provide a detailed comparison over the key hyperparameter, the regularization coefficient, and show that TDRT attains higher clean rewards than SA-PPO in many settings. We reproduce a part of the results:
>
> ### Window-open
>
> | $\lambda$ | TDRT-PPO Clean Rewards (↑) | TDRT-PPO Best Attack Rewards (↓) | SA-PPO Clean Rewards (↑) | SA-PPO Best Attack Rewards (↓) |
> | --- | --- | --- | --- | --- |
> | 0.03 | **4512 ± 38** | 489 ± 526 | 4483 ± 23 | **401 ± 391** |
> | 0.1 | **4430 ± 47** | 397 ± 219 | 4219 ± 76 | **253 ± 31** |
> | 0.2 | **4403 ± 52** | **253 ± 321** | 4198 ± 87 | 284 ± 21 |
> | 0.3 | **4383 ± 57** | **254 ± 214** | 4092 ± 461 | 272 ± 32 |
> | 0.5 | **4313 ± 59** | **263 ± 298** | 4015 ± 212 | 268 ± 23 |
>
> ### Drawer-open
>
> | $\lambda$  | TDRT-PPO Clean Rewards (↑) | TDRT-PPO Best Attack Rewards (↓) | SA-PPO Clean Rewards (↑) | SA-PPO Best Attack Rewards (↓) |
> | --- | --- | --- | --- | --- |
> | 0.03 | **4818 ± 9** | 1098 ± 192 | 4799 ± 42 | **809 ± 210** |
> | 0.1 | **4860 ± 1** | **792 ± 94** | 4801 ± 31 | 823 ± 194 |
> | 0.2 | **4843 ± 2** | **394 ± 10** | 4766 ± 31 | 670 ± 79 |
> | 0.3 | **4802 ± 27** | **378 ± 10** | 4161 ± 1537 | 403 ± 49 |
> | 0.5 | **4839 ± 25** | 413 ± 12 | 3984 ± 76 | **405 ± 24** |
>
> **TDRT-PPO achieves higher clean rewards than SA-PPO while maintaining robustness across a wide range of hyperparameter settings.**
>
> We will make this point explicit in the Experiments section.
>
> ---
>
> ### **Thank you again for taking the time to review our paper.**
>
> We hope this addresses your concerns, and we would greatly appreciate it if you could reconsider your score.

---

> ### Author Response · Authors · 2025-11-28
> **Follow-up on Rebuttal**
>
> Dear Reviewer 185Q,
>
> Thank you again for your thoughtful and constructive review. As the discussion period ends on December 2, we would greatly appreciate it if you could briefly review our responses and let us know whether they resolve your concerns or if any points remain unclear. We are happy to provide further clarification if needed.

---

### Meta-Review · Area_Chair_Zexn · 2026-01-15

**Summary:**

The paper introduces Behavior Imitation Attack (BIA), which learns behavior-targeted state-perturbation attacks from demonstrations under black-box/no-box access, and proposes Time-Discounted Robust Training (TDRT), a time-discounted policy-smoothing regularizer motivated by a bound that emphasizes early-trajectory sensitivity.

The paper has a clear threat-model framing and broad empirical coverage (Meta-World, MuJoCo, MiniGrid) with generally strong robustness/clean-reward trade-offs for TDRT. The initial reviews of the paper were mixed. The main weaknesses and concerns raised by the reviewers (and verified by the AC) include (1) the attack’s effectiveness degrades in high-dimensional observations, (2) the defense remains largely an empirical regularization without certified guarantees, (3) novelty vs. prior SA-RL/SA-PPO-style formulations.

The rebuttal improves clarity and strengthens the empirical story (including added comparisons and implementation fixes). Regarding novelty, the key theorem in the paper has a close connection to prior work, but the setting studied in this work is indeed a new and valid one. The lack of certified guarantees and high-dimensional scalability is also a common challenge in similar methods. Although the review score is on the borderline of acceptance, the AC believes the setting presented in this work is interesting enough, and the new evaluations and experiments added during the rebuttal period greatly strengthened the paper. The final decision is to accept the paper.

**Reviewer Concerns:**

Several concrete issues were convincingly addressed in the rebuttal and discussion, especially around clarity and missing checks. The authors clarified that TDRT is not “adversarial training with BIA” but a regularization-based smoothing method independent of the specific attack algorithm, which directly fixes a core misunderstanding from Reviewer 185Q. They also resolved presentation/reproducibility issues raised by Reviewer vKMc and added a direct discussion of computational cost trade-offs for BIA versus white-box baselines. In response to Reviewer kHJs and Yovk, they expanded related work and added at least one new comparison line to diffusion-based defenses, including reporting DMBP numbers against behavior-targeted attacks with mixed outcomes across tasks.

Some concerns remain only partially resolved. First and most importantly, concerns about novelty relative to SA-RL/SA-PPO remain. The rebuttal argues the objective/threat-model shift (demonstrations vs reward access; behavior-targeted gain vs reward-minimization) is the core contribution, but the key algorithmic pieces are indeed close adaptations of established methods. Second, a substantial limitation is the scalability and practicability of the setting. The paper has acknowledged that adversarial-policy attacks are less effective in high-dimensional state spaces. Given that prior published work, such as SA-PPO, also has a similar limitation, the AC believes it is ok to address the scalability in future work.

**Reviewer Scores:**

Reviewer 185Q (initial rating 4) had a misunderstanding about TDRT and raised novelty/fairness concerns; correcting the misunderstanding and pointing to broader hyperparameter comparisons could plausibly move them from a clear reject to a weak accept (5 or 6). Reviewer Yovk (initial rating 4) raised conceptual and evaluation concerns; the authors added extra experiments, expanded related works, and provided a diffusion-defense comparison. Most concerns were addressed and the reviewer could possibly move the score to at least 5 or 6.

---

### Decision · Program_Chairs · 2026-01-26

Accept (Poster)